# Lossy Compression for Lossless Prediction

**Yann Dubois**
Vector Institute
yanndubois96@gmail.com

**Benjamin Bloem-Reddy**
The University of British Columbia
benbr@stat.ubc.ca

**Karen Ullrich**
Facebook AI Research
karenu@fb.com

**Chris J. Maddison**
University of Toronto
Vector Institute
cmaddis@cs.toronto.edu

## Abstract

Most data is automatically collected and only ever "seen" by algorithms. Yet, data compressors preserve perceptual fidelity rather than just the information needed by algorithms performing downstream tasks. In this paper, we characterize the bit-rate required to ensure high performance on all predictive tasks that are invariant under a set of transformations, such as data augmentations. Based on our theory, we design unsupervised objectives for training neural compressors. Using these objectives, we train a generic image compressor that achieves substantial rate savings (more than $1000\times$ on ImageNet) compared to JPEG on 8 datasets, without decreasing downstream classification performance.

## 1   Introduction

Progress in important areas requires processing huge amounts of data. For climate prediction, models are still data-limited [1], despite the Natl. Center for Computational Sciences storing 32 million gigabytes (GB) of climate data [2]. For autonomous driving, capturing a realistic range of rare events with current methods requires around 3 trillion GB of data.[1] At these scales, data are only processed by task-specific algorithms, and storing data in human-readable formats can be prohibitive. We need compressors that retain only the information needed for algorithmic execution of downstream tasks.

Existing lossy compressors are not up to the challenge, because they aim to reconstruct the data for human perception [5–10]. However, much of perceptual information is not needed to perform the tasks that we care about. Consider classifying images, which can require about 1 MB to store. Classification is typically invariant under small image transformations, such as rescalings or rotations, and could instead be performed using a representation that discards such information (see Fig. 1). The amount of unnecessary perceptual information is likely substantial, as illustrated by the fact that typical image classification can be performed using a detailed caption, which requires only about 1 kB to store ($1000\times$ fewer bits).

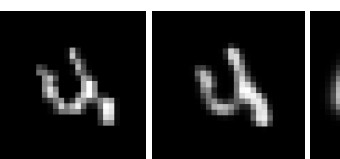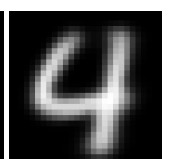

Source        Standard rec.        Our rec.

Figure 1: Our unsupervised coder improves compression by only keeping information necessary for typical tasks. (left) source augmented MNIST digit; (center) a neural perceptual compressor achieves 130 bit-rate; (right) our invariant compressor achieves 48 bit-rate.

---

[1]Kalra and Paddock [3] estimated that autonomous vehicles would have to drive hundreds of billions of miles to demonstrate reliability in rare events. At 1 TB / hour [4] and 30 miles / hour, this is 3e12 GB.

35th Conference on Neural Information Processing Systems (NeurIPS 2021).

Our goal is to quantify the bit-rate needed to ensure high performance on a collection of prediction tasks. In the simple case of a *single* supervised task, the minimum bit-rate is achieved by compressing predicted labels, and essentially corresponds to the Information Bottleneck (IB; [11]). Our challenge, instead, is to ensure good performance on *any* future tasks of interest, which will rarely be completely known at compression time, or might be too large to enumerate.

We overcome this challenge by focusing on sets of tasks that are *invariant* under user-defined transformations (e.g., translation, brightness, cropping), as is the case for many tasks of interest to humans [12, 13]. This structure allows us to characterize a worst-case invariant task, which bounds the relative predictive performance on all invariant tasks. As a result, the bit-rate required to perform well on *all* invariant tasks is exactly the rate to compress the worst-case labels. At a high level, the worst-case task is to recognize which examples are transformed versions of one another, and rate savings come from discarding information from those transformations.

We also provide two unsupervised neural compressors to target the optimal rates. One is similar to a variational autoencoder [14] that reconstructs canonical examples (Fig. 1). Our second is a simple modification of contrastive self-supervised learning (SSL; [15]), which allows us to convert pre-trained SSL models into powerful, generic compressors. Our contributions are:

- We formalize the notion of compression for downstream predictive tasks.
- We characterize the bits needed for high performance on any task invariant to augmentations.
- We provide unsupervised objectives to train compressors that approximate the optimal rates.
- We show that our compressor outperforms JPEG by orders of magnitude on 8 datasets on which it was never trained (i.e., zero-shot). E.g., on ImageNet [16], it decreases the bit-rate by $1000\times$.

## 2  Rate-distortion theory background

The goal of lossy compression theory is to find the number of bits (*bit-rate*) required to store outcomes $x$ of a random variable (r.v.)  $X$, so that it can be reconstructed within a certain tolerance. This is accomplished in Shannon's [17] rate-distortion (RD) theory by mapping $X$ into a r.v. $Z$ with low mutual information $I[X; Z]$. Specifically, given a distortion measure $D[X, Z]$, the RD theory characterizes the minimal achievable bit-rate for a distortion threshold $\delta$ by

$$Rate(\delta) = \min_{p(Z|X)} I[X; Z] \quad \text{such that} \quad D[X, Z] \leq \delta. \tag{1}$$

In lossy compression, $Z$ is usually a reconstruction of $X$, i.e., it aims to faithfully approximate $X$. As a result, typical distortions, e.g., the mean squared error (MSE), assume that the sample spaces $\mathcal{X}, \mathcal{Z}$ of both r.v.s are the same. This assumption is not required. Indeed, any distortion $d : \mathcal{X} \times \mathcal{Z} \to \mathbb{R}_{\geq 0}$ of the form $D[X, Z] = E_{p(X,Z)}[d(X, Z)]$, where there exists a $z \in Z$ such that $D[X, z]$ is finite, is a valid choice [18]. This shows that RD theory can be used outside of reconstructions. In the following we refer to $Z$ as a compressed *representation* of $X$ to distinguish it from a reconstruction.

## 3  Minimal bit-rate for high predictive performance

In this section, we characterize the bit-rate needed to represent $X$ to ensure high performance on downstream tasks. Our argument has three high-level steps: (i) define a distortion that controls downstream performance when predicting from $Z$ instead of $X$; (ii) simplify and validate this distortion when desired tasks satisfy an invariance condition; (iii) apply RD theory with the valid distortion. For simplicity, our presentation is relatively informal; formal proofs are in Apps. A and B.

### 3.1  A distortion for worst-case predictive performance

Suppose $X$ is an image. Potential downstream tasks might include $Y_{\text{dog}}$, whether the image displays a dog; or $Y_{\text{hd}}$, whether the image is hand-drawn. Formally, these and other downstream tasks are expressed as $\mathcal{T} = \{Y_{\text{dog}}, Y_{\text{hd}}, \ldots\}$, a set of random variables that are jointly distributed with $X$. Let $R[Y \mid X]$ denote the Bayes (best possible) risk when predicting $Y$ from $X$. For ease of presentation in the main paper, we consider only classification tasks $\mathcal{T}$ and Bayes risk of the standard log loss $R[Y \mid X] := \inf_q E_{p(X,Y)}[-\log q(Y|X)]$. We deal with MSE and regression in Appx. B.6.

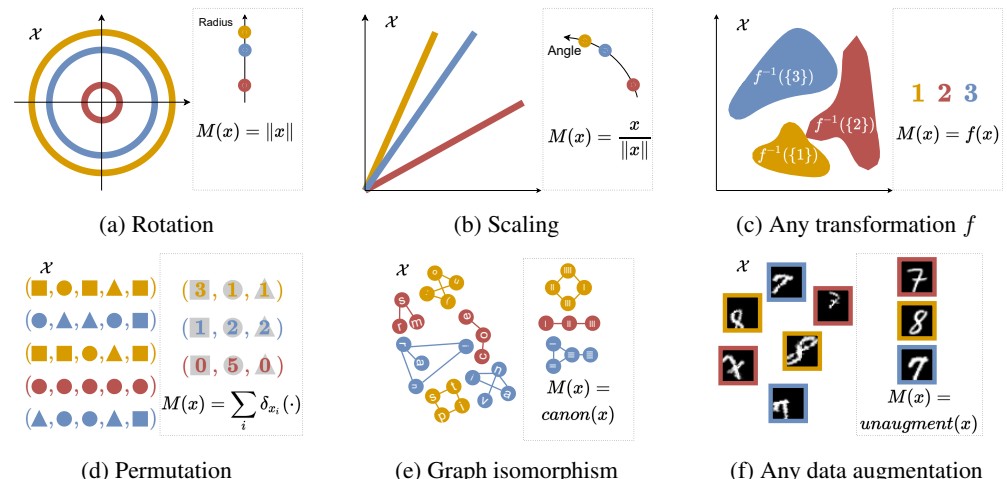

Figure 2: Maximal invariants $M(X)$ are representatives of equivalence classes. Example $M$s include the: (a) Euclidean norm for rotations; (b) unit vector for scaling; (c) $f$ when equivalence classes are pre-images by $f$; (d) empirical measure for permutations; (e) canonical graph for graph isomorphisms; (f) unaugmented input for data augmentations.

In this setting, a meaningful distortion $D_\mathcal{T}[X, Z]$ quantifies the difference between predicting any $Y \in \mathcal{T}$ from the compressed $Z$, as opposed to using $X$. This is the worst-case excess risk,

$$D_\mathcal{T}[X, Z] := \sup_{Y \in \mathcal{T}} \quad R[Y \mid Z] - R[Y \mid X]. \tag{2}$$

If $D_\mathcal{T}[X, Z] = 0$, it is possible to achieve *lossless prediction*: performing as well using $Z$ as using $X$. More generally, bounding $D_\mathcal{T}$ by $\delta$ ensures that $R[Y \mid Z] - R[Y \mid X] \leq \delta$ for all tasks in $\mathcal{T}$. However, there are two issues that need to be addressed before Eq. (2) can be used. First, it is not clear whether $D_\mathcal{T}$ is a valid distortion for RD theory. Second, the worst excess-risk $D_\mathcal{T}$ assumes access to all downstream tasks of interest $\mathcal{T}$ during compression, which is unrealistic in general.

## 3.2 Invariant tasks

The tasks that we care about are not arbitrary, and often share structure. One such structure is invariance to certain pre-specified transformations of input data. For example, computer vision tasks are often invariant to mild transformations such as brightness changes. Such invariance structure is common in realistic tasks, as seen by the wide-spread use of data augmentations [13] in machine learning (ML), which encourage predictions to be the same for an unaugmented $x$ and an augmented $x^+$. Motivated by this we focus on sets of invariant tasks $\mathcal{T}$.

We consider a general notion of invariance, namely invariance specified by an equivalence relation $\sim$ on $\mathcal{X}$.[2] The equivalence induces a partition of $\mathcal{X}$ into disjoint *equivalence classes*, and we are interested in tasks whose conditional distributions are constant within these classes.

**Definition 1.** The set of *invariant tasks of interest* with respect to an equivalence $(\mathcal{X}, \sim)$, denoted $\mathcal{T}_\sim$, is all random variables $Y$ such that $x \sim x^+ \implies p(Y \mid x) = p(Y \mid x^+)$ for any $x, x^+ \in \mathcal{X}$.

## 3.3 Rate-distortion theory for invariant task prediction

The key to simplifying $D_{\mathcal{T}_\sim}$ is the existence of a (non-unique) worst-case invariant task, denoted $M(X)$. Such task contains all and only information to which tasks $Y \in \mathcal{T}_\sim$ are not invariant; we call them *maximal invariants*. A maximal invariant $M(\bullet)$ with respect to $\sim$ is any function satisfying[3]

$$x \sim x^+ \iff M(x) = M(x^+) \quad \text{for any } x, x^+ \in \mathcal{X}. \tag{3}$$

---

[2]As a reminder, $\sim$ is an equivalence relation iff for all $x, x', x'' \in \mathcal{X}$: (reflexivity) $x \sim x$, (symmetry) $x \sim x' \iff x' \sim x$, and (transitivity) $x \sim x'$ and $x' \sim x'' \implies x \sim x''$.

[3]This extends the definition of maximal invariants [19] beyond invariances to group actions.

A maximal invariant removes all information that tasks are invariant to, as it maps equivalent inputs to the same output, i.e., $M(x) = M(x^+)$. Yet, it retains the minimal information needed to perform invariant tasks, by mapping non-equivalent inputs $x \not\sim x^-$ to different outputs $M(x) \neq M(x^-)$. In other words, $M(x)$ indexes the equivalence classes. For example, the Euclidean norm is a maximal invariant for rotation invariance, as all vectors that are rotated versions of one another can be characterized by their radial coordinate. For data augmentations, the canonical (unaugmented) version of the input is a maximal invariant. Other examples are shown in Fig. 2.

We prove in Appx. B.2 that under weak regularity conditions, maximal invariant tasks exist in $\mathcal{T}_\sim$, and that they achieve the supremum in Eq. (2). This allows us to show that $\mathrm{D}_{\mathcal{T}_\sim}$ reduces to the Bayes risk of predicting $M(X)$ from $Z$ and that it is a valid distortion measure. Crucially, this allows us to quantify downstream performance without enumerating invariant tasks.

**Proposition 1.** Let $(\mathcal{X}, \sim)$ be an equivalence relation and $M$ a maximal invariant that takes at most countably many values, with $\mathrm{H}[M(X)] < \infty$. Then $\mathrm{D}_{\mathcal{T}_\sim}$ (2) with log loss is a valid distortion and
$$\mathrm{D}_{\mathcal{T}_\sim}[X, Z] = \mathrm{R}[M(X) \,|\, Z] \;. \tag{4}$$

Here we used $\mathrm{R}[M(X) \,|\, X] = 0$, as $M$ is a deterministic function. Also, note that the countable requirement holds when tasks are invariant to some rounding of the input, as is typically the case due to floating-point storage. We accommodate the uncountable case for squared-error loss in Appx. B.6.

With a valid distortion in hand, we invoke the RD theorem with $\mathrm{D}_{\mathcal{T}_\sim}$ to obtain our "Rate-Invariance" (RI) theorem. The RI theorem characterizes the bit-rate needed to store $X$ while ensuring small log-loss on invariant tasks. We obtain analogous results for squared-error loss.

**Theorem 2** (Rate-Invariance). Assume the conditions of Prop. 1. Let $\delta \geq 0$, and $Rate(\delta)$ denote the minimum achievable bit-rate for transmitting $Z$ such that for any $Y \in \mathcal{T}_\sim$ we have $\mathrm{R}[Y \,|\, Z] - \mathrm{R}[Y \,|\, X] \leq \delta$. Then $Rate(\delta) = 0$ if $\delta \geq \mathrm{H}[M(X)]$ and otherwise it is finite and
$$Rate(\delta) \quad = \quad \underbrace{\mathrm{H}[M(X)]}_{\substack{\text{information needed} \\ \text{to predict } \mathcal{T}_\sim}} \quad \underbrace{- \delta}_{\substack{\text{acceptable decrease} \\ \text{in predictive loss}}} \quad = \quad \underbrace{\mathrm{H}[X]}_{\substack{\text{standard} \\ \text{compression}}} \quad \underbrace{- \mathrm{H}[X \,|\, M(X)]}_{\substack{\text{gains due to} \\ \text{invarainces}}} \quad - \delta. \tag{5}$$

To ensure lossless prediction, i.e., $Rate(0)$, our theorem states that we require a bit-rate of $\mathrm{H}[M(X)]$. Intuitively, this is because $M(X)$ contains the minimal information needed to predict losslessly any $Y \in \mathcal{T}_\sim$.[4] Furthermore, the theorem relates compression and prediction by showing that allowing a $\delta$ decrease in log-loss performance on all tasks can save *exactly* $\delta$ bits. Intuitively, this is a linear relationship, because expected log-loss is measured in bits. On the right of Eq. (5) we further decompose $\mathrm{H}[M(X)]$ into two terms to provide another interpretation: (i) $\mathrm{H}[X]$, which, for discrete $X$, is the bit-rate required to losslessly compress $X$, and (ii) $\mathrm{H}[X \,|\, M(X)]$, which quantifies the information removed due to the invariance of desired tasks. Importantly, removing this information does not impact the best possible predictive performance. See Fig. 3.

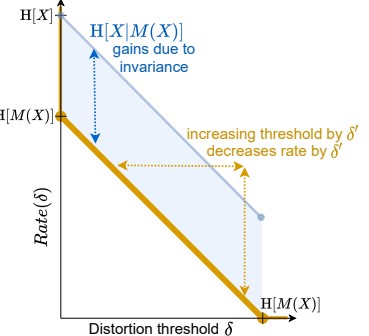

Figure 3: Rate-Invariance function.

The bit-rate gains can be substantial, depending on the invariances. Consider compressing a sequence of $n$ i.i.d. fair coin flips. Suppose one is only interested in predicting permutation invariant labels. Then instead of compressing the entire sequence in $\mathrm{H}[X^n] = n\,\mathrm{H}[X] = n$ bits, one could compress the number of heads, which is a maximal invariant for permutation invariance, in $\mathcal{O}(\log n)$ bits.[5] As more interesting examples, we recover in Appx. B.4 results from (i) unlabeled graph compression [20]; (ii) multiset compression [21]; (iii) single task compression (IB; [11]). The equivalence $\sim$ can be induced by any transformations, such as transforming an image to its caption. We use this idea in Sec. 5.3 to obtain $>1000\times$ compression on ImageNet without sacrificing predictive performance.

---

[4] We prove in Appx. B.5 that $Rate(0) = \mathrm{H}[M(X)]$ for any losses used in ML.

[5] The number of heads $K$ follow a binomial distribution so $\mathrm{H}[K] \in \mathcal{O}(\log n)$. Here $K$ is also a minimal sufficient statistic for $X^n$. More generally, if $P(X')$ is invariant to $\sim$ and $\sim$ is the coarsest such relation, then minimal sufficiency coincides with maximal invariance. In practice, however, $P(X')$ will rarely be $\sim$ invariant.

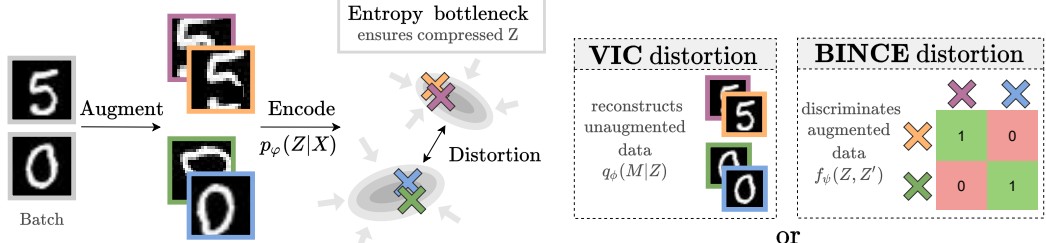

Figure 4: Our unsupervised objectives for invariant image compression under data augmentation use the same encoder, but differ in their approximation to the invariance distortion. Both models encode the augmented data, pass the representation through an entropy bottleneck which ensures that they are compressed, and use a distortion to retain the information about the identity of the original data. The models differ in how they retain that information: (VIC) by reconstructing unaugmented inputs; (BINCE) by recognizing which inputs come from the same original data.

## 4 Unsupervised training of invariant neural compressors

In this section, we design practical, invariant neural compressors that bound optimal rates. Derivations are in Appx. C. In particular, we are interested in the $\arg\min$ encoders $p(Z|X)$ of the RD function (Eq. (1)) under the invariance distortion $D_{\tau_\sim}$. To accomplish this, we can optimize the following equivalent (i.e., it induces the same RI function) Lagrangian, where $\beta$ takes the role of $\delta$,[6]

$$\underset{p(Z|X)}{\arg\min} \quad I[X;Z] + \beta \cdot R[M(X)\,|\,Z]. \tag{6}$$

In ML, the maximal invariant $M$ is often not available. Instead, invariances are implicitly specified by sampling a random augmentation from $A$, applying it to a datapoint $X$, and asking that the model's prediction be invariant between $X$ and $A(X)$. For example, invariance to cropping can be enforced by randomly cropping images while retaining the original label. We show in Appx. C, that in such case, we can treat the augmented $A(X)$ as the new source, $Z$ as the representation of $A(X)$, and the unaugmented $X$ as the maximal invariant task $M(A(X))$. Indeed, $R[M(A(X))\,|\,Z]$ is equal to $R[X\,|\,Z]$ up to a constant, so we can rewrite Eq. (6) as the following equivalent objective,

$$\underset{p(Z|A(X))}{\arg\min} \quad I[A(X);Z] + \beta \cdot R[X\,|\,Z]. \tag{7}$$

Such reformulation is possible if random augmentations retain the invariance structure $X \sim A(X)$ but "erase" enough information about equivalent inputs, specifically, if $X \perp\!\!\!\perp A(X)\,|\,M(X)$. We discuss the second requirement in Appx. C but note that it will likely not be a practical issue if the dataset is small compared to the support $|\mathcal{D}| \ll |\mathcal{X}|$. With this, we have an objective whose r.v.s. are easy to sample from. However, both terms in Eq. (7) are still challenging to estimate.

In the following, we develop two practical variational bounds to Eq. (7), which can be optimized by stochastic gradient descent [23] over the encoder's parameters. Both approximations use the standard lossy neural compression bound $I[Z;A(X)] \leq H[Z] \leq \min_\theta E_{p(Z)}[-\log q_\theta(Z)]$ where $q_\theta(Z)$ is called an entropy model (or a prior) [24, 25]. This has the advantage that the learned $q_\theta$ can be used for entropy coding $Z$ [26, 27]. See Ballé et al. [28] for possible entropy models. Our two approximations differ in how they upper bound $R[X\,|\,Z]$. The first uses a reconstruction loss, which attempts to reconstruct the unaugmented input $x \in \mathcal{D}$ from $A(x)$. The second uses a discrimination loss, which attempts to recognize which examples are augmented versions of the input.

### 4.1 Variational Invariant Compressor (VIC)

Our first model is a modified neural compressor in which inputs are augmented but target reconstructions are not. We refer to it as a *variational invariant compressor* (VIC). See Fig. 4 for an illustration. VIC has an encoder $p_\varphi(Z|A(X))$, an entropy model $q_\theta(Z)$, and a decoder $q_\phi(X|Z)$. Given a data sample $x \in \mathcal{D}$, we apply a random augmentation $A(x)$, and encode it to get a representation $Z$. The decoder then attempts to reconstruct the unaugmented $x$ from $Z$. This leads to the objective,

$$\mathcal{L}_{\text{VIC}}(\phi,\theta,\varphi) := -\sum_{x\in\mathcal{D}} E_{p(A)p_\varphi(Z|A(x))}[\log q_\theta(Z) + \beta \cdot \log q_\phi(x\,|\,Z)]. \tag{8}$$

---

[6]As the RI function is not strictly convex (Fig. 3), it should be beneficial to use $R[M(X)\,|\,Z]^2$ to ensure that sweeping over $\beta$ is equivalent to sweeping over $\delta$ [22]. We did not see any difference in practice.

The term $\log q_\theta(Z)$ is an entropy bottleneck, which bounds the rate $\mathrm{I}[A(X); Z]$ and ensures that unnecessary information is removed. The term $\log q_\phi(x \mid Z)$ bounds the distortion $\mathrm{R}[X \mid Z] \leq \mathrm{E}_{p(X,Z)}[-\log q_\phi(X \mid Z)]$ and ensures that VIC preserves the information needed for invariant tasks.

## 4.2 Bottleneck InfoNCE (BINCE)

Our second compressor retains all predictive information without reconstructing the data. It has two components: an entropy bottleneck and an InfoNCE [15] objective, which is the standard in contrastive SSL. We refer to this as the *bottleneck InfoNCE* (BINCE), see Fig. 4. BINCE has an advantage over VIC in that it avoids the problem of reconstructing possibly high dimensional data.

Algorithm 1 shows how to train BINCE, where each call to $A$ returns an independent augmentation of its input. As with VIC, for every datapoint $x \in \mathcal{D}$, we obtain a representation $Z$ by applying an augmentation $A(x)$ and passing it through the encoder $p_\varphi(Z \mid A(X))$. We then sample a "positive" example $Z^+$ by encoding a different augmented version of the same underlying datapoint $x$. Finally, we sample $n$ "negative" examples $Z_i^-$ by encoding augmentations $A(x_i^-)$ of datapoints $x_i^- \in \mathcal{D}$ that are different from $x$. This results in a sequence $\boldsymbol{Z} = (Z^+, Z_1^-, \ldots, Z_n^-)$. For conciseness we will denote the above sampling procedure as $p_\varphi(Z, \boldsymbol{Z} \mid A, \mathcal{D}, x)$. The final loss uses a discriminator $f_\psi$ that is optimized to score the equivalence of two representation,

---

**Algorithm 1** BINCE's forward pass for $x$

---

**Require:** $p_\varphi, q_\theta, f_\psi, \mathcal{D}, A, \beta, n, x$
1: $\tilde{x} \leftarrow \text{sample}(A(x))$          ▷ Augment
2: $z \leftarrow \text{sample}(p_\varphi(Z|\tilde{x}))$          ▷ Encode
3: $\text{rate\_loss} \leftarrow -\log q_\theta(z)$
4: $\{x_i^-\}_{i=1}^n \leftarrow \text{select}(\mathcal{D} \setminus \{x\})$ $n$ times
5: $\tilde{\mathbf{x}} \leftarrow \text{sample}([A(x), A(x_1^-), \ldots, A(x_n^-)])$
6: $\mathbf{z} \leftarrow \text{sample}(p_\varphi(Z|\tilde{\mathbf{x}}))$
7: $z^+ \leftarrow \mathbf{z}[0]$
8: $\text{softmax} \leftarrow \frac{\exp f_\psi(z^+, z)}{(\sum_{z' \in \mathbf{z}} \exp f_\psi(z', z))}$
9: $\text{distortion\_loss} \leftarrow -\log(\text{softmax})$
10: **return** $\text{rate\_loss} + \beta \cdot \text{distortion\_loss}$

---

$$\mathcal{L}_{\text{BINCE}}(\varphi, \theta, \psi) := -\sum_{x \in \mathcal{D}} \mathrm{E}_{p(A)p_\varphi(Z, \boldsymbol{Z}|A, \mathcal{D}, x)} \left[ \log q_\theta(Z) + \beta \cdot \log \frac{\exp f_\psi(Z^+, Z)}{\sum_{Z' \in \mathbf{Z}} \exp f_\psi(Z', Z)} \right]. \quad (9)$$

BINCE retains the necessary information by classifying (as seen by the softmax) which $Z$ is associated with an equivalent example $X$. Both VIC and BINCE give rise to efficient compressors by passing $X$ through $p_\varphi(Z|X)$ and entropy coding using $q_\theta(Z)$. In theory they can both recover the optimal rate for lossless predictions, i.e., $\mathrm{H}[M(X)]$, in the limit of infinite samples ($|\mathcal{D}|$,$n$) and unconstrained variational families. In practice, BINCE has the advantage over VIC of (i) not requiring a high dimensional decoder; and (ii) giving (for suitable $f_\psi$) representations that are approximately linearly separable [29–31] and thus easy to predict from [15, 32]. The disadvantages of BINCE are that it (i) does not provide to reconstructions diminishes interpretability; and (ii) has a high bias, unless the number of negative samples $n$ is large [33, 34], which is computationally intensive.

## 5 Experiments

We evaluated our framework focusing on two questions: (i) What compression rates can our framework achieve at what cost? (ii) Can we train a general purpose predictive image compressor? For all experiments, we train the compressors, freeze them, train the downstream predictors, and finally evaluate both on a test set. For classical compressors, standard neural compressors (VC) and our VIC, we used either reconstructions $\tilde{X}$ as inputs to the predictors or representations $Z$. As BINCE does not provide reconstructions, we predicted from the compressed $Z$ using a multi-layer perceptron (MLP). We used ResNet18 [35] for encoders and image predictors. For entropy models we used Ballé et al.'s [28] hyperprior, which uses uniform quantization. We optimized hyper-parameters on validation using random search. For classification tasks, we report classification error instead of log-loss. The former is more standard and gave similar results (see Appx. F.2). For experimental details see Appx. E. For additional results see Appx. F. Code is at `github.com/YannDubs/lossyless`.

### 5.1 Building intuition with toy experiments

To build an visual intuition, we compressed samples from a 2D banana source distribution [36], assuming rotation invariant tasks, e.g., classifying whether points are in the unit circle. We also compressed MNIST digits as in Fig. 1. Digits are augmented (rotations, translations, shearing, scaling) both at train and test time to ensure that our invariance assumption still holds.

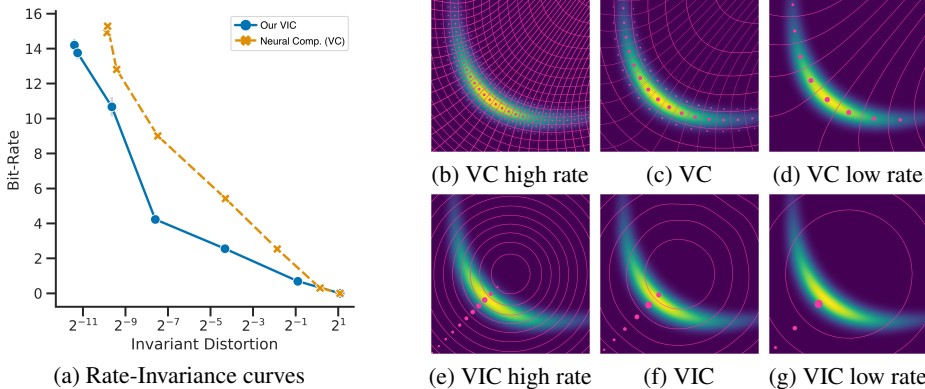

(a) Rate-Invariance curves     (b) VC high rate     (c) VC     (d) VC low rate

(e) VIC high rate     (f) VIC     (g) VIC low rate

Figure 5: Compression rates of a Banana source [36] can be decreased when downstream tasks are rotation invariant. (Left) Our invariant compressor (VIC, blue) outperforms neural compressors (VC, orange). 5 runs with standard errors in gray. (Right) VIC quantizes the space using disks to remove unnecessary angular information. Pink lines are quantization boundaries, dots are code vectors with size proportional to learned probabilities. Low rates correspond to low $\beta$ in Eq. (7).

**Where do our rate gains come from?** For rotation invariant tasks, our method (VIC) discards unnecessary angular information by learning disk-shaped quantizations (Fig. 5, bottom right). Specifically, VIC retains only radial information by mapping all randomly rotated points (disks) back to maximal invariants (pink dots). In contrast, standard neural compressors (VC) attempt to reconstruct all information, which requires a finer partition (Fig. 5, top right). As a result (Fig. 5a), VIC needs a smaller bit-rate ($y$-axis) for the same desired performance ($D_{\mathcal{T}_\sim}$, $x$-axis). The area under the RD curve (AURD) for VIC is $35.8\pm4.2$ against $48.1\pm0.3$ for VC, i.e., expected bit-rate gains are around $70\%$. Similar gains are achieved for augmented MNIST in Fig. 6 by reconstructing canonical digits.

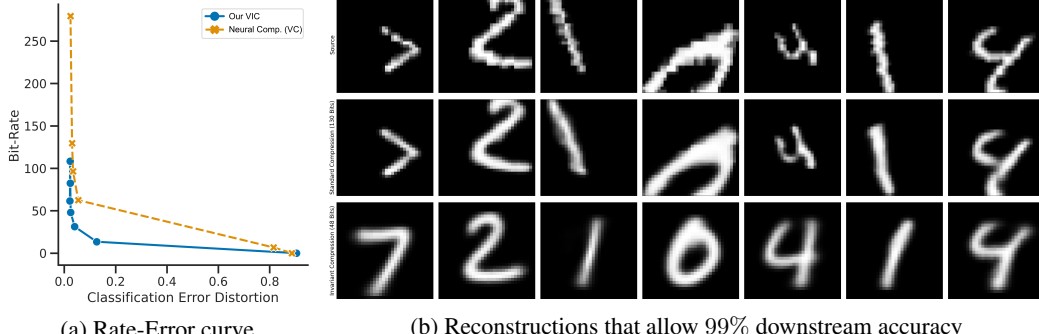

(a) Rate-Error curve        (b) Reconstructions that allow $99\%$ downstream accuracy

Figure 6: (Left) By reconstructing prototypical digits our VIC (blue) achieves higher compression of augmented MNIST digits than standard neural compressors (VC, orange) without hindering downstream classification. 5 runs. (Right) The source examples (first row) as well as reconstructions for the non-invariant (second row) and invariant compressor (last row).

**Can we recover the optimal bit-rate?** We investigated whether our losses can achieve the optimal bit-rate for lossy prediction by using supervised augmentations, i.e., $A(x)$ randomly samples a train example $x^+$ that has the same label. For MNIST the single-task optimal bit-rate is $H[Y] = \log(10) \approx 3.3$ bits. VIC and BINCE respectively achieve $5.7$ and $5.9$ bits, which shows that our losses are relatively good despite practical approximations. Details in Appx. F.2.

**What is the impact of the choice of augmentations?** The choice of augmentation $A$ implicitly defines the desired task-set $\mathcal{T}$, i.e., $\mathcal{T}$ is the set of all tasks for which $A$ does not remove information. As a result Theorem 2 can be rewritten as $Rate(\delta) = I[X; A(X)] - \delta$, so the rate decreases when $A$ removes more information from $X$. To illustrate this we trained our VIC using three augmentation sets on MNIST, all of which keep the true label invariant but progressively discard more $X$ information. VIC respectively achieves a rate of $185.3$, $79.0$, and $5.7$ bits, which shows the importance of using augmentations that remove $X$ information. Details and BINCE results are in Appx. F.2.

## 5.2 Evaluating our methods with controlled experiments

To investigate our methods, we compressed the STL10 dataset [37]. We augment (flipping, color jittering, cropping) the train *and* test set, to ensure that the task invariance assumptions are satisfied. We focus on more realistic settings in the next section. In each experiment, we sampled 100 combinations of hyper-parameters to ensure equal computational budget across models and baselines.

Table 1: Invariant compressors (BINCE, VIC) outperform classical (PNG, JPEG, WebP) and neural (VC) compressors on STL10. BINCE achieves lossless prediction but compresses $121\times$ better.

|  | PNG [38] | JPEG [39] | WebP [40] | VC $\tilde{X}$ | VIC $\tilde{X}$ | VIC $Z$ | **BINCE** |
|---|---|---|---|---|---|---|---|
| Decrease in test acc. | 0 | 0.7 | 1.1 | 21.0 | 25.1 | 16.1 | 0.0 |
| Compression gains | $1\times$ | $3\times$ | $13\times$ | $63\times$ | $269\times$ | $175\times$ | $121\times$ |

**How do our BINCE and VIC compare to standard compressors?** In Table 1 we compare compressors at the lowest downstream error that they achieved. As benchmark, we use PNG's lossless compression. Predicting from PNG corresponds to standard image classification, and obtains a rate of $1.42\mathrm{e}4$ bits per image for $80.8\%$ accuracy. Classical lossy methods (JPEG, WebP) achieved up to $13\times$ bit-rate gains with little drop in performance. In comparison, our BINCE method achieved $121\times$ compression gains with no impact on predictions. Both our invariant (VIC) and standard (VC) neural compressors significantly decreased classification accuracy, which we believe can be explained by the encoders architecture (ResNet18) that we use for consistency (see Appx. F.3).

**Should we predict from representations $Z$ or reconstructions $X$?** In Table 1 we analyzed the impact of predicting from $Z$ instead of $\tilde{X}$ for VIC and see that this increases accuracy by $9\%$. In contrast, predicting from $Z$ for VC decreases performance by $12\%$ (see Appx. F.3). This suggests that invariant reconstructions $\tilde{X}$ might not be easy to predict from with standard image predictors.

**Are we learning invariant compressors?** Invariant compressors should provide RD curves that are robust to test distribution shift in the desired augmentations. We thus trained our VIC by applying the augmentations $50\%$ of the time but varying that probability $p$ at test time. In Appx. F.3 we show that this distribution shift have negligible influence on RD curves.

## 5.3 A zero-shot compressor using pre-trained self-supervised models

BINCE includes a standard contrastive SSL loss. So, we investigated whether existing pre-trained SSL models [32, 41] can be used to build generic compressors. In particular, we investigated whether CLIP [41] could be quickly turned into a powerful task-centric compressor for computer vision. In the introduction, we motivated large compression gains by noting that typical image classification tasks can be predicted from detailed captions instead of images (around $1000\times$ more bits). CLIP is a vision transformer [42] pre-trained on 400M pairs of images and text $(x_{image}, x_{text}^+)$ using a contrastive loss. The "augmentation" $A$ is then a function that maps $x_{image}$ to its associated $x_{text}^+$ and vis-versa. This will partition the images and texts into sets, each of which are associated directly or by transitivity in CLIP's dataset. This suggests that CLIP is retaining the image information that corresponds to a detailed caption, and may be turned into a generic compressor for image classification.

CLIP can essentially be seen as a BINCE model with an image-to-text augmentation, but without an entropy bottleneck. (For details about the CLIP-BINCE relation see Appx. C.5.) We thus constructed an approximation of our desired image-to-text BINCE compressor by two simple steps. First, we downloaded and froze CLIP's parameters. Second, we trained, on the small MSCOCO dataset [43], an entropy bottleneck to compress CLIP's representation. The latter step can be done by training any lossy compressor on CLIP's representations, we did so using Ballé et al.'s [28] hyperprior entropy model with a learned rounded precision. We then evaluated our resulting compressor on 8 datasets (various classification tasks and image shapes) that were never seen during training (zero-shot), by training an MLP for downstream predictions on each dataset. One can see this as a multi-task setting (each dataset is a distinct task). We investigate the case of multiple labels per images in Appx. F.5.

**Can we use pretrained SSL to obtain a generic compressor?** Table 2 shows that we can exploit existing state-of-the-art (SOTA) SSL models to get a powerful image compressor, which achieves $1000\times$ bit-rate gains on ImageNet compared to JPEG (at the quality level used for storing ImageNet). The bit-rate gains (1st row) are significant across all zero-shot datasets, even for biological tissues (PCam; [44]). Importantly, these gains come at little cost in test performance. Indeed, the test

Table 2: Converting a pretrained SSL model into a zero-shot compressor achieves substantial bit-rate gains while allowing test accuracies similar to supervised models predicting from raw images.

|  | ImageNet | STL | PCam | Cars | CIFAR10 | Food | Pets | Caltech |
|---|---|---|---|---|---|---|---|---|
| Rate gains vs JPEG | 1104× | 35× | 64× | 131× | 7× | 109× | 150× | 126× |
| Our Acc. [%] | 76.3 | 98.7 | 80.9 | 79.6 | 95.2 | 88.3 | 89.5 | 93.4 |
| Supervised Acc. [%] | 76.1 | 99.0 | 82.6 | 49.1 | 96.7 | 81.8 | 90.4 | 94.5 |

accuracies of MLPs from our representations (2nd row) is similar to a near SOTA model trained on the uncompressed images (3rd row is from Radford et al. [41]). These results are not surprising as JPEG is optimized to retain perceptual rather than classification information. Note that the large variance in rate gains come from JPEG rates due to different images shapes (see Table 3).

**Our CLIP compressor retains all the information needed to get 0 error for those tasks.** Table 2 provides the test performance for MLPs, while our theory discusses Bayes risk, which is independent of specific predictors and generalization. We estimated the excess Bayes risk for our datasets by counting the images (in train and test) that get compressed to the same $Z$ but have different labels. We found that we are in the lossless prediction regime for those datasets.

Table 3: Our entropy bottleneck (EB) on CLIP improves compression of representations up to $17\times$ with little impact on predictions. The same compressor is used across datasets. Rates are per image.

|  |  | ImageNet | STL | PCam | Cars | CIFAR10 | Food | Pets | Caltech |
|---|---|---|---|---|---|---|---|---|---|
| **Bit-Rate** | JPEG | 1.49e6 | 4.71e4 | 9.60e4 | 1.92e5 | 1.05e4 | 1.54e5 | 1.81e5 | 1.69e5 |
|  | CLIP | 1.52e4 | 1.52e4 | 1.52e4 | 1.52e4 | 1.52e4 | 1.52e4 | 1.52e4 | 1.52e4 |
|  | +EB high $\beta$ | 2.47e3 | 2.46e3 | 2.61e3 | 2.59e3 | 2.53e3 | 2.39e3 | 2.33e3 | 2.46e3 |
|  | +EB $\beta$ | 1.35e3 | 1.34e3 | 1.49e3 | 1.47e3 | 1.41e3 | 1.27e3 | 1.21e3 | 1.34e3 |
|  | +EB low $\beta$ | 9.63e2 | 9.52e2 | 1.09e3 | 1.07e3 | 1.02e3 | 8.89e2 | 8.35e2 | 9.53e2 |
| **Test Acc.** | CLIP | 76.5 | 98.6 | 84.5 | 80.8 | 95.3 | 88.5 | 89.7 | 93.2 |
|  | +EB high $\beta$ | 76.6 | 98.7 | 82.7 | 80.4 | 95.3 | 88.5 | 89.6 | 93.5 |
|  | +EB $\beta$ | 76.3 | 98.7 | 80.9 | 79.6 | 95.2 | 88.3 | 89.5 | 93.4 |
|  | +EB low $\beta$ | 76.0 | 98.7 | 80.1 | 78.9 | 94.8 | 87.6 | 88.6 | 92.9 |

**What is the effect of the entropy bottleneck?** In Table 3 we compare the pretrained CLIP, to our CLIP compressor with an entropy bottleneck (EB) trained at different values for $\beta$. When trained with a high $\beta$, our EB improves bit-rates by an average of $6\times$ without impacting predictions. For our compressor from Table 2 (CLIP+EB $\beta$) the gains increase to $11\times$ with little predictive impact. The sacrifice in predictions is more clear for $16\times$ bit-rate gains (low $\beta$). This shows that CLIP's raw representations retain unnecessary information as it not explicitly trained to discard information.

**How would end-to-end BINCE compare to staggered training?** Compression gains can likely be larger by end-to-end training of BINCE, which would require access to CLIP's original dataset.[7] To get an idea of potential gains we compared end-to-end and staggered BINCE on augmented MNIST in Appx. F.2. We found significant rate improvements (358 to 131 bits) for similar test accuracy.

**Our CLIP compressor is simple to use.** In Appx. E.7, we provide a minimal script (150 lines) to train a generic compressor in less than five minutes on a single GPU. The script contains an efficient entropy coder for our model (200 images/second), which shows its practicality. As usual in SSL, the compressed representations are also more computationally efficient to work with than standard compressors. In our minimal script we achieve the desired performance (98.7% on STL) using a linear model that is trained in one second, which is $1000\times$ faster than the baseline in Table 2. This shows that our pipeline can improve computational efficiency in addition to storage efficiency.

**What augmentations to use for SSL compression?** Table 4 compares two ResNet50 pretrained with contrastive learning using invariance to text-image (CLIP) or standard image augmentations (SimCLR [32]) such as cropping or flipping. We see that CLIP's augmentation usually give better compression and downstream performance, which shows the importance of the choice of augmentations. This also supports our motivation of using text-image augmentations, which are likely label-preserving for a vast amount of tasks but discard large amounts of unnecessary information.

---

[7]We investigated finetuning CLIP on MSCOCO but it suffered from catastrophic forgetting.

Table 4: Text-image invariance is better than invariance to standard augmentations for image classification. CLIP and SimCLR are both ResNet50 pretrained with InfoNCE but different augmentations.

|  |  | ImageNet | STL | PCam | Cars | CIFAR10 | Food | Pets | Caltech |
|---|---|---|---|---|---|---|---|---|---|
| Rate | CLIP+EB | 2108 | 1962 | 1949 | 2421 | 2111 | 1991 | 1867 | 1968 |
|  | SimCLR+EB | 2811 | 2732 | 2769 | 2751 | 2950 | 2077 | 2839 | 2502 |
| Acc. | CLIP+EB | 63.2 | 92.0 | 78.6 | 68.0 | 65.5 | 74.1 | 81.8 | 83.0 |
|  | SimCLR+EB | 62.8 | 91.9 | 81.4 | 29.6 | 78.6 | 60.0 | 78.9 | 79.0 |

# 6  Related work

In Appx. D we discuss more related work, including invariances in compression and the link to SSL.

**Task-centric compression.** To our knowledge, our paper is the first to formalize compression only for predictions. IB [11] uses a task-centric distortion, but is not used for compression as it requires supervised training, so there are no advantages compared to compressing predicted labels. Some authors used heuristics to bypass the supervised issue, e.g., focusing on low frequencies for classification [45] or high frequencies for segmentation [46]. Other authors have incorporated predictive errors to perceptual distortions [47, 48], but cannot compress without the perceptual distortion for the same reason as IB. One exception is Weber et al.'s [49] compressor, which (when removing their perceptual distortion) minimizes MSE in the hidden layers of a pretrained classifier. Even more related is Singh et al.'s [50] work on compressing pretrained features for transfer learning, which is practice is similar to our compression of SSL features. Their work do not provide theoretical justifications, and are constrained to tasks that are similar to those used for pretraining.

# 7  Discussion and Outlook

Given the ever increasing amount of data that is processed by task-specific algorithms, it is necessary to rethink the current task-agnostic compression paradigm. We formalized the first compression framework for retaining only the information necessary for high performance on desired tasks. Using our theory, we provide two unsupervised objectives for training neural compressors. Experimentally, we show that these compressors can achieve bit-rates that are orders of magnitude ($1000\times$ on ImageNet) smaller than standard image compressors without losing predictive performance.

There are a number of caveats that should be addressed. First, to achieve better rates, our theory requires an irrecoverable loss of information. This can be an issue if the set of desired tasks changes. For example, if one uses text-image invariances then it may be impossible to perform image segmentation from the compressed representations. One solution would be to keep an original copy and use invariant compression for duplicated data, e.g., for the thousands copies of ImageNet. A second issue is the interpretability of the compressed representations. This can be partially addressed by reconstructing prototypical data as in Fig. 1 (post-hoc decoders could be trained for BINCE). A third caveat is that the compressed representations may be harder to learn from, e.g., neural networks may struggle to predict from representations even if the information is retained. Although our experiments actually showed the opposite, this should be addressed theoretically, e.g., using decodable information [51, 52]. Finally, successful use of our framework requires access to label-preserving augmentations $A$ that discard significant information about $X$. Finding such an $A$ may be challenging for some tasks. Given that augmentations are ubiquitous in ML, the community will hopefully continue developing task-specific augmentations which we could take advantage of.

Nevertheless, we achieved orders of magnitude improvements in compression for predictions, and we believe that our improvements are just the beginning. For example, many tasks can be answered by referencing a detailed natural language description of the data. In these cases, the improvements can be very large, potentially $1M\times$ for videos.[8] In the long-term, we hope that abandoning perceptual reconstructions will enable individuals to process data at scales that are currently only possible at large institutions, and our society to take advantage of large data sources in a more sustainable way.

---

[8] A movie takes around 10GB to store, but the information relevant to humans (e.g., "what happened to the house?", "how did the movie end?") can likely be stored in a detailed movie script and would require 100KB.

## Acknowledgments and Disclosure of Funding

We would like to thank Alex Alemi, David Duvenaud, Andriy Mnih, Emile Mathieu, Jonah Philion, Yangjun Ruan, and Ilya Sutskever for their helpful feedback and encouragements. Resources used in preparing this research were provided, in part, by the Province of Ontario, the Government of Canada through CIFAR, and companies sponsoring the Vector Institute. BBR acknowledges the support of the Natural Sciences and Engineering Research Council of Canada (NSERC): RGPIN-2020-04995, RGPAS-2020-00095, DGECR-2020-00343.

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
