# Supplementary Material:
# Lossy Compression for Lossless Prediction

## Table of Contents

# A Preliminaries

## A.1 Notation

**Probability**    We assume a background standard probability space $(\Omega, \mathcal{H}, \mathbb{P})$ that is rich enough to support all random variables used. Letters that are upper-case $X$ represent a random variable, while realizations are denoted with the associated lower case $x$. The sample space of a random variable will be written using a calligraphic $\mathcal{X}$, and we will say that $X$ takes value in (t.v.i) $\mathcal{X}$. We denote the probability distribution of $X$ as $P(X)$ and the probability density function, if it exists, as $p(X)$. $X \stackrel{d}{\sim} \mathcal{N}(0, 1)$ denotes that $X$ has a certain distribution (here, Gaussian). Expectations are written as: $\mathrm{E}_{P(X)}[X]$, or $\mathrm{E}_{p(X)}[X]$ when the density exists. Independence between two random variables $X$ and $Y$ is denoted with $X \perp\!\!\!\perp Y$. To denote conditional independence between two random variables $X$ and $Y$ given $Z$ we either use $X \perp\!\!\!\perp Y \mid Z$ or say that $X - Z - Y$ forms a Markov Chain. $f \circ g$ denotes a composition of two functions $f$ and $g$, but in the case of random variable we also use the shorthand $f(X) := f \circ X$.

**Information theory**    For notational convenience (see Assumption 5 below), when dealing with log loss we will always assume the existence of probability densities, in which case the KL divergence between two probability distributions on $\mathcal{X}$, $P$ and $Q$, is $\mathrm{D_{KL}}[p(X) \| q(X)] := \int p(X) \log \frac{p(X)}{q(X)} \, \mathrm{d}x$. The mutual information between random variables $X$ and $Z$ is $\mathrm{I}[X; Z] := \mathrm{D_{KL}}[p(X, Z) \| p(X) p(Z)]$. The (differential or discrete) entropy of a random variable is $\mathrm{H}[X] := \mathrm{E}_{P(X)}[-\log p(X)]$, while the conditional (differential) entropy is $\mathrm{H}[X \mid Z] := \mathrm{E}_{P(X,Z)}[-\log p(X|Z)]$.

**Equivalence**    $x \sim x'$ denotes that $x$ and $x'$ are equivalent with respect to (w.r.t.) an equivalence relation on $\mathcal{X}$ (the exact relation being implicit). The equivalence class of $x$ under $\sim$ consist of all elements that are equivalent to $x$, i.e. $[x] := \{x' \in \mathcal{X} \mid x' \sim x\}$. The set of all equivalence classes (the quotient set) will be denoted as $\mathcal{X}/\!\sim \, := \{[x] \mid x \in \mathcal{X}\}$, while the canonical projection is denoted as $\pi_\sim : x \mapsto [x]$.

**Risk minimization**    We will often use variational optimization. When the variational family is not made explicit it means that the optimization is over all functions with the correct domain and codomain, e.g. $\min_{q(Y \mid X)}$ means that that the optimization is done over the collection of all conditional probability densities on $\mathcal{Y}$ given the random variable $X$.

For a fixed "action" or "decision" space $\mathcal{A}$, a loss function is defined as $L : \mathcal{Y} \times \mathcal{A} \to \mathbb{R}$. The (expected) risk of a predictor $h : \mathcal{X} \to \mathcal{A}$ is $\mathrm{E}_{P(X,Y)}[L(Y, h(X))]$. The Bayes (best achievable) risk when predicting $Y$ from $X$ using some (unspecified) loss is denoted as $\mathrm{R}[Y \mid X]$. When the loss $L$ is specified, we denote the Bayes risk as $\mathrm{R_L}[Y \mid X] := \inf_{h : \mathcal{X} \to \mathcal{A}} \mathrm{E}_{P(X,Y)}[L(Y, h(X))]$. For the case of log loss (always assumed in the main text) we have $\mathrm{R_{log}}[Y \mid X] := \inf_q \mathrm{E}_{P(X,Y)}[-\log q(Y|X)]$. For MSE loss we have $\mathrm{R_{mse}}[Y \mid X] := \inf_{f : \mathcal{X} \to \mathcal{Y}} \mathrm{E}_{P(X,Y)}[\|Y - f(X)\|^2]$. Letters $X$, $Z$, and $Y$ refer to the input, representation and target of a predictive task, respectively.

## A.2 Assumptions

In this section, we discuss the assumptions that we make throughout our paper. Specifically, we discuss why we make those assumption and why such assumptions should hold in practice. **All our assumptions should hold in most practical scenarios.** The following assumptions will be implicit in the rest of our work.

**Assumption 1** (Finite risk)**.** We restrict ourselves to tasks $Y$, such that $|\mathrm{R}[Y \mid \eta]| < \infty$ for any finite constant $\eta$ in the domain of the predictor $f$. Similarly we restrict ourselves to $X$ with $\mathrm{R}[X \mid \eta] < \infty$, and to equivalences relations on $\mathcal{X}$ such if there exists a maximal invariant then there exists *some* maximal invariant $M(X)$ with $\mathrm{R}[M(X) \mid \eta] < \infty$ for any finite constant $\eta$.

Assumption 1 ensures that we can take differences of Bayes risks as in Definition 6. For the case of log loss our assumption is equivalent to requiring finite (differential or discrete) entropy of $\mathrm{H}[X]$, $\mathrm{H}[M(X)]$ and $\mathrm{H}[Y]$. For MSE loss, this is equivalent to finite variance for $X$, $Y$ and $M(X)$. Specifically, we will restrict ourselves to random variables $Y$ and $M(X)$ that are bounded in $L^2(\Omega, \mathcal{H}, \mathbb{P})$. (See Assumption 6 in Appx. B.6.) Note that $\mathrm{R}[M(X) \mid \eta] < \infty$ comes directly from $\mathrm{R}[X \mid \eta] < \infty$ for the two main losses that we consider. Indeed, for log loss this comes directly from

the data processing inequality. For MSE such $M(X)$ can easily be constructed by mapping any $x$ to a value in $[x]$ that is smaller than the expected value over the equivalence class $\mathrm{E}[X|X \in [x]]$.

**Assumption 2** (Existence of regular conditional probabilities). We restrict ourselves to standard Borel measurable spaces, so that the existence of regular conditional probability distributions is ensured. This is necessary to ensure the existence of probability kernel in Lemma 4. This technical assumption essentially holds for all practical purposes. Unless stated otherwise, we denote $\mathcal{B}(\mathcal{Y})$ the Borel $\sigma$-algebra of a set $\mathcal{Y}$.

**Assumption 3** (Measurability of functions). We assume that all functions introduced in the following sections are measurable with respect to the "natural" measurable spaces of the functions' domain and codomain. (A few special functions will be shown to be measurable.) In particular, we require (i) the measurability of $M(\cdot)$ which implies that $M(X)$ is a random variable; and (ii) the measurability of the projection $\pi_\sim : \mathcal{X} \to \mathcal{X}/\sim$, which implies that there always exists a maximal invariant in the form of the projection $\pi_\sim$. This technical assumption holds for essentially all practical purposes.

Assumptions 1 to 3 are used throughout our work. Two further assumptions are needed for log loss, which we remove in Appx. B.6 when we obtain results for MSE.

**Assumption 4** (Countably many equivalence classes). For the log loss risk (Appx. B.3) we restrict our discussion to equivalences such that the quotient set $\mathcal{X}/\sim$ is countable. This ensures that $M(X)$ is a discrete random variable thereby ensuring that our invariance distortion $\mathrm{R}[M(X)\,|\,Z]$ is independent of the choice of maximal invariant $M$ as the conditional entropy is invariant to bijections.

Note that Assumption 4 holds when $\mathcal{X}$ is countable which always happens in practice due to floating point arithmetic, i.e. every real number has to be rounded to the closest 64 bits number. Another perspective is to say that $\mathcal{X}$ is actually uncountable, but that all tasks we care about are always invariant to rounding to the nearest 64 bits number due to floating point arithmetic. As a result, the maximal invariant is the usual maximal invariant rounded to the closest floating point. For example, if $X$ is a 2D Gaussian we cannot work directly with translations on the y-axis (which gives uncountably many $[x]$, one for each real number on the x-axis), but can work with y-axis invariance combined with invariance to rounding on the x-axis (e.g. closest 64 bits number).

**Assumption 5** (Convenience assumption: Existence of densities). In sections Apps. B.2 and B.3, where we work with log loss, we restrict ourselves to cases where the (conditional) probability mass/density function exist, i.e., to probability distributions that are absolutely continuous w.r.t. to some (shared) underlying measure. This assumption is not needed but it simplifies the notation, and ensures that the differential entropy of random variables is well defined. Such assumption could be removed by using the general definition of mutual information as a supremum over partitions and by defining continuous entropy as $\mathrm{H}[X] = \mathrm{I}[X; X]$ [1, 2], also known as Jaynes's [3] limiting density of discrete points.

### A.3 Definitions

In the main paper we were relatively informal in our definitions, here we restate our main definitions more formally.

**Definition 1** (Maximal invariant). Let $\sim$ denote an equivalence relation on $\mathcal{X}$. We say that a measurable function $M : \mathcal{X} \to \mathcal{M}$ is a *maximal invariant* w.r.t. $(\mathcal{X}, \sim)$ if

$$\forall x, x' \in \mathcal{X} \quad x \sim x' \iff M(x) = M(x') \tag{1}$$

Note that our notion of maximal invariants generalizes the notion of maximal invariants in probabilistic group theory [4]. We refer the reader to Lehmann and Romano [5] for many examples in the group case. As in the group case, a maximal invariant typically is not unique.

The invariance structure that we want our tasks to have is based on their conditional distributions given $X$, defined as follows.

**Definition 2** (Conditional invariance). We say that $Y$ is conditionally invariant w.r.t. $(\mathcal{X}, \sim)$, if the regular conditional distribution $x \mapsto P(Y \mid x)$ is invariant w.r.t. $\sim$, i.e. $\forall x, x' \in \mathcal{X}$ we have

$$x \sim x' \implies P(Y \mid x) = P(Y \mid x') \tag{2}$$

**Definition 3** (Invariant tasks of interest). The set of all invariant tasks of interest $\mathcal{T}_\sim$ w.r.t. to a loss and an equivalence $(\mathcal{X}, \sim)$ is the set of all random variables $Y$ that are conditionally invariant w.r.t. $(\mathcal{X}, \sim)$ and that satisfy Assumption 1 (finite risk).

First we require the notion of a valid distortion [6], which ensures that we can apply the rate distortion theorem.

**Definition 4** (Valid distortion). Let $X$ and $Z$ be two random variables that take values in $\mathcal{X}$ and $\mathcal{Z}$, respectively. Then an (expected) distortion D is *valid* w.r.t. $X, Z$ if there exists a point-wise distortion $d : \mathcal{X} \times \mathcal{Z} \to \mathbb{R}_{\geq 0}$ such that $\mathrm{E}_{p(X,Z)}[d(X, z)] \leq \infty$ for some $z \in \mathcal{Z}$ and

$$\mathrm{D}[X, Z] := \mathrm{E}_{p(X,Z)}[d(X, Z)] \ . \tag{3}$$

In the context of the current work, a representation $Z$ which arises by encoding $X$ using $p(Z \mid X)$ should not depend on any particular task $Y$.

**Definition 5** (Representation for a task set). Let $X, Z$ be two random variables and $\mathcal{T}$ be a set of random variables. $Z$ is a *representation* of $X$ for $\mathcal{T}$ if for all $Y \in \mathcal{T}$ such that $Y$ and $Z$ are not almost surely equal, we have the pairwise conditional independence $Y \perp\!\!\!\perp Z \mid X$.

Note that if $Z \notin \mathcal{T}$ then it is not almost surely equal to any $Y \in \mathcal{T}$. The condition allows for the possibility that $Z \in \mathcal{T}$, but it must be conditionally independent, given $X$, of all other random variables in $\mathcal{T}$.

We now recall the excess risk distortion $\mathrm{D}_\mathcal{T}$.

**Definition 6** (Excess risk distortion). Let $X$ and $Z$ be two random variables. Let $\mathcal{T}$ be a set of random variables such that under a loss $L$, the Bayes risks in (4) below are well defined for each $Y \in \mathcal{T}$. The *excess risk distortion* $\mathrm{D}_\mathcal{T}$ is defined as:

$$\mathrm{D}_\mathcal{T}[X, Z] := \sup_{Y \in \mathcal{T}} \ \ \mathrm{R}[Y \mid Z] - \mathrm{R}[Y \mid X] \tag{4}$$

# B   Proofs: optimal bit-rate

In this section we prove all results from Sec. 3.

## B.1   Basic properties of equivalence relations and maximal invariants

To begin, we collect some basic properties of equivalence relations and maximal invariants. As these a general result that might be of interest beyond our work (especially Lemma 4) we will prove them without assuming the existence of densities, i.e., without Assumption 5. Recall that $\pi_\sim \colon \mathcal{X} \to \mathcal{X}/\sim$ is the projection from $\mathcal{X}$ onto its quotient by $\sim$, denoted $\mathcal{X}/\sim$.

**Lemma 1** (Mac Lane and Birkhoff [7], Theorem 19)**.** Given an equivalence relation $\sim$ on $\mathcal{X}$, let $f \colon \mathcal{X} \to \mathcal{S}$ be any function such that $x \sim x' \Rightarrow f(x) = f(x')$. Then there is exactly one function $g \colon \mathcal{X}/\sim \to \mathcal{S}$ for which $f = g \circ \pi_\sim$. If $f$ is a surjection and $f(x) = f(x') \Rightarrow x \sim x'$, then $g$ is a bijection.

**Lemma 2.** Let $M \colon \mathcal{X} \to \mathcal{M}$ and $M' \colon \mathcal{X} \to \mathcal{M}'$ be two maximal invariants w.r.t. $(\mathcal{X}, \sim)$. Then there exists a bijective function $f \colon \mathcal{M} \to \mathcal{M}'$ such that $M' = f \circ M$.

*Proof.* From Lemma 1, $M$ is a maximal invariant if and only if there is a bijective function $g \colon \mathcal{X}/\sim \to \mathcal{M}$ such that the maximal invariant is the composition of $g$ and the projection onto equivalence classes, i.e. $M = g \circ \pi_\sim$. Let $g'$ be the corresponding bijection for $M'$. Then we have $M' = f \circ M$ with $f := g' \circ g^{-1}$ which is indeed bijective: $f^{-1} := g \circ g'^{-1}$.                                           $\square$

**Lemma 3.** Let $M$ be any maximal invariant w.r.t. $(\mathcal{X}, \sim)$. Then a measurable function $f \colon \mathcal{X} \to \mathcal{S}$ is invariant with respect to $(\mathcal{X}, \sim)$ if and only if there exists a measurable function $h \colon \mathcal{M} \to \mathcal{S}$ such that $f(x) = (h \circ M)(x)$ for all $x \in \mathcal{X}$, in which case $f$ is measurable with respect to the $\sigma$-algebra generated by $M$.

*Proof.* Clearly $f = h \circ M$ is $(\mathcal{X}, \sim)$-invariant because $M$ is, and measurability of $f$ follows from measurability of $M$ and $h$.

From Lemma 1, if $f$ is $(\mathcal{X}, \sim)$-invariant then there is a function $s \colon \mathcal{X}/\sim \to \mathcal{X}$ such that $f = s \circ \pi_\sim$. Since $\pi_\sim$ and $f$ are measurable, so too is $s$. Again by Lemma 1, there exists a bijective mapping $g \colon \mathcal{X}/\sim \to \mathcal{M}$ such that $M = g \circ \pi_\sim$. We thus conclude that $f = h \circ M$, for $h := s \circ g^{-1}$. The measurability of $h$ follows from the measurability of $s$ and of $g^{-1}$.                      $\square$

Finally, we establish a key conditional independence relationship, which shows that for invariant tasks, $Y - M(X) - X$ forms a Markov Chain. This is a generalization of an probabilistic group theoretical results (Theorem 4.4 in Eaton [4], Theorem 7 in Bloem-Reddy and Teh [8]), to any equivalences (rather than only group orbits) and without making the assumption of (marginal) invariance of $P(X)$ to $\sim$.

**Lemma 4.** Let $X$ and $Y$ be two random variables, and $M \colon \mathcal{X} \to \mathcal{M}$ be a maximal invariant w.r.t. $(\mathcal{X}, \sim)$ as in Definition 1. Then $Y$ is conditionally invariant w.r.t. $(\mathcal{X}, \sim)$ as in Definition 2 if and only if $Y \perp\!\!\!\perp X \mid M(X)$.

*Proof.* Let $Y$ be a $\mathcal{Y}$-valued random variable that is conditionally invariant w.r.t. $(\mathcal{X}, \sim)$. Recall that $\mathcal{B}(\mathcal{Y})$ is the Borel $\sigma$-algebra of $\mathcal{Y}$. By Assumption 2, there exists a probability kernel $\kappa_Y(A, x)$ from $(\mathcal{X}, \mathcal{B}(\mathcal{X}))$ into $(\mathcal{Y}, \mathcal{B}(\mathcal{Y}))$, such that for each set $A \in \mathcal{B}(\mathcal{Y})$, $x \mapsto \kappa_Y(A, x)$ is a measurable function mapping $\mathcal{X} \to \mathbb{R}_{\geq 0}$.

Conditional invariance means that $x \sim x' \implies \kappa_Y(A, x) = \kappa_Y(A, x')$ for each $A \in \mathcal{B}(\mathcal{Y})$. That is, as a function of $x$, $\kappa_Y(A, \bullet)$ is invariant w.r.t. $(\mathcal{X}, \sim)$. By Lemma 3, $x \mapsto \kappa_Y(A, x) = \kappa'_Y(A, M(x))$, where $\kappa'_Y$ is a probability kernel from $(\mathcal{M}, \mathcal{B}(\mathcal{M}))$ into $(\mathcal{Y}, \mathcal{B}(\mathcal{Y}))$. Therefore, for any $A \in \mathcal{B}(\mathcal{Y}), B \in \mathcal{B}(\mathcal{X})$,
$$\mathbb{E}_{X,Y}[\mathbb{1}_B(X)\mathbb{1}_A(Y)] = \mathbb{E}_X[\mathbb{1}_B(X)\kappa_Y(A, X)] = \mathbb{E}_X[\mathbb{1}_B(X)\kappa'_Y(A, M(X))] ,$$
which can be extended to arbitrary measurable functions on $\mathcal{X} \times \mathcal{Y}$ by a standard (monotone class) argument. This in turn implies that $P_{Y|M(X)}$ is a version of $P_{Y|X}$, i.e., they are equal almost surely $\mathbb{P}(X)$, and therefore $Y \perp\!\!\!\perp X \mid M(X)$.                                   $\square$

Finally, we prove that in realistic settings, there exists at least one $M(X) \in \mathcal{T}_\sim$.

**Lemma 5.** Let $\mathcal{T}_\sim$ be the invariant tasks of interest w.r.t. $(\mathcal{X}, \sim)$ and any loss function. Then there exists at least one maximal invariant that belongs to $\mathcal{T}_\sim$.

*Proof.* First, we have to prove by construction that a maximal invariant always exists. By definition equivalent elements have the same equivalence class and so $x \sim x' \iff \pi_\sim(x) = \pi_\sim(x')$. The projection map is measurable by assumption (Assumption 3), so it is a maximal invariant.

Second, due to the existence of at least one maximal invariant $M = \pi_\sim$ we have by Assumption 1 that that there exists at least one $M$ s.t. $\mathrm{R}[M(X) \,|\, \eta]$. This $M$ is therefore in $\mathcal{T}_\sim$.

$\square$

We close this section by establishing some properties of Bayes risk in this context. The following lemma, a data-processing inequality, appears in Xu and Raginsky [9]; we include it here for completeness, and provide a slightly more detailed proof.

**Lemma 6** (Data-processing inequality for Bayes risk). Let $Z - X - Y$ be a Markov chain of random variables. For any loss function $L$,
$$\mathrm{R}[Y \,|\, X] \le \mathrm{R}[Y \,|\, Z] \ . \tag{5}$$

*Proof.* Recall that one characterization of conditional independence is that $Y \perp\!\!\!\perp Z \,|\, X$ if and only if $Z = f(X, U)$ almost surely for some measurable function $f$ and $U \sim \mathrm{Unif}(0,1)$ with $U \perp\!\!\!\perp (X, Y)$ [10, Prop. 6.13].

Let $\psi_z$ be a Bayes decision rule for predicting $Y$ from $Z$, and likewise for $\psi_x$. By definition,
$$\mathrm{R}[Y \,|\, Z] = \mathbb{E}_{Z,Y}[L(Y, \psi_z(Z))] = \mathbb{E}_{X,Y,U}[L(Y, \psi_z(f(X, U)))] \ .$$
For any $u \in (0,1)$, $\psi_z(f(\bullet, u))$ is a valid decision rule for predicting $Y$ from $X$ with risk at least as great as $\psi_x$. Therefore, $\mathrm{R}[Y \,|\, X] \le \mathrm{R}[Y \,|\, Z]$.

$\square$

**Corollary 7.** Let $\mathcal{T}_\sim$ be the invariant tasks of interest with respect to $(\mathcal{X}, \sim)$ and any loss function, and $M$ any maximal invariant. For any $Y \in \mathcal{T}_\sim$,
$$\mathrm{R}[Y \,|\, M(X)] = \mathrm{R}[Y \,|\, X] \ . \tag{6}$$

*Proof.* The result follows from applying Lemma 6 to the trivial conditional independence $Y \perp\!\!\!\perp M(X) \,|\, X$ and the non-trivial conditional independence from Lemma 4 $Y \perp\!\!\!\perp X \,|\, M(X)$. $\square$

### B.2 Proposition 1: simplifying and validating $\mathrm{D}_\mathcal{T}$ for log loss

In this section we show that Definition 6 is a valid distortion for log loss, and we prove the equivalence between Definition 6 and $\mathrm{R}_{\log}[M(X) \,|\, Z]$. That equivalence is the key to prove Theorem 2.

The main steps in the proof are the following:

1. Using the strict properness of the log loss, we relate the Bayes risk to the entropy:
$$\mathrm{R}_{\log}[Y \,|\, Z] = \mathrm{H}[Y \,|\, Z] \tag{7}$$

2. We show that the supremum is achieved by $M(X)$:
$$\sup_{Y \in \mathcal{T}_\sim} \mathrm{R}_{\log}[Y \,|\, Z] - \mathrm{R}_{\log}[Y \,|\, X] = \mathrm{H}[M(X) \,|\, Z] - \mathrm{H}[M(X) \,|\, X] \tag{8}$$

3. Since $M$ is a deterministic function and $M(X)$ is discrete, we have $\mathrm{H}[M(X) \,|\, X] = 0$. Therefore,
$$\sup_{Y \in \mathcal{T}_\sim} \mathrm{R}_{\log}[Y \,|\, Z] - \mathrm{R}_{\log}[Y \,|\, X] = \mathrm{H}[M(X) \,|\, Z] \tag{9}$$

4. We conclude, as desired, that
$$\sup_{Y \in \mathcal{T}_\sim} \mathrm{R}_{\log}[Y \,|\, Z] - \mathrm{R}_{\log}[Y \,|\, X] = \mathrm{R}_{\log}[M(X) \,|\, Z] \tag{10}$$

The first step consists of relating the log loss Bayes risk and conditional entropy. This is a simple lemma that directly comes from the fact that the conditional distribution $p(Y \mid Z)$ is the Bayes predictor.

**Lemma 8.** Let $Y, X$ be random variables then the log loss Bayes risk is equal to the conditional (discrete or differential) entropy:

$$\mathrm{R}_{\log}[Y \mid X] = \mathrm{H}[Y \mid X] \tag{11}$$

*Proof.*

$$
\begin{aligned}
\mathrm{R}_{\log}[Y \mid X] &= \inf_{q(Y \mid X)} \mathrm{E}_{P(X,Y)}[-\log q(Y|X)] && \text{Definition} && (12) \\
&= \mathrm{E}_{P(X,Y)}[-\log p(Y|X)] && \text{Strict Proper.} && (13) \\
&= \mathrm{H}[Y \mid X] && \text{Definition} && (14)
\end{aligned}
$$

Where Eq. (13) uses the strict properness of the logarithmic scoring function rule [11]. □

In the rest of the section, we will often be working with $\mathrm{H}[M(X)]$ and $\mathrm{H}[M(X) \mid Z]$. Importantly, we would like our results to be independent of the choice of maximal invariant $M$. We now prove that this will indeed be the case as all these (conditional) entropy terms are independent of the choice of $M$. We only prove it for the marginal entropy $\mathrm{H}[M(X)]$ but the same proof holds for conditional entropies.

**Lemma 9.** Let $\sim$ denote an equivalence relation on $\mathcal{X}$ satisfying Assumption 4. Let $M$ and $M'$ be two different maximal invariants w.r.t. $(\mathcal{X}, \sim)$. Then $\mathrm{H}[M(X)] = \mathrm{H}[M'(X)]$.

*Proof.* Due to Assumption 4, $M(X)$ is a discrete random variable and so $\mathrm{H}[M(X)]$ is the discrete entropy, which is invariant to bijective functions [12]. From Lemma 2 we know that there exists a bijection between $M$ and $M'$ from which we conclude that $\mathrm{H}[M(X)] = \mathrm{H}[M'(X)]$ as desired. □

One of the requirements on $Y$ to be set of downstream tasks $\mathcal{T}$ is the finiteness of $\mathrm{H}[Y]$. Thus, as a consequence of Lemmas 5 and 9, in the case of log loss, all $M(X)$ are always in the set of downstream tasks $\mathcal{T}$.

**Lemma 10.** Let $\mathcal{T}_\sim$ be the invariant tasks of interest w.r.t. $(\mathcal{X}, \sim)$ and the log loss. Then all maximal invariants are in $\mathcal{T}_\sim$.

*Proof.* Any $M(X)$ is conditionally invariant due to the Definition 1. From Assumption 1 we know that there exists at least one $M(X)$ with finite entropy, by Lemma 9 they must all have finite entropy. We conclude that all $M(X) \in \mathcal{T}_\sim$ (Definition 3). □

We are now ready to prove the desired proposition.

**Proposition 1** (Invariant Distortion for log loss)**.** Let $\mathcal{T}_\sim$ be the invariant tasks of interest w.r.t. $(\mathcal{X}, \sim)$ and the log loss. Let $M$ be any maximal invariant, and $Z$ be a representation of $X$ for $\mathcal{T}_\sim$. Then the excess distortion w.r.t. log loss, $\mathrm{D}_{\mathcal{T}_\sim}$, is a valid distortion and

$$\mathrm{D}_{\mathcal{T}_\sim}[X, Z] = \mathrm{R}_{\log}[M(X) \mid Z] \tag{15}$$

*Proof.* We first prove that $\mathrm{D}_{\mathcal{T}_\sim}[X, Z] = \mathrm{H}[M(X) \mid Z]$, from which it is straightforward to show that $\mathrm{D}_{\mathcal{T}_\sim}$ is a valid distortion. Starting from the definition of $\mathrm{D}_{\mathcal{T}_\sim}$, we have

$$
\begin{aligned}
\mathrm{D}_{\mathcal{T}_\sim}[X, Z] &:= \sup_{Y \in \mathcal{T}_\sim} \mathrm{R}_{\log}[Y \mid Z] - \mathrm{R}_{\log}[Y \mid X] && \text{Definition 6} && (16) \\
&= \sup_{Y \in \mathcal{T}_\sim} \mathrm{H}[Y \mid Z] - \mathrm{H}[Y \mid X] && \text{Lemma 8} && (17) \\
&= \sup_{Y \in \mathcal{T}_\sim} \mathrm{H}[Y \mid Z] - \mathrm{H}[Y \mid X, M(X), Z] && Y \perp\!\!\!\perp (M(X), Z)|X && (18) \\
&= \sup_{Y \in \mathcal{T}_\sim} \mathrm{H}[Y \mid Z] - \mathrm{H}[Y \mid M(X), Z] && Y \perp\!\!\!\perp X|(M(X), Z) && (19)
\end{aligned}
$$

$$\begin{aligned}
&= \sup_{Y \in \mathcal{T}_\sim} \mathrm{I}[Y; M(X)|Z] && \text{Def.} && (20)\\
&= \sup_{Y \in \mathcal{T}_\sim} \mathrm{H}[M(X)\,|\,Z] - \mathrm{H}[M(X)\,|\,Y,Z] && \text{Symmetry and def.} && (21)\\
&= \mathrm{H}[M(X)\,|\,Z] - \inf_{Y \in \mathcal{T}_\sim} \mathrm{H}[M(X)\,|\,Y,Z] && && (22)\\
&= \mathrm{H}[M(X)\,|\,Z] - 0 && \text{Discrete H and } Y = M(X) && (23)\\
&= \mathrm{R}_{\log}[M(X)\,|\,Z] && \text{Lemma 8} && (24)
\end{aligned}$$

Eq. (18) uses the fact that $Y \perp\!\!\!\perp M(X)\,|\,X$ (Lemma 4), that $Y \perp\!\!\!\perp Z\,|\,X$ by Definition 5, and that $M(X) \perp\!\!\!\perp Z\,|\,X$ because $M(X) \in \mathcal{T}_\sim$ (again using Definition 5). Eq. (19) uses Lemma 4. To go from Eq. (19) to Eq. (21) we use the symmetry of conditional mutual information. Eq. (23) uses the discreteness of $M(X)$ due to Assumption 4, so $\mathrm{H}[M(X)\,|\,Y,Z] \geq 0$ with equality when $Y = M(X)$ which is possible due to Lemmas 5 and 10.

From Eq. (23) it is easy to see that $\mathrm{D}_{\mathcal{T}_\sim}$ is valid as $\mathrm{D}_{\mathcal{T}_\sim}[X, Z] = \mathrm{H}[M(X)\,|\,Z] = \mathrm{E}_{p(X,Z)}[d(X, Z)]$ with $d(x, z) := -\log p(M(x)\,|\,z)$ which due to the discreteness of $M(X)$ (Assumption 4) is a function whose codomain is $\mathbb{R}_{\geq 0}$ as desired. Due to Assumption 1 we know that for all constant $z \in \mathcal{Z}$ we have $\mathrm{H}[M(X)\,|\,z] \leq \infty$, so $\mathrm{D}_{\mathcal{T}_\sim}$ is valid. $\qquad\square$

Note that $\mathrm{D}_{\mathcal{T}_\sim}[X, Z] = \mathrm{R}_{\log}[M(X)\,|\,Z]$ is very simple to work with if we have access to some $M(X)$. Unfortunately, in practice $M(X)$ might not be known, but often we will have access to some other random variable $\tilde{X}$ which has all the information necessary about $M(X)$. See for example Appx. C.3. We now prove that in such case we can optimize $\mathrm{R}_{\log}\!\left[\tilde{X}\,|\,Z\right]$ instead of $\mathrm{R}_{\log}[M(X)\,|\,Z]$.

**Proposition 11** (Invariant Distortion without $M(X)$). *Let* $\mathcal{T}_\sim, \sim, M, X, Z, \mathrm{D}_{\mathcal{T}_\sim}[X, Z]$ *be as in Prop. 1. Let* $\tilde{X}$ *be a random variable such that* $\tilde{X} \sim X$ *and* $\tilde{X} \perp\!\!\!\perp X\,|\,M(X)$ *almost surely. Then*

$$\mathrm{D}_{\mathcal{T}_\sim}[X, Z] = \mathrm{R}_{\log}\!\left[\tilde{X}\,|\,Z\right] + c \tag{25}$$

*where* $c$ *depends only on* $\tilde{X}$ *and not on* $Z$.

*Proof.* From Definition 5 and the fact that $M(X) \in \mathcal{T}_\sim$ (Lemma 10) we know that $Z - X - M(X)$ forms a Markov Chain (MC). Due to our assumption $\tilde{X} \sim X$ a.s. we also have that $M(\tilde{X}) = M(X)$ a.s. so $M(X) - \tilde{X} - M(X)$ forms a MC. Putting all together we obtain the MC $Z - X - M(X) - \tilde{X} - M(X)$, which allows us to derive the following.

$$\begin{aligned}
\mathrm{D}_{\mathcal{T}_\sim}[X, Z] &= \mathrm{R}_{\log}[M(X)\,|\,Z] && \text{Prop. 1} && (26)\\
&= \mathrm{H}[M(X)] - \mathrm{I}[M(X); Z] && \text{Lemma 8} && (27)\\
&= \mathrm{H}[M(X)] - \mathrm{I}\!\left[(M(X), \tilde{X}); Z\right] && Z - M(X) - \tilde{X} && (28)\\
&= \mathrm{H}[M(X)] - \mathrm{I}\!\left[\tilde{X}; Z\right] && Z - \tilde{X} - M(X) && (29)\\
&= \mathrm{H}[M(X)] - \mathrm{H}\!\left[\tilde{X}\right] + \mathrm{H}\!\left[\tilde{X}\,|\,Z\right] && && (30)\\
&= \mathrm{H}\!\left[\tilde{X}\,|\,M(X)\right] + \mathrm{R}_{\log}\!\left[\tilde{X}\,|\,Z\right] && \text{Lemma 8} && (31)
\end{aligned}$$

where the last line uses the Markov Chain $M(X) - \tilde{X} - M(X)$ to provide a more interpretable value for the constant. $\qquad\square$

### B.3 Theorem 2: optimal bit-rate under log loss

Our main theoretical result is to characterize the minimal achievable rate to bound the Bayes risk of any invariant task. Here we provide the proof for the case of log loss risk. The result follows from Shannon's [13] rate distortion theorem, and the validity of $\mathrm{D}_{\mathcal{T}_\sim}$ (Proposition 1).

For convenience, we restate the well known rate distortion theorem.

**Lemma 12.** (Shannon [13]; Theorem 7.2.4 and 7.2.5 from Berger [14]) Let $\mathrm{D}[X;Z]$ be a valid distortion. The minimum achievable bit-rate for transmitting an i.i.d. source $X$ with expected distortion less than $\delta \geq 0$ is given by the rate-distortion function:

$$R(\delta) = \min_{p(Z|X) \text{ such that } \mathrm{D}[X;Z] \leq \delta} \mathrm{I}[X;Z] \tag{32}$$

We can now state our rate-invariance theorem.

**Theorem 2** (Rate-invariance for log loss)**.** Let $\delta \geq 0$. Let $\sim$ be an equivalence relation on $\mathcal{X}$ that partitions $\mathcal{X}$ into countably many equivalence classes (Assumption 4). Let $\mathcal{T}_\sim$ be the invariant tasks of interest w.r.t. $(\mathcal{X},\sim)$ and the log loss, $M$ be any maximal invariant, and $Z$ be a representation of $X$ for $\mathcal{T}_\sim$. Let $Rate(\delta)$ denote the minimum achievable bit-rate for transmitting an i.i.d. source of $Z$ such that for any $Y \in \mathcal{T}_\sim$ we have $\mathrm{R}_{\log}[Y \mid Z] \leq \delta + \mathrm{R}_{\log}[Y \mid X]$. Then $Rate(\delta)$ is finite and given by

$$Rate(\delta) = \max(0, \ \mathrm{H}[M(X)] - \delta) \tag{33}$$
$$= \max(0, \ \mathrm{H}[X] - \mathrm{H}[X \mid M(X)] - \delta) \tag{34}$$

*Proof.* We first prove that $Rate(\delta) \geq \max(0, \ \mathrm{H}[M(X)] - \delta)$. We then prove that the rate $\max(0, \ \mathrm{H}[M(X)] - \delta)$ is achievable and so Eq. (33) holds. Finally, we prove that $\mathrm{H}[M(X)] = \mathrm{H}[X] - \mathrm{H}[X \mid M(X)]$ so Eq. (34) holds which concludes the proof.

We want to transmit $Z$ such that $\forall Y \in \mathcal{T}_\sim$ we have $\mathrm{R}_{\log}[Y \mid Z] \leq \delta + \mathrm{R}_{\log}[Y \mid X]$, in other words we would like $\sup_{Y \in \mathcal{T}} \mathrm{R}_{\log}[Y \mid Z] - \mathrm{R}_{\log}[Y \mid X] =: \mathrm{D}_{\mathcal{T}_\sim}[X,Z] \leq \delta$. As $\mathrm{D}_{\mathcal{T}_\sim}$ is valid (Proposition 1) we can directly apply the rate distortion theorem (Lemma 12):

$$Rate(\delta) = \min_{p(Z|X) \text{ s.t. } \mathrm{D}_{\mathcal{T}_\sim}[X,Z] \leq \delta} \mathrm{I}[X;Z] \qquad \text{Lemma 12 and Proposition 1} \tag{35}$$

$$\geq \min_{p(Z|X) \text{ s.t. } \mathrm{D}_{\mathcal{T}_\sim}[X,Z] \leq \delta} \mathrm{I}[M(X);Z] \qquad \text{DPI} \tag{36}$$

$$= \min_{p(Z|X) \text{ s.t. } \mathrm{D}_{\mathcal{T}_\sim}[X,Z] \leq \delta} \mathrm{H}[M(X)] - \mathrm{H}[M(X) \mid Z] \tag{37}$$

$$= \min_{p(Z|X) \text{ s.t. } \mathrm{D}_{\mathcal{T}_\sim}[X,Z] \leq \delta} \mathrm{H}[M(X)] - \mathrm{D}_{\mathcal{T}_\sim}[X,Z] \qquad \text{Proposition 1 and Lemma 8} \tag{38}$$

$$\geq \min_{p(Z|X) \text{ s.t. } \mathrm{D}_{\mathcal{T}_\sim}[X,Z] \leq \delta} \mathrm{H}[M(X)] - \delta \tag{39}$$

$$= \mathrm{H}[M(X)] - \delta \qquad \text{No } Z \tag{40}$$

Where Eq. (36) uses the data processing inequality (DPI). As the rate is always non-negative we have $Rate(\delta) \geq \max(0, \mathrm{H}[M(X)] - \delta)$.

We now prove that $\max(0, \mathrm{H}[M(X)] - \delta)$ is attainable and so $Rate(\delta) = \max(0, \mathrm{H}[M(X)] - \delta)$. Specifically we need to find a representation $Z$ of $X$ such that

$$Rate(\delta) = \begin{cases} 0 & \text{If } \delta \geq \mathrm{H}[M(X)] \\ \mathrm{H}[M(X)] - \delta & \text{Else} \end{cases} \tag{41}$$

The first case is trivial: set $Z$ to be independent of $M(X)$ and $X$, e.g. a constant. Then, $\mathrm{D}_{\mathcal{T}_\sim}[X,Z] = \mathrm{H}[M(X) \mid Z] = \mathrm{H}[M(X)] \leq \delta$ and $Rate(\delta) = \mathrm{I}[Z;X] = 0$.

For the second case we need $Rate(\delta) \geq \mathrm{H}[M(X)] - \delta$ to be an equality when $\delta < \mathrm{H}[M(X)]$. This happens iff inequalities Eq. (36) and Eq. (39) are equalities, i.e. iff $X \perp\!\!\!\perp Z \mid M(X)$ (for equality of the DPI [15]) and $\mathrm{D}_{\mathcal{T}_\sim}[X,Z] = \delta$. We do so by starting from $Z = M(X)$ (such that $X \perp\!\!\!\perp Z \mid M(X)$) and "erasing" a fraction $\alpha$ of bits, similarly to binary erasure channels, until $\mathrm{D}_{\mathcal{T}_\sim}[X,Z] = \delta$. Let $\mathcal{Z} := \mathcal{M} \cup \{\epsilon\}$ for some $\epsilon \notin \mathcal{M}$ and let $Z$ be a random variable that t.v.i. in $\mathcal{Z}$ and have the following conditional density parametrized by $\alpha \in [0,1[$:

$$\forall z \in \mathcal{Z}, \forall m \in \mathcal{M}, \quad p(z \mid m) = \begin{cases} 1 - \alpha & \text{if } z = m \\ \alpha & \text{if } z = \epsilon \\ 0 & \text{else} \end{cases} \tag{42}$$

A simple computation then gives $\mathrm{D}_{\mathcal{T}_\sim}[X,Z] := \mathrm{R}_{\log}[M(X) \mid Z] = \mathrm{H}[M(X) \mid Z] = (1 - \alpha)\,\mathrm{H}[M(X) \mid Z = M(X)] + \alpha\,\mathrm{H}[M(X) \mid Z = \epsilon] = 0 + \alpha\,\mathrm{H}[M(X)]$, where the first equality uses

Lemma 8 and the last equality uses $\mathrm{H}[M(X) \mid M(X)] = 0$ due to the discreteness of $M(X)$ (Assumption 4). We can thus achieve $\mathrm{D}_{\mathcal{T}_\sim}[X, Z] = \delta$ by setting $\alpha = \frac{\delta}{\mathrm{H}[M(X)]}$. Note that we will never divide by zero as $\mathrm{H}[M(X)] = 0$ would be in the first case of Eq. (41). Importantly this $Z$ still satisfies $X \perp\!\!\!\perp Z \mid M(X)$ as it was constructed solely using $M(X)$ and independent noise.

We thus proved that $\max(0, \ \mathrm{H}[M(X)] - \delta)$ is obtainable and that $Rate(\delta) \geq \max(0, \ \mathrm{H}[M(X)] - \delta)$. From which we conclude that the best achievable bit-rate is $Rate(\delta) = \max(0, \ \mathrm{H}[M(X)] - \delta)$. Eq. (34), follows from $\mathrm{H}[M(X)] = \mathrm{I}[M(X); X] = \mathrm{H}[X] - \mathrm{H}[X \mid M(X)]$, which is a valid decomposition as both (differential conditional) entropy term are finite due to Assumption 1. The finiteness of $Rate(\delta)$ comes from the fact that $Rate(\delta) \leq \mathrm{H}[M(X)] < \infty$ due to Assumption 1. $\qquad\square$

By setting $\delta = 0$ we directly get the best achievable rate for the lossless prediction but lossy compression setting.

**Corollary 13** (Invariant source coding for log loss). Let $X, \sim, \mathcal{T}_\sim, M, Z$ be as in Theorem 2. Let $Rate(0)$ denote the minimum achievable bit-rate for transmitting an i.i.d. source of $Z$ such that for any $Y \in \mathcal{T}_\sim$ we have $\mathrm{R}_{\log}[Y \mid Z] = \mathrm{R}_{\log}[Y \mid X]$. Then $Rate(0)$ is finite and given by

$$Rate(0) = \mathrm{H}[M(X)] \tag{43}$$
$$= \mathrm{H}[X] - \mathrm{H}[X \mid M(X)] \tag{44}$$

## B.4 Recovering previous results in the literature

Corollary 13 recovers many previous results in the literature:

**Unlabeled Graphs** Let us consider the task of compressing unlabeled graphs, here we consider tasks that are invariant to graph isomorphisms. A possible maximal invariant is the graph canonization and $\mathrm{H}[M(X)]$ becomes the well known *structural entropy* (also called topological information content) [16, 17]. If all isomorphic graphs are permissible and equiprobable, Yongwook Choi and Szpankowski [17] show that the structural entropy is $\mathrm{H}[S] = \mathrm{H}[X] - \mathrm{E}_{x \sim p(X)}\left[\log \frac{n!}{|\mathrm{Aut}_{\mathcal{G}}(x)|}\right]$. This is Eq. (44), where the second term corresponds to $\mathrm{H}[X \mid M(X)]$ with a uniform distribution on isomorphic graphs.

**Multisets** Let us derive the best achievable bit-rate for compressing multisets. Let $X$ be any sequence and $\mathcal{T}_\sim$ be invariant to permutations of that sequence. One possible maximal invariant in that case is the empirical measure (also called type), i.e., the counts $K_1, \ldots, K_n$ of each of the $n$ elements that are present in the sequence $X$. Lossless compression of multisets thus requires $\mathrm{H}[M(X)] = \mathrm{H}[K_1, \ldots, K_n]$, as discussed by Varshney and Goyal [18]. Using Eq. (44) we can also characterize the bits gains that you obtain by considering the invariance, namely, $\mathrm{H}[X|M(X)]$. This recovers theorem 1 of Varshney and Goyal [18], where $\mathrm{H}[X|M(X)]$ is called the "order entropy". Note that similarly to our example in the main text about i.i.d. coin flips, the amount of bits needed to losslessly compress the multiset grows as $\Theta(\log n)$ [18].

**Information Bottleneck (IB)** Suppose you are interested in predicting a single task $Y = t(X)$, where $t$ is a (deterministic) "target function". The task is invariant to any transformations between examples in the preimage of the labeling. So the maximal invariant is $t(\cdot)$ and the distortion becomes $\mathrm{H}[t(X) \mid Z] = \mathrm{H}[Y \mid Z]$. Then the rate-distortion function (Eq. (32)) becomes the information bottleneck (IB) [19]. Using Corollary 13 we see that for lossless predictions the optimal rate is $Rate(0) = \mathrm{H}[Y] = \mathrm{H}[X] - \mathrm{H}[X \mid Y] = \mathrm{I}[X; Y]$ as shown in [20, 21]. From a compression stand point this is nevertheless not very useful as $Rate(0) = \mathrm{H}[Y]$, so IB for deterministic labels tells you to entropy code the labels $Y$.

**Lossless** Let $X$ be discrete. Every task will always be invariant to the equality "=" equivalent relation. In this case the maximal invariant is the identity function, and we recover Shannon's source coding theorem $Rate(0) = \mathrm{H}[M(X)] = \mathrm{H}[X]$.

## B.5 Generalizing Theorem 2: optimal bit rate for lossless prediction and any loss

Corollary 13 characterizes the minimal achievable bit rate for the lossless prediciton regime w.r.t. log loss. Here we show that the same result generalizes to essentially all loss function of practical interest.

The invariant source coding theorem does not hold for *any* loss, for example if a loss is a constant function then the Bayes risk $R_L[Y \mid Z]$ will not depend on the input $Z$, and so the best achievable bit rate will trivially be 0 which is different than $H[M(X)]$. But it essentially holds for all losses that are minimized only by the "correct" predictor. Specifically it holds for all losses that we dub information preserving.

**Definition 7** (Information preserving losses). Let $L : \mathcal{Y} \times \mathcal{A} \to \mathbb{R}_{\geq 0}$ be any loss function such that the Bayes risk $R_L[Y \mid Z]$ is well defined for all random variable $Z$. We say that $L$ is an *information preserving loss* iff the optimal risk of deterministic targets is achieved only using inputs that have all the information about the output, i.e., iff for any function $t : \mathcal{X} \to \mathcal{Y}$ and and r.v.s $Z, X$ we have

$$R_L[t(X) \mid X] = R_L[t(X) \mid Z] \implies \exists h : \mathcal{Z} \to \mathcal{Y} \text{ s.t. } t(X) \overset{\text{a.s.}}{=} h(Z) \tag{45}$$

In particular we have that if $t(X)$ is a discrete r.v. then $R_L[t(X) \mid X] = R_L[t(X) \mid Z] \implies H[t(X) \mid Z] = 0$.

Essentially all losses used in practice satisfy Definition 7. For example it holds for the following very general families of losses:

**Strictly proper scoring rules** Let $L : \mathcal{Y} \times \mathcal{P}(\mathcal{Y}) \to \mathbb{R}_{\geq 0}$ be a scoring rule that essentially quantifies with $L(y, q(Y), )$ the price/loss incurred by probabilistic prediction $q(Y)$ when $y$ is observed (lower is better). $L$ is *strictly proper* [11] (w.r.t. $\mathcal{P}(\mathcal{Y})$) iff:

$$\forall p, q \in \mathcal{P}(\mathcal{Y}) \quad E_{p(Y)}[L(y, q(Y))] \leq E_{p(Y)}[L(y, p(Y))] \tag{46}$$

with equality if and only if $p = q$. Common examples are the log loss [22], Brier score [23], spherical score [24], or the maximum mean discrepancy with characteristic bounded kernels [25, 26].

**Point-wise loss functions** Let $L : \mathcal{Y} \times \hat{\mathcal{Y}} \to \mathbb{R}_{\geq 0}$ be a loss function that essentially quantifies with $L(y, \hat{y})$ the price/loss incurred by the point prediction $\hat{y}$ when $y$ is observed (lower is better). As is standard [27] we assume that :

$$L(y, \hat{y}) = 0 \iff \hat{y} = y \tag{47}$$

This holds for most pointwise losses of interest: mean squared error, mean absolute error, 0-1 loss (accuracy), Huber loss, . . .

**Lemma 14.** Strictly proper scoring rules and point-wise loss functions are information preserving (Definition 7).

*Proof.* First let us consider point-wise loss functions. Suppose that $Z$ is such that $R_L[t(X) \mid Z] = R_L[t(X) \mid X]$. As $L$ is a point-wise loss function (Eq. (47)) and $t$ is a deterministic function, we have that $R_L[t(X) \mid X] = 0$. As a result we have $R_L[t(X) \mid Z] = 0$ which using again Eq. (47) is equivalent to the existence of $h$ s.t. $h(Z) \overset{\text{a.s.}}{=} t(X)$ as desired.

Now let us consider the case of proper scoring rules. Suppose that $Z$ is such that $R_L[t(X) \mid Z] = R_L[t(X) \mid X]$. As $L$ is a strictly proper scoring rule we have that $p(t(X)|Z) \overset{\text{a.s.}}{=} p(t(X)|X)$. The latter is a delta function, so the former must also. We thus have the existence of $h$ s.t. $h(Z) \overset{\text{a.s.}}{=} t(X)$ as desired. $\qquad \square$

We now have all the tools to prove the general invariant source coding theorem.

**Theorem 15** (General invariant source coding). Let $\sim$ be an invariance relation on $\mathcal{X}$ that satisfies Assumption 4. Let $\mathcal{T}_\sim$ be the invariant tasks of interest w.r.t $(\mathcal{X}, \sim)$ and any loss function $L : \mathcal{Y} \times \mathcal{A} \to \mathbb{R}_{\geq 0}$ as in Definition 3, $M$ be any $(\mathcal{X}, \sim)$ maximal invariant as in Definition 1, and $Z$ be a representation of $X$ for $\mathcal{T}_\sim$ as in Definition 5. Let $Rate(0)$ denote the minimum achievable bit-rate for transmitting an i.i.d. source of $Z$ such that for any $Y \in \mathcal{T}_\sim$ we have $R_L[Y \mid Z] = R_L[Y \mid X]$. Then $Rate(0)$ is finite and given by

$$Rate(0) = H[M(X)] \tag{48}$$
$$= H[X] - H[X \mid M(X)] \tag{49}$$

*Proof.* By Corollary 7 we know that for all $Y \in \mathcal{T}_\sim$ we have $\mathrm{R}_\mathrm{L}[Y \mid X] = \mathrm{R}_\mathrm{L}[Y \mid M(X)]$ for any loss function. As a result, the lossless prediction bit rate is at most $Rate(0) \le \mathrm{H}[M(X)]$ because by Shannon's [28] source coding theorem $M(X)$ can be transmitted using $\mathrm{H}[M(X)]$ bits as it is discrete (Assumption 4) and its entropy is finite (Assumption 1).

Let us now show that it is not possible to achieve a lower rate. By Lemma 5 there exists at least one maximal invariant such that $M(X) \in \mathcal{T}_\sim$. We now prove that there is no $Z$ such that $\mathrm{R}_\mathrm{L}[M(X) \mid Z] = \mathrm{R}_\mathrm{L}[M(X) \mid X]$ and can be transmitted with less than $\mathrm{H}[M(X)]$ bits. Suppose that $\mathrm{R}_\mathrm{L}[M(X) \mid Z] = \mathrm{R}_\mathrm{L}[M(X) \mid X]$, then because $L$ is a meaningful loss function we have there exists function $h$ s.t. $h(Z) = M(X)$. Using the discreteness of $M(X)$ (Assumption 4), we thus have $\mathrm{H}[M(X) \mid Z] = 0$. Using the RD theorem (Lemma 12) we know that the minimum bit rate for transmitting $Z$ under the constraint $\mathrm{H}[M(X) \mid Z] = 0$ is $\mathrm{I}[Z; M(X)] = \mathrm{H}[M(X)] - \mathrm{H}[M(X) \mid Z] = \mathrm{H}[M(X)] - 0$. We thus find that transmitting a $Z$ which ensures lossless predictions cannot require less than $\mathrm{H}[M(X)]$ bits which concludes the proof that $Rate(0) = \mathrm{H}[M(X)]$. To get Eq. (49) we use the same decomposition as in Theorem 2. $\qquad\square$

### B.6 Generalizing Theorem 2: optimal bit rate under MSE loss

In Appx. B.3 we proved the rate-invariance theorem for the case of log loss. Log loss is the standard loss function for classification in ML, but in the case of regresssion it is more common to use the MSE loss function. In Theorem 15 we have seen that our results for lossless prediction regime also holds for MSE (and other losses). Due to the importance of MSE in ML, we also provide a full rate-invariance theorem for MSE.

We assume that $Y \in \mathbb{R}^k$ for all $Y \in \mathcal{T}_\sim$, with some $k < \infty$. Tasks taking values in fewer dimensions can always be padded with zeros. In this section $||\bullet||$ denotes the Euclidean norm, the $L^2$-norm of a random variable $X$ is $\sqrt{\mathbb{E}[||X||^2]}$, and $L^2(\Omega, \mathcal{H}, \mathbb{P})$ is the Hilbert space of all $\mathbb{R}^k$-valued random variables with finite $L^2$-norm (random variables that are almost surely equal are identified as the same element of $L^2$). Since $\Omega$ and $\mathbb{P}$ remain unchanged but we may consider different $\sigma$-algebras, we use, e.g., $L^2(\mathcal{H})$ for short.

Importantly, we do not require Assumption 4 (countable $\mathcal{X}/\sim$). Instead, we require the following common (known as a finite power constraint in compression [15]) regularity condition on $\mathcal{T}_\sim$ to ensure that we can attain a relevant supremum.

**Assumption 6** ($L^2$-boundedness). We assume that $\mathcal{T}_\sim$ is bounded in $L^2$. That is, there is some $0 < B^2 < \infty$ such that $\mathbb{E}[||Y||^2] \le B^2$ for all $Y \in \mathcal{T}_\sim$.

Note that this is a slightly more stringent version of Assumption 1, as it essentially requires bounded variance of $Y$ rather than only finite variance.

Let $\mathcal{B}(\mathcal{M})$ denote the $\sigma$-algebra generated by a maximal invariant (all maximal invariants generate the same $\sigma$-algebra because they are one-to-one functions of each other Lemma 2), and let $\mathcal{M}_b$ denote all $\mathbb{R}^k$-valued functions that are $\mathcal{B}(\mathcal{M})$-measurable and bounded in $L^2$. Then $\mathcal{M}_b \subset \mathcal{T}_\sim$, and by Lemma 5, $\mathcal{M}_b$ is non-empty.

The main step in the proof of a rate-invariance theorem for MSE is the following proposition, which shows that the excess risk distortion for MSE is a valid distortion that can be expressed in terms of a maximum over $\mathcal{M}_b$.

**Proposition 16** (Invariant Distortion for MSE). Let $\mathcal{T}_\sim$ be the invariant tasks of interest w.r.t. $(\mathcal{X}, \sim)$ and w.r.t. the MSE. Fix any maximal invariant $M$ that is also in $\mathcal{T}_\sim$, and let $Z$ be a representation of $X$ for $\mathcal{T}_\sim$. Then the excess distortion w.r.t. MSE, $\mathrm{D}_{\mathcal{T}_\sim}$, is a valid distortion and

$$\mathrm{D}_{\mathcal{T}_\sim}[X, Z] = \sup_{f \in \mathcal{M}_b} \mathrm{R}_\mathrm{mse}[f(M(X)) \mid Z] \ . \tag{50}$$

*Proof.* For compactness of notation, we use, for example, $\mathbb{E}_{X,Y}$ to denote expectation with respect to $\mathbb{P}$, and $\mathbb{E}_X \mathbb{E}_{Y|X}$ to denote an iterated expectation. Our proof makes use of conditional expectation in $L^2$ being defined as projection in a Hilbert space. See [e.g., 29, Ch. 22-23].

Firstly, fix some $Y \in \mathcal{T}_\sim$. It is well known that

$$\mathbb{E}[||Y - \phi(X)||^2] = \mathbb{E}[\mathbb{E}[||Y||^2 \mid X] - ||\mathbb{E}[Y \mid X]||^2] + \mathbb{E}[||\mathbb{E}[Y|X] - \phi(X)||^2] \ .$$

Taking the infimum over all measurable $\phi\colon \mathcal{X} \to \mathbb{R}^k$, we have

$$\inf_{\phi\colon \mathcal{X} \to \mathbb{R}^k} \mathbb{E}_{X,Y}[||Y - \phi(X)||^2] = \mathbb{E}_X\left[\,\mathbb{E}_{Y|X}[||Y||^2] - ||\mathbb{E}_{Y|X}[Y]||^2\,\right]\,,$$

when $\phi(X) = \mathbb{E}[Y \mid X]$ $\mathbb{P}(X)$-almost everywhere. Now by the conditional invariance, $Y \perp\!\!\!\perp X \mid M(X)$ (Lemma 4), which implies $\mathbb{E}_{Y|X}[f(Y)] = \mathbb{E}_{Y|M(X)}[f(Y)]$ for any measurable function $f$. Therefore,

$$\inf_{\phi\colon \mathcal{X} \to \mathbb{R}^k} \mathbb{E}_{X,Y}[||Y - \phi(X)||^2] = \mathbb{E}_{M(X)}\left[\,\mathbb{E}_{Y|M(X)}[||Y||^2] - ||\mathbb{E}_{Y|M(X)}[Y]||^2\,\right]\,, \qquad (51)$$

when $\phi(X) := \phi'(M(X)) = \mathbb{E}_{Y|M(X)}[Y]$ $\mathbb{P}(X)$-almost everywhere.

Similarly, for fixed $Z$ t.v.i $\mathcal{Z}$ with $Z \perp\!\!\!\perp Y \mid M(X)$,

$$\mathbb{E}_{Z,Y}[||Y - \psi(Z)||^2] = \mathbb{E}_{M(X)}\mathbb{E}_{Y,Z|M(X)}[[||Y - \psi(Z)||^2] \qquad (52)$$

$$= \mathbb{E}_{M(X)}\left[\,\mathbb{E}_{Y|M(X)}[||Y||^2] - ||\mathbb{E}_{Y|M(X)}[Y]||^2\,\right]$$

$$+ \mathbb{E}_{M(X)}\left[\,||\mathbb{E}_{Y|M(X)}[Y] - \mathbb{E}_{Z|M(X)}[\psi(Z)]||^2 + \mathbb{E}_{Z|M(X)}[||\psi(Z) - \mathbb{E}_{Z|M(X)}[\psi(Z)]||^2]\,\right]$$

Observe that for any $Y \in \mathcal{T}_\sim$, (51) and the first term of (52) will cancel in the excess risk distortion. Therefore,

$$\mathrm{D}_{\mathcal{T}_\sim}[X, Z] = \sup_{Y \in \mathcal{T}_\sim} \inf_{\psi\colon \mathcal{Z} \to \mathbb{R}^k} \mathbb{E}_{M(X)}\left[\,||\mathbb{E}_{Y|M(X)}[Y] - \mathbb{E}_{Z|M(X)}[\psi(Z)]||^2\right.$$

$$\left.+ \mathbb{E}_{Z|M(X)}[||\psi(Z) - \mathbb{E}_{Z|M(X)}[\psi(Z)]||^2]\,\right]\,.$$

When taking the supremum over $Y \in \mathcal{T}_\sim$, $Y$ can only affect $\mathrm{D}_{\mathcal{T}_\sim}$ through its conditional expectation given $M(X)$, $\mathbb{E}_{Y|M(X)}[Y]$. That conditional expectation is a $\mathcal{B}(\mathcal{M})$-measurable function, so $\mathbb{E}_{Y|M(X)}[Y] \in \mathcal{M}_b$ for all $Y \in \mathcal{T}_\sim$. Therefore,

$$\{\mathbb{E}_{Y|M(X)}[Y]\colon Y \in \mathcal{T}_\sim\} \subset \mathcal{M}_b \subset \mathcal{T}_\sim\,,$$

and we can take the supremum over functions $f \in \mathcal{M}_b$ instead of $Y \in \mathcal{T}_\sim$, which yields

$$\mathrm{D}_{\mathcal{T}_\sim}[X, Z] = \sup_{f \in \mathcal{M}_b} \inf_{\psi\colon \mathcal{Z} \to \mathbb{R}^k} \mathbb{E}_{M(X)}\left[\,||f(M(X)) - \mathbb{E}_{Z|M(X)}[\psi(Z)]||^2\right.$$

$$\left.+ \mathbb{E}_{Z|M(X)}[||\psi(Z) - \mathbb{E}_{Z|M(X)}[\psi(Z)]||^2]\,\right]\,.$$

Expanding each quadratic and canceling terms involving $\mathbb{E}_{M(X)}[||\mathbb{E}_{Z|M(X)}[\psi(Z)]||^2]$, we find

$$\mathrm{D}_{\mathcal{T}_\sim}[X, Z] = \sup_{f \in \mathcal{M}_b} \inf_{\psi\colon \mathcal{Z} \to \mathbb{R}^k} \mathbb{E}_{M(X),Z}\left[||f(M(X)) - \psi(Z)||^2\right] \qquad (53)$$

$$= \sup_{f \in \mathcal{M}_b} \mathrm{R}_{\mathrm{mse}}[f(M(X)) \mid Z] \qquad (54)$$

$$= \sup_{f \in \mathcal{M}_b} \mathbb{E}_{M(X),Z}\left[||f(M(X)) - \mathbb{E}_{M(X)|Z}[f(M(X))]||^2\right]\,. \qquad (55)$$

Now, since conditional expectation given $Z$ is just projection onto the (Hilbert) subspace $L^2(\mathcal{B}(\mathcal{Z}))$, we have

$$\mathrm{D}_{\mathcal{T}_\sim}[X, Z] = \sup_{h \in \mathcal{M}_b \cap L^2(\mathcal{B}(\mathcal{Z}))^\perp} \mathbb{E}_X[||h(M(X))||^2]\,,$$

where $L^2(\mathcal{B}(\mathcal{Z}))^\perp$ is the subspace orthogonal to $L^2(\mathcal{B}(\mathcal{Z}))$ in $L^2(\mathcal{H})$. Since $L^2(\mathcal{B}(\mathcal{Z}))^\perp$ and $L^2(\mathcal{B}(\mathcal{M}))$ are both closed (sub-)Hilbert spaces, it is straightforward to show that so too is their intersection $L^2(\mathcal{B}(\mathcal{M})) \cap L^2(\mathcal{B}(\mathcal{Z}))^\perp$. The bounded (by $B$) elements of $\mathcal{M}_b \cap L^2(\mathcal{B}(\mathcal{Z}))^\perp$ are just the closed ball of radius $B$, so

$$\mathrm{D}_{\mathcal{T}_\sim}[X, Z] = \sup_{\substack{h \in L^2(\mathcal{B}(\mathcal{M})) \cap L^2(\mathcal{B}(\mathcal{Z}))^\perp \\ \mathbb{E}_X[||h(M(X))||^2] \leq B^2}} \mathbb{E}_X[||h(M(X))||^2]\,.$$

Now, since neither $\mathcal{M}_b$ nor $L^2(\mathcal{B}(\mathcal{Z}))^\perp$ is empty, their intersection is empty if and only if $\mathcal{M}_b \subset L^2(\mathcal{B}(\mathcal{Z}))$, i.e., $\mathcal{B}(\mathcal{M})yeah \subset \mathcal{B}(\mathcal{Z})$: all maximal invariants can be written as functions of $Z$. In that case, $\mathrm{D}_{\mathcal{T}_\sim}[X, Z] = 0$. Alternatively, if $\mathcal{M}_b \cap L^2(\mathcal{B}(\mathcal{Z}))^\perp$ is not empty, then choose some $h^*$ from it such that $\mathbb{E}_X[||h^*(M(X))||^2] = B^2 = \sup_{f \in \mathcal{M}_b} \mathrm{R}_{\mathrm{mse}}[f(M(X)) \mid Z]$.

Defining $d(x, z) := ||h^*(M(x))||^2$ yields a valid distortion $\mathrm{D}_{\mathcal{T}_\sim}$. $\qquad \square$

Note that the last last part of the proof makes it clear that for MSE the invariance distortion is either $0$ or $B^2$. Intuitively this happens because MSE risk is not invariant to bijections so it possible to make any predictive mistake arbitrarily bad by setting $M(X)$ to be arbitrarily large at this mistaken prediction. This suggests that for the MSE risk (and other loss functions that are not invariant to bijections) the expected excess risk might be better suited than the worst case excess risk that we considered.

As the invariant distortion under MSE is valid, we can now simply incorporate it into the rate distortion theorem to get the desired theorem.

**Theorem 17** (Rate-invariance for MSE). Let $\delta \geq 0$. Let $\mathcal{T}_\sim$ be the invariant tasks of interest w.r.t. $(\mathcal{X},\sim)$ and the MSE, $M$ be any maximal invariant in $L^2(\Omega, \mathcal{H}, \mathbb{P})$, and $Z$ be a representation of $X$ for $\mathcal{T}_\sim$. Let $Rate(\delta)$ denote the minimum achievable bit-rate for transmitting an i.i.d. source of $Z$ such that for any $Y \in \mathcal{T}_\sim$ we have $\mathrm{R}_{\mathrm{mse}}[Y \mid Z] \leq \delta + \mathrm{R}_{\mathrm{mse}}[Y \mid X]$. Then $Rate(\delta)$ is given by

$$Rate(\delta) = \inf_{P(Z|X) \text{ s.t. } \mathrm{D}_{\mathcal{T}_\sim}[X,Z] \leq \delta} \mathrm{I}[X; Z] \ . \tag{56}$$

where $\mathrm{D}_{\mathcal{T}_\sim}[X, Z] := \sup_{f \in \mathcal{M}_b} \mathrm{R}_{\mathrm{mse}}[f(M(X)) \mid Z]$.

*Proof.* The result (56) follows from the fact that $\mathrm{D}_{\mathcal{T}_\sim}$ is a valid distortion (Prop. 16) and the rate-distortion theorem (12). $\square$

As a corollary, we obtain the following lower bound for the rate, which may be useful in practice.

**Corollary 18.** Let $M$ be any $\mathbb{R}^k$-valued maximal invariant with a probability density with respect to Lebesgue measure. Let $g \colon \mathcal{M} \to \mathbb{R}^k$ be any homeomorphism of $M$ (including the identity map), and $f^*$ any maximum distortion achieving function. Then the following lower bounds hold:

$$Rate(\delta) \geq h(g(M(X))) - \frac{k}{2} \ln(2\pi e \delta / k) \tag{57}$$

$$Rate(\delta) \geq h(f^*(M(X))) - \frac{k}{2} \ln(2\pi e \delta / k) \ . \tag{58}$$

*Proof.* First, by the DPI, for any homeomorphism $g$ of $M$,

$$\mathrm{I}[X; Z] \geq \mathrm{I}[M(X); Z] = \mathrm{I}[g(M(X)); Z] \geq \mathrm{I}[f^*(M(X)); Z] \ .$$

The first inequality is an equality if and only if $X \perp\!\!\!\perp Z \mid M(X)$; the second if and only if $f^*$ is a homeomorphism of $M$ (and therefore is itself a maximal invariant). Second, using the translation-invariance of differential entropy and the fact that conditioning reduces differential entropy,

$$\mathrm{I}[M(X); Z] = h(M(X)) - h(M(X) \mid Z) \tag{59}$$
$$\geq h(M(X)) - h(M(X) - \mathbb{E}_{M(X)|Z}[M(X)] \mid Z) \tag{60}$$
$$\geq h(M(X)) - h(M(X) - \mathbb{E}_{M(X)|Z}[M(X)]) \ , \tag{61}$$

Now,

$$\mathbb{E}_{M(X),Z} \left[ ||M(X) - \mathbb{E}_{M(X)|Z}[M(X)]||^2 \right] \leq \mathbb{E}_{M(X),Z} \left[ ||f^*(M(X)) - \mathbb{E}_{M(X)|Z}[f^*(M(X))]||^2 \right]$$
$$= \mathrm{D}_{\mathcal{T}_\sim}[X, Z] \ .$$

The maximum entropy distribution subject to this second-moment constraint is the $k$-dimensional Gaussian distribution $\mathcal{N}(0, K)$, where $K$ is a diagonal covariance matrix with entries $K_{ii} = \mathbb{E}_{M(X),Z} \left[ (f^*(M(X))_{ii} - \mathbb{E}_{M(X)|Z}[f^*(M(X))]_{ii})^2 \right]$. The differential entropy of that Gaussian distribution is $\frac{1}{2} \log \left( (2\pi e)^k \det(K) \right)$, and by Jensen's inequality,

$$\log \det(K) = \sum_{i=1}^{k} \log K_{ii} = \sum_{i=1}^{k} \log \left( \mathbb{E}_{M(X),Z} \left[ (f^*(M(X))_{ii} - \mathbb{E}_{M(X)|Z}[f^*(M(X))]_{ii})^2 \right] \right)$$

$$\leq k \log \left( \sum_{i=1}^{k} \frac{1}{k} \mathbb{E}_{M(X),Z} \left[ (f^*(M(X))_{ii} - \mathbb{E}_{M(X)|Z}[f^*(M(X))]_{ii})^2 \right] \right)$$

$$= k \log(\mathrm{D}_{\mathcal{T}_\sim}[X, Z]/k) \,.$$

Putting this together with the first inequality in (59),

$$\mathrm{I}[M(X); Z] \geq h(M(X)) - \frac{k}{2}\log(2\pi e) - \frac{k}{2}\log(\mathrm{D}_{\mathcal{T}_\sim}[X, Z]/k)$$
$$\geq h(M(X)) - \frac{k}{2}\log(2\pi e) - \frac{k}{2}\log(\delta/k) \,.$$

The same argument holds for either of $g(M(X))$ or $f^*(M(X))$, yielding the stated lower bounds. $\quad\square$

## C   Variational objectives

In this section we will derive the variational bounds for estimating the rate and the distortion. In contrast to the proofs of main theoretical results (previous section) derivations will be less formal. Throughout this section we focus on the log loss and implicitly make all assumptions described in Appx. A.2.

Recall that the optimal bit-rate is simply the Rate Distortion function using our invariance distortion (Rate-Invariance function; Eq. (35) ), so any optimal encoder (for a given $\delta$) can be obtained by using the following arg minimum:

$$\underset{p(Z|X) \text{ s.t. } \mathrm{R}_{\log}[M(X)\,|\,Z]\leq\delta}{\arg\min} \quad \mathrm{I}[X;Z] \tag{62}$$

As optimization in machine learning is typically unconstrained, we prefer using the following Lagrangian formulation.

$$\underset{p(Z|X)}{\arg\min} \quad \mathrm{I}[X;Z] + \beta \cdot \mathrm{R}_{\log}[M(X)\,|\,Z] \tag{63}$$

Both of these formulations are equivalent in that the set of encoders that minimize Eq. (63) for some $\beta \in \mathbb{R}_{\geq 0}$ is equal to the set of encoders that minimize Eq. (62) for some $\delta \in \mathbb{R}_{\geq 0}$ [30, 31].

Note that due to the piece-wise linearity of our RI function (Fig. 3), Kolchinsky et al. [32] tells us that sweeping over $\beta \in \mathbb{R}_{\geq 0}$ using Eq. (63) will only enable us to learn the vertices of the RI function, namely, $Rate(\mathrm{H}[M(X)]) = 0$ for $\beta \in [0, 1[$ and $Rate(0) = \mathrm{H}[M(X)]$ for $\beta > 1$, while any point on the RI curve is simultaneously optimal for $\beta = 1$. In other words, although the solutions of Eq. (63) span the entire RI curve, it is, in theory, not possible to decide which points on the RI curve to obtain by sweeping over beta. Kolchinsky et al. [32] shows that this can be easily solved by considering the squared distortion $\mathrm{R}_{\log}[M(X)\,|\,Z]^2$, in which case sweeping over $\beta$ would be equivalent to sweeping over delta $\delta$ in Eq. (62). We did not see any difference in practice so preferred using the more understandable $\mathrm{R}_{\log}[M(X)\,|\,Z]$.

Both terms $\mathrm{I}[X;Z]$ and $\mathrm{R}_{\log}[M(X)\,|\,Z]$ are hard to estimate from samples, so the rest of the section is devoted to deriving variational upper bounds on them.

### C.1   Variational upper bound for the rate term $\mathrm{I}[Z;X]$

Let us discuss how to approximate the rate term $\mathrm{I}[X;Z]$. The mutual information is well known to be hard to estimate from samples [33, 34], but fortunately many variational bounds have previously proposed, see Poole et al. [35] for examples. In the following we denote a family of variational distributions over $Z$ (priors or entropy models) as $\mathcal{Q} := \{q \in \mathcal{P}(\mathcal{Z})\}$.

#### C.1.1   Mutual information bottleneck

The first bound that we consider is the standard upper bound on $\mathrm{I}[X;Z]$, e.g., in VAE [36] or VIB [37]. Specifically:

$$\mathrm{I}[Z;X] := \mathrm{H}[Z] - \mathrm{H}[Z\,|\,X] \tag{64}$$

$$= \mathrm{E}_{p(Z)}[-\log p(Z)] - \mathrm{H}[Z\,|\,X] \tag{65}$$

$$= \inf_{q\in\mathcal{Q}} \mathrm{E}_{p(X)p(Z|X)}\left[-\log \frac{p(Z)q(Z)}{q(Z)}\right] - \mathrm{H}[Z\,|\,X] \tag{66}$$

$$= \inf_{q\in\mathcal{Q}} \mathrm{E}_{p(X)p(Z|X)}[-\log q(Z)] - \mathrm{E}_{p(X)p(Z|X)}\left[\log \frac{p(Z)}{q(Z)}\right] - \mathrm{H}[Z\,|\,X] \tag{67}$$

$$= \inf_{q\in\mathcal{Q}} \mathrm{E}_{p(X)p(Z|X)}[-\log q(Z)] - \mathrm{D}_{\mathrm{KL}}[p(Z)\|q(Z)] - \mathrm{H}[Z\,|\,X] \tag{68}$$

$$\leq \inf_{q\in\mathcal{Q}} \mathrm{E}_{p(X)p(Z|X)}[-\log q(Z)] - \mathrm{H}[Z\,|\,X] \tag{69}$$

The approximation gap is then $\min_{q\in\mathcal{Q}} \mathrm{D}_{\mathrm{KL}}[p(Z)\|q(Z)]$. The bound has the advantage that if $p(Z) \in \mathcal{Q}$ then the bound is tight. The major issue with the mutual information bottleneck, is that no efficient compressors can in general achieve the rate given by it [38].[1] For example, if we decided to

---

[1]See Flamich et al. [39] or Schulman [40] for an $\mathcal{O}(\exp(\mathrm{I}[Z;X]))$ algorithm.

entropy code $Z$ using the entropy model $q(Z)$ then we would achieve $\mathrm{E}_{p(X)p(Z|X)}[-\log q(Z)]$ bits which is $\mathrm{H}[Z\,|\,X]$ more than what is given by our bound.[2]

### C.1.2 Entropy bottleneck

One specific case of the mutual information bottleneck which enables efficient compression, is when $Z$ is discrete and arises from a deterministic transformation of $X$. Indeed, in this case $\mathrm{H}[Z\,|\,X]=0$ so entropy coding (e.g. [42, 43]) can reach the rate given by our bound. Using the same derivation as for the mutual information bottleneck, we get,

$$\mathrm{I}[Z;X]=\mathrm{H}[Z]\leq\inf_{q\in\mathcal{Q}}\mathrm{E}_{p(X)p(Z|X)}[-\log q(Z)]. \tag{70}$$

This is the standard bound used in neural compressors [44, 45]. The entropy bottleneck bound has the following downsides compared to mutual information bottleneck:

- It is generally not true that for any $\delta$ the optimal rate can be achieved by a discrete and deterministic $Z$. For the specific case of $\delta=0$ and with Assumption 4 it is the case, as we can simply set $Z=M(X)$.
- It is not suitable for gradient based optimization w.r.t. to the encoder (due to the discreteness of $Z$) so we typically have to add noise during training [44] which can cause a mismatch between training and testing [38].

Despite these issues we will mostly use the entropy bottleneck bound in experiments as we want our method to give rise to practical compressors.

## C.2 Variational upper bound for the distortion term $\mathrm{R}_{\log}[M(X)\,|\,Z]$

Let us now consider variational upper-bounds on the distortion $\mathrm{R}_{\log}[M(X)\,|\,Z]$. For conciseness we will consider the same setting as in the main paper, i.e., log loss risk and countably many equivalence classes (Assumption 4). But it is easy to see that the direct distortion bound generalizes to any loss without Assumption 4. [3]

### C.2.1 Direct distortion

The obvious variational bound on the Bayes risk is the Bayes risk constrained to some functional family. $\mathcal{Q}':=\{q:\mathcal{Z}\rightarrow\mathcal{P}(\mathcal{M})\}$ denotes a variational family of regular conditional distributions (decoders), then,

$$\mathrm{R}_{\log}[M(X)\,|\,Z]:=\inf_{q}\mathrm{E}_{p(X)p(Z|X)}[-\log q(M(X)\,|\,Z)] \tag{71}$$

$$\leq\inf_{q'\in\mathcal{Q}'}\mathrm{E}_{p(X)p(Z|X)}[-\log q'(M(X)\,|\,Z)] \tag{72}$$

which comes from the fact that we are taking an inf over a subset $\mathcal{Q}'$ of all possible distribution. A simple derivation shows that the approximation gap is $\min_{q'\in\mathcal{Q}'}\mathrm{D}_{\mathrm{KL}}\left[p(M(X),Z)\|q'(M(X)\,|\,Z)p(Z)\right]$, so the bound is tight if $p(M(X)\,|\,Z)\in\mathcal{Q}'$. This direct distortion is simple, but typical variational families will require predicting ("reconstructing") an expected prediction $\mathrm{E}_{q'(M(X)|Z)}[M(X)|Z]$ which is challenging when $\mathcal{M}$ is in high dimension (e.g. unaugmented images).

### C.2.2 Contrastive distortion

We now consider a bound that does not require explicitly predicting $M(X)$, by considering a noise contrastive estimator [46]. Suppose that for any $Z$ we can sample from a sequence $\mathbf{M}=(M^{+},M_{1}^{-},\dots,M_{n}^{-})$, where $M^{+}\overset{\mathrm{d}}{\sim}p(M(X)\,|\,Z)$ and $\left\{M_{i}^{-}\right\}_{i=1}^{n}\overset{\mathrm{i.i.d.}}{\sim}p(M(X))$. Let $\mathcal{F}:=\{f:\mathcal{M}\times\mathcal{Z}\rightarrow\mathbb{R}\}$ be a variational family of discriminators which is used scores how likely $M(X),Z$ are to be sampled from the joint $p(M(X),Z)$ rather than the product of the marginal $p(M(X))p(Z)$, then,

$$lhs=\mathrm{R}_{\log}[M(X)\,|\,Z] \tag{73}$$

$$=\mathrm{H}[M(X)\,|\,Z] \qquad\qquad\text{Lemma 8}\quad \tag{74}$$

$$=\mathrm{H}[M(X)]-\mathrm{I}[M(X);Z] \tag{75}$$

---

[2]Bits-back coding [41] can efficiently reach the desired bit-rate only because it is in the lossless setting.

[3]$\mathrm{R}_{\mathrm{L}}[M(X)\,|\,Z]\leq\inf_{h\in\mathcal{H}'}\mathrm{E}_{p(X)p(Z|X)}[L(M(X),h(Z))]$ which comes from the fact that we are taking an inf over a subset $\mathcal{H}'$ of all possible predictors.

$$\leq \mathrm{H}[M(X)] - \inf_{f \in \mathcal{F}} \mathrm{E}_{p(Z)p(\boldsymbol{M}|Z)} \left[ \log n + \log \frac{\exp f(M^+, Z)}{\sum_{M' \in \mathbf{M}} \exp f(M', Z)} \right] \quad \text{InfoNCE} \quad (76)$$

$$= \inf_{f \in \mathcal{F}} \mathrm{E}_{p(Z)p(\boldsymbol{M}|Z)} \left[ -\log \frac{\exp f(M^+, Z)}{\sum_{M' \in \mathbf{M}} \exp f(M', Z)} \right] + (const) \quad (77)$$

Eq. (76) uses InfoNCE [47], which is a lower bound on mutual information [35, 48]. The last equation removes constants w.r.t. $p(Z \mid X)$ and $\mathcal{F}$, as these terms do not have to be optimized over. We see that we are only left with a log softmax term [4] that essentially aims to classify which of all the $M'$ comes from the $p(M(X) \mid Z)$. The bound is tight if the variational family $\mathcal{F}$ contains the optimal predictor and as the number of negatives tends to infinity. For a detailed discussion about noise contrastive estimation under the log loss, refer to [46, 49, 50].

Note that the contrastive distortion has the advantage of not having to reconstruct high dimensional data (e.g. for images), but it suffers from bias in the case where the number of negatives $n$ is small [35].

One additional derivation which we will need in the following section, is that an upper bound can also be obtained by replacing $M(X)$ by any other r.v. $U$ s.t. $U - M(X) - Z$ forms a Markov Chain. Indeed starting from Eq. (75), we have,

$$lhs = \mathrm{H}[M(X)] - \mathrm{I}[M(X); Z] \quad (78)$$

$$\leq \mathrm{H}[M(X)] - \mathrm{I}[U; Z] \quad \text{DPI} \quad (79)$$

$$\leq \inf_{f \in \mathcal{F}} \mathrm{E}_{p(Z)p(\boldsymbol{U}|Z)} \left[ -\log \frac{\exp f(U^+, Z)}{\sum_{U' \in \mathbf{U}} \exp f(U', Z)} \right] + (const) \quad (80)$$

The bound can still be tight if in addition we have the following Markov Chain $M(X) - U - Z$, which implies that $\mathrm{I}[U; Z] = \mathrm{I}[M(X); Z]$.

### C.3 Case study: VIC and BINCE under data augmentations

The derivations in the previous 2 subsections are relatively general and abstract. As a case study, we now discuss the two objectives that we propose in the main paper for the case where we have access to the desired data augmentation and where we use neural functional families. Namely, the variational families for the entropy model $p_\phi(Z)$, the encoder $p_\varphi(Z \mid X)$, the decoder $p_\phi(M(X) \mid Z)$, and the discriminator $f_\psi(X, Z)$ are all parametric neural families. Throughout this subsection we will consider that we only have access to $p(X)$ through a dataset $\mathcal{D} := \{x_i\}_i$ of samples which were independently sampled from $p(X)$.

Let us formalize what we mean by having access to the correct data augmentations. Let us denote as $A$ the r.v. over a set of augmentations $a : \mathcal{X} \to \tilde{\mathcal{X}}$, i.e., a stochastic process. Let $\tilde{X} = A(X)$ be the augmented source. Note that by $A(X)$ we mean the r.v. which arises by sampling an augmentation $a$ from the stochastic process $p(A)$, and then applying it to some samples $x$ from $X$.

**Assumption 7** (Augmentations). We assume knowledge of a random augmentation generator $A$ that satisfies the following two key properties

- **Retain invariance**. We require $A$ to retain the invariance structure to $(\sim, \mathcal{X})$, specifically, $X \sim A(X)$ almost surely.
- **Remove information**. We require $A$ to remove as much information as possible about the input. Specifically, $X \perp\!\!\!\perp A(X) \mid M(X)$ almost surely.

The first requirement is simple but clearly not sufficient. For example, the identity function does satisfy such requirement for any equivalence relation ($X \sim X$ by definition), yet it does not correspond to what we think as an augmentation because it does not remove any information about the input. The second requirement formalizes exactly what is required, namely that the augmentation must remove all information about the input besides the knowledge about its equivalence class (which is needed for the first requirement).

---

[4]Taking exponentials is not necessary, any function $g : \mathcal{M} \to \mathbb{R}_{\geq 0}$ would work as a discriminator, we use $g := \exp \circ f$ to ensure positivity as this has a nice softmax interpretation and is standard in practice. Our derivation is still general as we can set $f := log \circ g$.

The first requirement will typically hold. The second in more stringent. Note that it holds if for all equivalent examples $x \sim x'$ in $\mathcal{X}$ we have $p(A(x)) = p(A(x'))$. Indeed $p(A(x)) = p(\tilde{X}|x) = p(\tilde{X}|x, M(x))$ and similarly $p(A(x')) = p(\tilde{X}|x', M(x'))$, using $M(x) = M(x')$ we have $p(\tilde{X}|x, M(x)) = p(\tilde{X}|x', M(x))$ for all equivalent $x, x'$ which implies that $X \perp\!\!\!\perp \tilde{X} \mid M(X)$ as desired. In practice this only needs to hold for examples in our datasets, i.e., the second requirement holds if for all equivalent $x, x' \in \mathcal{D}$ in a dataset we have $p(A(x)) = p(A(x'))$. This is likely to hold in practice as the number of examples that are equivalent in a dataset will be small if $|\mathcal{X}| \gg |\mathcal{D}|$ as is typically the case. In particular, if a dataset does not contain any equivalent examples, i.e., for any $x, x' \in \mathcal{D}$ we have $x \not\sim x'$ then the requirement trivially holds.

### C.4   Issue: dealing with unknown $M(X)$

One issue which arises in practice is that we generally do not have access to $M(X)$. We will overcome this issue by taking advantage of the fact that we have access to data augmentations $A$ that induce our equivalence relation. Intuitively, we will treat the augmented r.v. $\tilde{X} := A(X)$ as if it were the source, and use the actual source $X$ instead of $M(\tilde{X})$. Note that by $A(X)$ we mean the r.v. which arises by sampling an augmentation $a$ from the stochastic process $p(A)$, and then applying it to some samples $x$ from $X$. From now on, let us denote as $Z \overset{\mathrm{d}}{\sim} p(Z|\tilde{X})$ the representation that arises from the augmented source. Under suitable conditions on $A$ we can replace the previous objective Eq. (63) with the following equivalent objective, which we denote as $\mathcal{L}_{\mathrm{RI}}^{\beta}$,

$$\underset{p(Z|A(X))}{\arg\min} \quad \mathrm{I}[A(X); Z] + \beta \cdot \mathrm{R}_{\log}[X \mid Z]. \tag{81}$$

By equivalence of those objectives we mean that for any $\beta \in \mathbb{R}_{\geq 0}$ the set of RD tuples $(\mathrm{I}[X; Z], \mathrm{R}_{\log}[M(X) \mid Z])$ that are achieved by solutions of Eq. (63) is equal to the set of RD tuples $(\mathrm{I}[\tilde{X}; Z], \mathrm{R}_{\log}[X \mid Z])$ that are achieved by solutions of Eq. (81). In other words, they generate the same segment of the RI function.

First, let us show why we can replace $X$ by $\tilde{X}$, i.e., show that for any $\beta$ we have that $\arg\min_{p(Z|X)} \quad \mathrm{I}[X; Z] + \cdot \mathrm{R}_{\log}[M(X) \mid Z]$ is equivalent to $\arg\min_{p(Z|\tilde{X})} \quad \mathrm{I}[\tilde{X}; Z] + \beta \mathrm{R}_{\log}[M(\tilde{X}) \mid Z]$. This can be seen from the fact that the optimal bit rate in Theorem 2 only depends on $\mathrm{H}[M(X)]$ and $\delta$. In particular, $Rate(\delta)$ does not depend on the distribution of the source inside the equivalence classes, $p(X|M(X))$. Indeed, an optimal representation will compress all that information.[5] As a result, we can attain the same optimal bit rate by considering any source $\tilde{X}$ that is a transformed version of $X$ as long as the transformation does not change the distribution of the maximal invariant, i.e., $p(M(X)) = p(M(\tilde{X}))$. This is clearly the case for $\tilde{X} := A(X)$ as our augmentation retains invariance (Assumption 7).

Now let us consider why and when replacing $M(\tilde{X})$ by $X$ makes sense. Using Prop. 11 we know that if $A$ is s.t. $\tilde{X} \sim X$ and $\tilde{X} \perp X \mid M(X)$ forms a Markov Chain then we can replace (up to constants which are not important for $\arg\min$) the distortion term $\mathrm{R}_{\log}[M(\tilde{X}) \mid Z]$ by $\mathrm{R}_{\log}[X \mid Z]$. These are exactly our requirements on augmentations (Assumption 7).

For the rest of this section we will thus be working with $\mathcal{L}_{\mathrm{RI}}^{\beta}$ (Eq. (81)) instead of Eq. (63). Note that this means that, in theory, we should always use the augmented $\tilde{X}$ from now on, i.e., not only at train time but also at test time.

### C.4.1   Variational Invariant Compressor (VIC)

As seen in the main text the VIC loss is essentially a neural compressor where inputs are augmented but not the target reconstructions. We derive it by combining our entropy bottleneck bound (Appx. C.1.2) and our direct distortion (Appx. C.2.2), which gives the following upper bound on $\mathcal{L}_{\mathrm{RI}}^{\beta}$,

$$\mathcal{L}_{\mathrm{RI}}^{\beta}(\varphi) := \mathrm{I}[A(X); Z] + \beta \cdot \mathrm{R}_{\log}[X \mid Z] \tag{82}$$

$$\leq \inf_{\theta} \mathrm{E}_{p(A)p(X)p_{\varphi}(Z|A(X))}[-\log q_{\theta}(Z)] \qquad \text{Eq. (70)} \tag{83}$$

---

[5]Note that the bit rate gains $\mathrm{H}[X \mid M(X)] - \delta$ clearly depend on $p(X|M(X))$, but not the actual bit-rate $\mathrm{H}[M(X)] - \delta$.

$$+ \beta \cdot \inf_{\phi} \mathrm{E}_{p(A)p(X)p_{\varphi}(Z|A(X))}[- \log q_{\phi}(X \mid Z)] \qquad \text{Eq. (71)} \qquad (84)$$

$$= \inf_{\theta,\phi} - \mathrm{E}_{p(A)p(X)p_{\varphi}(Z|A(X))}[\log q_{\theta}(Z) + \beta \log q_{\phi}(X \mid Z)] \qquad (85)$$

Using a Monte Carlo estimate for the expectation over $p(X)$, we get our desired objective,

$$\mathcal{L}_{\text{VIC}}{}^{\beta}(\phi, \theta, \varphi) := -\frac{1}{|\mathcal{D}|} \sum_{x \in \mathcal{D}} \mathrm{E}_{p(A)p_{\varphi}(Z|A(x))}[\log q_{\theta}(Z) + \beta \cdot \log q_{\phi}(x \mid Z)]. \qquad (86)$$

In practice, we approximate the expectation over $A$ and $Z$ using a single sample for computational efficiency. A full algorithm is provided in Algorithm 1 and illustrated in Fig. 4 of the main text.

---

**Algorithm 1** Variational Invariant Compressor (VIC). Single sample forward pass.

---

**Require:** Encoder $p_{\varphi}(Z|A(X))$, Entropy Model $q_{\theta}(Z)$, Decoder $q_{\phi}(X|Z)$
**Require:** Dataset $\mathcal{D}$, random augmentation generator $A$, Lagrange multiplier $\beta$
1: $x \leftarrow \text{select}(\mathcal{D})$          ▷ sample
2: $\tilde{x} \leftarrow \text{sample}(A(x))$          ▷ random augment
3: $z \leftarrow \text{sample}(p_{\varphi}(Z|\tilde{x}))$          ▷ encode
4: $\text{rate\_loss} \leftarrow -\log q_{\theta}(z)$          ▷ Entropy Bottleneck
5: $\text{distortion\_loss} \leftarrow -\log q_{\phi}(x|z)$          ▷ Direct Distortion
6: **return** $\text{rate\_loss} + \beta \cdot \text{distortion\_loss}$

---

Note that $\mathcal{L}_{\text{VIC}}{}^{\beta}$ tends to $\mathcal{L}_{\text{RI}}{}^{\beta}$ when using unconstrained variational families and as the dataset grows to infinity. This essentially shows that VIC objective (with infinite samples and unconstrained families) will learn the optimal deterministic and discrete $Z$ (as discussed in Appx. C.1.2), in particular, when $\beta > 1$ it will learn an encoder which is optimal for the lossless prediction regime.

### C.4.2 Bottleneck InfoNCE (BINCE)

VIC for images and data augmentation suffers from the issue that it needs a predictor which reconstructs a high dimensional image (as discussed in Appx. C.2.1). To solve this issue we discuss our BINCE objective, which as seen in the main text, is essentially a standard contrastive self-supervised (SSL) objective with an additional entropy bottleneck. We derive it by combining our entropy bottleneck bound (Appx. C.1.2) and our contrastive distortion (Appx. C.2.2).

Note that in Appx. C.2.2 for each $Z$ we needed a sequence $\mathbf{M}$ of outcomes of $M(X)$ that are sampled either from the conditional $p(M(X)|Z)$ or the marginal $p(M(X))$. As we will replace $M(X)$ by $X$ ( see Appx. C.4) we now need a sequence of r.v. $\mathbf{X} := (X^{+}, X_{1}^{-}, \ldots, X_{n}^{-})$ s.t. $X^{+}$ is sampled from the conditional $p(X|Z)$ while each $X_{i}^{-}$ are independently sampled from the marginal $p(X)$. Furthermore, as is standard in self-supervised learning (e.g. [47, 51]) we will actually be using a sequence $\check{\mathbf{Z}}$ of positive and negative representations instead of $\mathbf{X}$. We do so by independently augmenting and encoding each r.v. in $\mathbf{X}$. Using our requirement on the augmentations (Assumption 7) we thus have the following Markov Chain $\check{Z} - X - M(X) - A(X) - Z$. As a result, we can use $\check{\mathbf{Z}}$ instead of $\mathbf{X}$ in InfoNCE (see Eq. (78)). For conciseness we will denote the above sampling procedure as $p_{\varphi}(Z, \check{\mathbf{Z}} \mid A, X)$. We then have the following upper bound on $\mathcal{L}_{\text{RI}}{}^{\beta}$,

$$\mathcal{L}_{\text{RI}}{}^{\beta}(\varphi) := \mathrm{I}[A(X); Z] + \beta \cdot \mathrm{R}_{\log}[X \mid Z] \qquad (87)$$

$$\leq \inf_{\theta} \mathrm{E}_{p(A)p(X)p_{\varphi}(Z|A(X))}[- \log q_{\theta}(Z)] + (const) \qquad (88)$$

$$+ \beta \cdot \inf_{\psi} \mathrm{E}_{p(A)p(X)p_{\varphi}(Z|A(X))p(\check{\mathbf{Z}}|Z)}\left[ - \log \frac{\exp f_{\psi}(\check{Z}^{+}, Z)}{\sum_{\check{Z}' \in \check{\mathbf{Z}}} \exp f_{\psi}(\check{Z}', Z)} \right] \qquad (89)$$

$$= \inf_{\theta,\psi} - \mathrm{E}_{p(A)p(X)p_{\varphi}(Z,\check{\mathbf{Z}} \mid A,X)}\left[ \log q_{\theta}(Z) + \beta \log \frac{\exp f_{\psi}(\check{Z}^{+}, Z)}{\sum_{\check{Z}' \in \check{\mathbf{Z}}} \exp f_{\psi}(\check{Z}', Z)} \right] \qquad (90)$$

Using a Monte Carlo estimate for the expectation over $p(X)$, we get our desired objective,

$$\mathcal{L}_{\text{BINCE}}(\varphi, \theta, \psi) := -\sum_{x \in \mathcal{D}} \mathrm{E}_{p(A)p_{\varphi}(Z,\check{\mathbf{Z}}|A,\mathcal{D},x)}\left[ \log q_{\theta}(Z) + \beta \cdot \log \frac{\exp f_{\psi}(\check{Z}^{+}, Z)}{\sum_{\check{Z}' \in \check{\mathbf{Z}}} \exp f_{\psi}(\check{Z}', Z)} \right]. \qquad (91)$$

---

**Algorithm 2** Batch forward pass for BINCE

---

**Require:** encoder $p_\varphi(Z|A(X))$, entropy model $q_\theta(Z)$, discriminator $f_\psi(Z', Z)$
**Require:** augmentations $A$, data $\mathcal{D}$, Lagrangian coefficient $\beta$, batch size $b$
  1: $\mathbf{x} \leftarrow \text{select}(\mathcal{D})$ $b$ times                  $\triangleright$ sample
  2: $\tilde{\mathbf{x}} \leftarrow \text{sample})(A(\mathbf{x}))$              $\triangleright$ Random augment 1
  3: $\tilde{\mathbf{x}}' \leftarrow \text{sample})(A(\mathbf{x}))$              $\triangleright$ Random augment 2
  4: $\mathbf{z}, \mathbf{z}' \leftarrow \text{sample}(p_\varphi(Z|\tilde{\mathbf{x}})), \text{sample}(p_\varphi(Z|\tilde{\mathbf{x}}'))$        $\triangleright$ Encode
  5: $\mathbf{zs} \leftarrow \text{concat}(\mathbf{z}, \mathbf{z}')$
  6: rate_loss $\leftarrow$ average $-\log q_\theta(z_i)$ over $z_i \in \mathbf{zs}$     $\triangleright$ Entropy Bottleneck
  7: distortion_loss $\leftarrow 0$
  8: **for** $i \leftarrow 1, \ldots, b$ **do**
  9:  $z^+ \leftarrow \mathbf{z}'[i]$                $\triangleright$ Select positive
10:  softmax $\leftarrow \exp f_\psi(z^+, z)/(\sum_{z' \in \mathbf{zs}} \exp f_\psi(z', z))$     $\triangleright$ Softmax
11:  distortion_loss $-= \frac{1}{b} \log(\text{softmax})$     $\triangleright$ Contrastive Distortion
12: **end for return** rate_loss $+ \beta \cdot$ distortion_loss

---

In practice we approximate the expectation over $A, \check{Z}$ and $Z$ using a single sample for computational efficiency. Just as with VIC we have that $\mathcal{L}_{\text{VIC}}{}^\beta$ tends to $\mathcal{L}_{\text{RI}}{}^\beta$ when using unconstrained variational families and as the dataset and number of negatives $n$ grows to infinity.

In the main paper we provided a simple algorithm (Algorithm 1) to compute BINCE for a single example $x \in \mathcal{D}$. This is computationally intensive as it requires sampling one sequence of r.v. for each example. In practice, this is nevertheless easily amenable to batch computation. Indeed, negative representations $\check{Z}^-$ are positive representations $\check{Z}^+$ for a different example. As a result, we can first sample a batch $\mathbf{x} := (x_1, \ldots, x_n)$ from $\mathcal{D}$. Then augment it to two different sequences $\tilde{\mathbf{x}}, \tilde{\mathbf{x}}'$. And finally represent each sequences to obtain $\mathbf{z}, \mathbf{z}'$. Then for any $z_i := \mathbf{z}[i]$ we have that $z_i' := \mathbf{z}'[i]$ is a positive example while all other $z \in \mathbf{z}, \mathbf{z}'$ are negatives. We thus only need to sample a single representation per example in the dataset. A full algorithm for batch computations is provided in Algorithm 2 (using only one of the 2 augmented batches for notational convenience).

## C.5 CLIP as BINCE's distortion

One of our main experiment (Sec. 5.3), consists in using a pretrained CLIP to make a powerful image compressor. We are able to do so by realizing that CLIP essentially corresponds to BINCE's distortion (second term in Eq. (9) with the following choices:

- **Augmentation**: text to image transformation. CLIP's dataset contains pairs of associated images and detailed text $(x_{img}, x_{txt})$. The "augmentation" is then a function that maps $x_{img}$ to its associated $x_{txt}$ and vis versa. This will partition the joint image-text space of $\mathcal{X} := \mathcal{X}_{img} \cup \mathcal{X}_{txt}$ into sets, each of which are associated (directly or by transitivity) with a common text description or image.

- **Discriminator**: a dot product, i.e, $f_\psi(Z', Z) = Z'^T Z$.

- **Encoder**: a deterministic function defined by cases. Specifically, sampling from $p(Z|X)$ gives the output of the visual transformer (image encoder) $Z = ViT(X)$ if $X$ is an image and the output of the text transformer (text encoder) $Z = transformer(X)$ if $X$ is a sentence.

The only minor difference is that CLIP performs contrastive learning between text-image and image-text but never text-text and image-image. BINCE would instead make no distinction between modalities as the equivalence class is on the joint image and text space. Both are nevertheless valid approximations to $\text{R}[M(X)\,|\,Z]$.

Although CLIP's augmentation will always give rise to a valid equivalence relation, it would in theory recover the degenerate solution of $[x] = \mathcal{X}$ for all $x \in \mathcal{X}$ if the dataset was "infinite". Indeed, any image could possible just be described by the text "an image", which would recover the aforementioned degenerate solution. There are different ways of collecting the datasets that could avoid this issue, e.g., ensuring that the description is more precise than that. In practice, this is unlikely to be an issue as the dataset is finite.

Another possible theoretical issue of CLIP's augmentation/equivalence structure, is that it is likely that very few images have a common associated text in CLIP's dataset (or vis-versa). In theory, this would thus recover the degenerate solution where no points are equivalent to one another, i.e., $[x] = \{x\}$ for all $x \in \mathcal{X}$. In practice, this issue is probably avoided due to the fact that images will get clustered as long as the the text description is similar enough for the text encoder to provide (essentially) the same text encoding (due to computational/architectural constraints). I.e., the images will actually get partitioned based on the value of the *representation* of their associated text rather than the text itself.

# D   Extended related work

**Invariances and symmetries.** Invariances are ubiquitous in ML, as seen by the use of data augmentations [52] and invariant models [8, 53–58]. These force models not to rely on nuisances to improve generalization [59–61]. We directly discard such nuisances from the data to improve compression. Others have used symmetries in $X$ for lossless compression of multisets [18], graphs [62–64], or structured images [65–69]. We, instead, use invariance of the tasks $Y$ for lossless prediction.

**Neural lossy compression.** Most research in neural compression is either focused on estimation and optimization of the rate term [38, 70–76] or on developing perceptually meaningful distortions [77–80]. Our paper also develops a new distortion, but does not optimize for perception. Improvements in the rate objectives are orthogonal to our work and can also help our method.

**Self-supervised learning.** Our objective (Eq. (7) in main text) can be seen as contrastive SSL [47, 51] with an information bottleneck, a version of [81, 82] with an information instead of a variance bottleneck, a SSL VIB [37], or an invariant VAE [36]. At a higher level our work differs on two key aspects. First, minimizing the information $I[A(X); Z]$ arises from our desire to perform compression rather than to (optimally [83]) help generalization [84, 85]. Second, we provide the first formalism of a minimal pretext task $M(X)$ that retains all information about any invariant task. This is related to the multi-view literature, where one only needs to retain information which is invariant across views [86–89]. The main difference is that we prove the existence of a single pretext task.

The most similar setting to ours is the recent work of Mitrovic et al. [90], which (in Appx. D) analyses contrastive learning using equivalence relations. Specifically, they also consider tasks $Y$ whose conditional distribution are invariant to an equivalence relation. Their Theorem 1 is then similar to our Lemma 4, but only considers the restricted case of deterministic labeling and finite sample space $\mathcal{X}$. Furthermore, they only talk about invariant representations (sufficiency), while we characterize all invariant representations using the maximal invariant (necessity and sufficiency).

**Information theory and predictions.** Theorem 2 relates exactly predictive loss and compression rate. Although such results is to our knowledge (surprisingly) new, it fits in a long line of work that relates Bayes predictions and generalized information theory [9, 11, 83, 91–94].

**Maximal invariants and minimal sufficient statistics.** As seen by our coin toss example, if the marginal $p(X)$ is invariant to the equivalence, i.e., $X \in \mathcal{T}_\sim$, then maximal invariants coincide with minimal sufficient statistics [95, 96]. In our work we are interested in predicting a target $Y$ rather than reconstructing the source $X$. A sufficient statistic w.r.t. to another r.v. $Y$ is referred to as adequate statistics.[6] Maximal invariants can thus be seen as minimal adequate statistics for the set of all invariant tasks of interest $\mathcal{T}_\sim$. Using minimal adequate statistics as good representations for performing a task has been well investigated in ML to improving generalization [83, 84, 99–102]. The main difference with our work is that (i) we consider adequacy for a collection of tasks instead of a single task; (ii) minimality arises from a compression perspective rather than for generalization. Although we are not aware of any use of minimal adequacy for compression (even single task), minimal sufficiency is often used for compressing distributions [103, 104].

---

[6]The standard definition of adequacy from Skibinsky [97] also requires the statistic to be sufficient, here we use "adequacy in the wide sense" as defined by Takeuchi and Akahira [98]

# E  Reproducibility

In this section we provide further details of the hyperparameters chosen for the various experiments in the main text. The code to reproduce all experiments can be found at `github.com/YannDubs/lossyless`. We checkpoint and use the model which achieves the smallest *validation* loss for evaluation. Unless stated otherwise, all the models are trained for 100 epochs, using Adam [105] as the optimizer, and a batch-size of 128. The learning rate starts at 1e-3 that decreases exponentially until reaching 1e-6 at the end of training. For all convolutional layers we use Kaiming normal initialization[106], for all linear layers we use Kaiming uniform initialization[106], while all biases are always initialized at 0. Activation functions are ReLUs while other unspecified parameters are PyTorch [107] defaults. For our invariant models, instead of optimizing $I[Z; X] + \beta\, D_{\mathcal{T}_\sim}[X, Z]$ we optimize $\lambda\, I[Z; X] + D_{\mathcal{T}_\sim}[X, Z]$, which is a more standard formulation for VIB, VAE, and neural compressors. In the following sections we will sometimes refer to $\beta$ as $\frac{1}{\lambda}$.

## E.1  Banana

For the Banana dataset most of the arguments were selected so as to replicate Fig.1.B. from [108]. [7]

**Data.**    The data distribution is obtained by starting from a bivariate Gaussian $X \sim \mathcal{N}(\mathbf{0}, \mathrm{diag}([3, 0.5]))$. It is then transformed to a banana distribution using the following transformation: $x_2 = x_2 + 0.1x_1^2 - 9$. We then rotate it and shift it: $\mathbf{x} = (\mathrm{Rot}(-40) \cdot \mathbf{x}) + [-3, -4]^T$. For every epoch we resample $1024000$ new points, i.e., examples are never seen twice during training).

**Hyperparameters.** For all Banana experiments we use a 2 dimensional representation $Z \in \mathbb{R}^2$, a learning rate of 1e-3 that decreases exponentially until reaching 1e-6 at the end of training, and a batch size of 8192. The encoder (and decoder if there is one) is always a 2-hidden layer MLP with 1024 hidden neurons, and softplus activation. In all cases we an entropy bottleneck with a factorized prior from [109].

**Experiment: Fig. 5.** We train both a standard variational compressor (VC) and our variational invariant compressor (VIC). The downstream performance loss is the MSE when predicting the maximal invariant. In both cases we use $\lambda = 0.07$, which was chosen so that the downstream performance is similar for both. For VIC the data is first augmented using rotations, passed through an encoder, then the decoder predicts the maximal invariant $M : x \mapsto \mathrm{Rot}(225) \cdot [0, \|x\|_2]^T$, i.e., the point with the same radius but positioned at 225 degrees. Note that we use this maximal invariant (instead of the more natural $\|x\|_2$) to ensure that the reconstructions (codebooks) can be plotted in in a nice way in the original space $\mathcal{X}$. The choice of maximal invariant does not impact the learned partition of the space.

Each plot (Fig. 5 right) is generated by first taking a meshed grid of $500^2$ source points in $[-5, 5]^2$. Then we quantize every point in the mesh by passing it through our encoder. The partition of the space (delimited with pink contours) corresponds to all points in the mesh that got mapped to the same quantized representation. To obtain the codebook (pink dots), we pass the quantized representations through our learned decoders. Finally, we plot the distribution of our learned entropy model by rescaling the codes so that their area is proportional to the rate assigned by the entropy model, i.e., $-\log q_\theta(z)$.

To obtain rate-invariance curves (Fig. 5 left), we sweep over $\lambda = 0.00001, 0.0001, 0.001, 0.01, 0.1, 1, 10, 100, 1000$. For each point in Fig. 5a we plotted the average over 5 seeds and plotted in gray the standard errors (both in the rate and distortion direction). To compute the area under the curve we used the trapezoidal rule on each of the RI curves obtained by a single seed, we then aggregated to area under the curve for the 5 seeds to obtain the mean and standard error.

**Experiment: ??.** Here augmentations are translations on the $x$-axis. The BINCE model was trained using Algorithm 1, i.e., without assuming knowledge of the maximal invariant. For VIC we used $M : x \mapsto [0, x_2]^T$ as the maximal invariant.

**Experiment: Fig. 2.** Here augmentations are translations on the $y$-axis. For VIC we used $M : x \mapsto [x_1, 0]^T$ as the maximal invariant. To plot of the induced distribution in $\mathcal{M}$ (here the $x$-axis), we

---

[7]Their code can be found at `https://github.com/tensorflow/compression/blob/master/models/toy_sources/toy_sources.ipynb`

sample 1024000 new points, pass them through our encoder and decoder to obtains the codes, and then plot a histogram of the obtained codes (shown in salmon). In blue we also plot the (approximate) distribution of the source when marginalized our invariances (i.e. only consider the $x$ component).

## E.2 General Image framework

Here we discuss the framework which we used for most of our image experiments. Unless stated otherwise, for all (ours or standard) neural image compressors we use the same general framework and architectures.

Specifically, an image $X$ first passes through a ResNet18 [110] to obtain a 128 dimensional representation $Z$ that t.v.i $\mathbb{R}^{128}$. We then pass $Z$ through an entropy bottleneck with a scaled hyperprior entropy model from Ballé et al. [109] which gives us the quantized $\hat{Z}$. For our entropy bottleneck we used Bégaint et al.'s [111] implementation which is a Pytorch re-implementation of [109]. Note that the choice of entropy model and quantizer is orthogonal to our work, and any choice that works neural compression would work for us.

In the case where we have to decode an image (VIC and VC models), we pass the quantized $\hat{Z}$ through a linear layer to reshape it to a latent image in $\mathbb{R}^{2 \times 2 \times 256}$. The latent image then passes through a 4-layer transposed CNN decoder where after each layer the number of channels gets divided by two and the width and height of the image doubles. The last layer outputs an image with the correct number of channels (1 for MNIST, 3 for other datasets), which is treated as the reconstruction $\hat{X}$ of the augmented input (for standard compressors) or the non-augmented input (for our VIC).

To simulate how well you could perform on downstream tasks of interest (that are not known when learning the compressors), we evaluate how well a model can classify the labels from the dataset. Specifically, once the models are trained we freeze them, apply them to the dataset and train neural network to classify the inputs using either the quantized representation $\hat{Z}$ or the reconstruction $\hat{X}$. In the former case we a $|\mathcal{Z}| - 2048 - 2048 - |\mathcal{Y}|$ MLP with preactivation batch normalization [112]. In the latter case we use a ResNet18 for predictions.

Finally, we obtain the desired bit-rate by considering the expected log loss of the trained entropy model on the test distribution (i.e. theoretical bit-rate). The desired distortion is obtained by evaluating the predictor on the compressed test dataset.

## E.3 MNIST

For our MNIST [113] experiments we compare again our VIC (as described in Algorithm 1) against a standard neural compressor.

**Data.** In order to evaluate our framework in a relatively well understood setting we use the well known MNIST [113] dataset, which we rescale to $32 \times 32$ pixels. For this toy setting we want to understand how our model performs when trained with augmentations that induce the equivalent relation w.r.t. which we are invariant, i.e., we assume that we know the "correct" augmentation. To do so we augment both the training and the test set in the same way. Specifically, we apply random rotations sampled from $[-45, 45]$ degrees, random translations between $[0, 25]$ percentage of pixels, random shearing between $[0, 25]$ degrees, and random scaling by a factor in $[0.6, 1.4]$.

**Experiment: Fig. 1 and Fig. 6b.** For a fair comparison we took trained a standard compressor and a VIC so that the downstream accuracy on augmented MNIST is the closest possible to $99\%$ accuracy (note that augmented MNIST is slightly harder than standard MNIST). We then randomly sampled reconstructions for the source, standard reconstructions, and VIC reconstructions which we plot. The quantitative results are average over 5 runs and standard errors are provided in Fig. 6b.

**Experiment: Fig. 6a.** For the rate-error curve we swept over $\lambda = 0.001, 0.01, 0.03, 0.1, 0.3, 1, 3, 10, 100$ and plotted the curves and computed the area under the curve in the same way as previously discussed for the Banana rate-invariance curve.

## E.4 STL10

**Data.** We use the STL10 dataset [114] which is similar to CIFAR but with fewer labeled training examples. There are 10 classes of 96x96 pixeled, colored images. There are 500 labeled and 100000 unlabeled examples for training, 800 labels for test. Note that the unlabeled images come from a broader distribution of images. For augmentations we use horizontal flips, resizing and cropping, the

color transform and we randomly transform the image to gray scale with a likelihood of 0.2. As for MNIST we augmented both the train and the test distribution. The compressors were trained on the unlabeled data, while the predictors were trained on the train distribution.

**Hyperparameters.** We used an entropy bottleneck with a scaled hyperprior entropy model from [109]. When training with BINCE, VIC or VC, the encoder is a ResNet18 architecture. For hyperparameter tuning we randomly sampled 100 hyperparameters from the following search space: latent dimension size $(32-512)$, rate-distortion trade-off $\lambda$ $(10^{-13}, 100)$, the optimizer's (ADAM) learning rate $(10^{-4}, 10^{-3})$, the learning rate schedule(exponential decay or cosine decay), and the batch size $(64-128)$. For prediction on the learned features $\hat{Z}$ we trained an MLP with $1024-4096$ hidden units, one or two layers, and dropout probability between $0.0-0.5$. We optimized again the learning rate of the ADAM optimizer as before. For predictions from the reconstructions $\hat{X}$, we trained a ResNet18, with the same optimizer parameters as above.

**Experiment: Table 1.** In this experiment we compare the compression performance of PNG [115], WebP [116], JPEG [117], VC, VIC and BINCE. Since VC and VIC allow to predict either on features or on reconstructions we test both. We sweep uniformly over the log-scale of $\lambda = 10^{-5}, 100$ for the neural compressors and sweep the classical compressors over an equivalent quality range. The extensive results from which Table 1 is derived are in Table 4. The rate-distortion curves belonging to this experiment are Fig. 4. The rate-distortion curves correspond to the pareto optimal curves of the encoders and predictors from the 1000 models.

### E.5 Galaxy Zoo

**Data.** Celestial objects and events emit radio frequencies. These frequencies are recorded through large antennas. Modern radio astronomy relies on the aggregation of radio signals in time and space. This means that one antenna records over long stretches of time. Due to the rotation of the earth, this translates to many spatial measurements. Further, the inclusion of many antennas in various locations can provide a dense net of observations. The entire system is refereed to as aperture synthesis telescope (AST). Images of the sky are generated by combining the sequences of observations stemming from different antennas. ASTs generate enormous amounts of data, much of which is redundant and further will never be observed by humans. In fact commonly applied techniques to the observations, such as weighting (e.g. Briggs weighting) and blurring of signals removes information from the original observations. Our approach is thus a natural extension to the techniques already present in the radio astronomy community. However, the process of image reconstruction from radio frequency observation series is too complex for the scope of this paper. We thus work on the Galaxy Zoo 2 (GZ2) dataset, that contains of already inferred images of celestial objects. We believe that good rates on this dataset should hint at even better possible rates when working directly with the raw data. GZ2 contains 37 classification tasks, such as answering queries about shapes and counts information of galaxies. Although the tasks are classification tasks, we use the standard GZ2 evaluation that consists in regressing (RMSE evaluation) the expected (over different labellers) label probability. Our data is hosted on the kaggle platform. This means we have no access to test labels but only for total test loss. This is why we compute summary statistics on the validation data set. We choose to reduce the original dataset by center cropping to 256 pixels per dimension. We applied random rotations, horizontal and vertical flips, scaling $(1-1.3\times)$ and color transforms to this data. We used CNN encoders [44, 71] and an entropy bottleneck with a scaled hyperprior entropy model from [109].

**Hyperparameters.** For all experiments we used ResNet50 when predicting from images (i.e. encoders and predictors from reconstructions). As with the STL10 experiments, we trained each model and baseline by selecting a set of 100 hyperparameters randomly selected from a large search space. When training with BINCE, we sampled the latent dimension size $(32-2048)$, rate-distortion trade-off $\lambda$ $(1e-12, 1e-4)$, the optimizer's (ADAM) learning rate $(1e-4, 1e-3)$, the learning rate schedule(exponential decay or cosine decay) and the batch size $(64, 128)$. For prediction on the learned features we would train an MLP with 2048 hidden unit, two layers and dropout probability $(0.0-0.5)$. We optimized the again the learning rate of the ADAM optimizer as before. For the classical compressors we trained a ResNet50 on their reconstructions, with the same optimizer parameters as above.

### E.6 Pretrained CLIP

**Data.** In addition to the pretrained CLIP, we trained the entropy bottleneck. As we do not have access to the dataset from CLIP, we could not train the entropy bottleneck on the initial data. Instead we had to use a different dataset. We used MSCOCO [118] for image captioning, as we initially thought that we would need access to pairs of images and sentences to finetune CLIP.[8] Note that in our experiments the choice of dataset for training the entropy bottleneck (e.g. CIFAR10 [119]) had very little impact on the quality of the final compressor.

To evaluate our compressor in the most realistic setting possible, we selected 10 different datasets. The datasets were chosen so that (i) the source images are of very different shapes and content; (ii) they are easily be accessible online; (iii) images are already compressed by JPEG; (iv) neither the entropy bottleneck nor CLIP should have been trained on the selected datasets; (v) the task are very different classification tasks. To ensure that CLIP was (nearly) not pretrained on the selected datasets we selected a subset of the datasets that CLIP was evaluated on and which did not show significant data overlap (see Radford et al.'s [120] section 5 for a discussion about data overlap). Table 1 shows the details about the 10 datasets that we use for evaluating our model. When there is no prespecified validation split, we randomly sampled 10% of the training data for validation.

Table 1: Datasets used to evaluate our zero-shot compressor. -1 for the shape means variable.

| Dataset | Classes | Train size | Valid size | Test size | Metric | Shape |
|---|---|---|---|---|---|---|
| ImageNet [121] | 1000 | 1,281,167 | | 50,000 | accuracy | (-1,-1,3) |
| CIFAR10 [119] | 10 | 50,000 | | 10,000 | accuracy | (32,32,3) |
| CIFAR100 [119] | 100 | 50,000 | | 10,000 | accuracy | (32,32,3) |
| Cars196 [122] | 196 | 8,144 | | 8,041 | accuracy | (-1,-1,3) |
| Pets37 [123] | 37 | 3,680 | | 3,669 | balanced acc. | (-1,-1,3) |
| Caltech101 [124] | 102 | 3,060 | | 6,085 | balanced acc. | (-1,-1,3) |
| Food101 [125] | 101 | 75,750 | | 25,250 | accuracy. | (-1,-1,3) |
| STL10 [114] | 10 | 5000 | | 8000 | accuracy | (96,96,3) |
| PCam [126, 127] | 2 | 262,144 | 32,768 | 32,768 | accuracy | (96,96,3) |

**Reproducing the results.** For clarity and reproducebility we also provide a self contained script to train a very similar version to our compressor in Source Code 2 and Source Code 1. The main changes being that we change the training data (using CIFAR10), the entropy bottleneck (to the simpler factorized prior from [109]), and use a simpler evaluation pipeline (only use STL10 with a simplified MLP). The entire script (including evaluation and actual compression of a dataset) takes less than ten minutes to run on a single GPU and provides a general zero-shot compressor. The full code that we used is accessible at `github.com/YannDubs/lossyless`.

**Training the zero-shot compressor.** To train our compressor we first download the official pretrained CLIP model[9]. Specifically the vision transformer [128] that they refer to as `"ViT-B/32"`. We then freeze it, and add an entropy bottleneck with Ballé et al.'s [109] hyperprior. We then train the entropy bottleneck on the MSCOCO dataset.

To train the entropy bottleneck we need a distortion measure. In theory, to get our BINCE objective, we should use the distortion that CLIP was trained with, i.e., we should compress the representation in such a way that CLIP can still distinguish examples from the same equivalence class. Minimizing such distortion can lead to catastrophic forgetting as the representations only ensure that CLIP can distinguish equivalent example from our very small dataset. [10] We instead use a very simple MSE distortion in the representation space. Specifically, we trained the entropy bottleneck to minimize $\lambda \|z - \hat{z}\| - \log(q_\theta(z))$, where $\hat{z}$ denotes the reconstructed (quantized) representation. This can be seen in line 22 of Source Code 2.

---

[8] At the end we did not use the sentences as freezing CLIP worked very well.

[9] https://github.com/openai/CLIP

[10] We tried many ways of finetuning CLIP with very small learning rates and frozen components, but although the rate gains were large (around 2 to 3×) this lead to significant decrease in performance, most probably due to catastrophic forgetting.

One important point to notice is that in standard neural compressors the quantization is a component wise rounding to the closest integer. This typically does not constrain the compressor, as the compressor is trained in an end-to-end fashion so that the encoder can increase or decrease some components of $z$ to effectively increase or decrease the quantization. As our encoder is frozen, it cannot learn to adaptively change the scale of $z$ so we needed to learn the size of the quantization interval instead, i.e., the rounding precision. A simple (and equivalent) way of doing that consists in passing the representation through a (learned) component wise linear transformation (i.e. 2 parameters per component) then through the entropy bottleneck (quantization) and finally we reverse the linear transformation. This can be seen in line 12 and 14 of Source Code 2.

Generally we found that training the entropy model was very robust to hyperparameter changes. We used the following: 50 epochs, a 512 dimensional $z$ (given by CLIP), a batch size of 64, a learning rate of 1e-3 with 3, a scheduler that decreases the learning rate by $10\times$ every 12 epochs (i.e. uniformly 3 times during training), Adam with decoupled weight decay (AdamW; [129]) as an optimizer, weight decay of 3e-8 and a 32 dimensional side information for the hyperprior. For the our main CLIP compressor (CLIP+EB) we use an RD hyperparameter $\lambda = $ 5e-2 which is linearly annealed from 1e-7 to 1e-2 in the first 5 epochs of training (although annealing did not seem important). For our CLIP+EB$^-$ and CLIP+EB$^+$ (see Table 6) we respectively use $\lambda = $ 1e-2 and $\lambda = $ 1e-1. For data augmentations we used similar ones as used by CLIP namely, normalizing the image by the mean and std form their training dataset (`mean=[0.48145466, 0.4578275, 0.40821073]` and `mean=[0.26862954, 0.26130258, 0.27577711]`), resizing the smallest side of the image to $224 \times 224$ pixels with bicubic interpolation, then taking a random $224 \times 224$ crop. During the evaluation the random cropping is replaced by a center cropping.

**Evaluating the zero-shot compressor.** For evaluating the rate obtained by our compressor, we provide the negative log likelihood of our entropy model for the each test dataset. For evaluating the downstream predictive performance for each dataset we train a 2 hidden layer MLP of dimensions $512 - 2048 - 2048 - |\mathcal{Y}|$ with batch normalization and ReLU activation. For each dataset we provide the best model from 25 randomly sampled models, that arise by randomly sampling the following hyperparameters:

- batch size: $[32, 64]$ with logarithmic sampling.
- optimizer: Adam, SGD, AdamW.
- weight decay: $[$1e-7$, $1e-4$]$ with logarithmic sampling.
- learning rate: $[$1e-5$, $1e-3$]$ with logarithmic sampling.
- scheduler: exponential decay (with total decay by 100 or 1000), decreasing learning rate on validation loss plateau, cosine scheduler, decaying learning rate at fixed intervals.
- dropout [130]: $[0, 0.5]$.

We then provide the result of the best model. In Table 2 we compare those results to the same vision transformer that we use for CLIP, but trained directly on the raw images. These results were obtained from Radford et al.'s [120] table 10. For a better comparison to standard SSL models, in Table 6 we also provide the test accuracy of a linear layer (a support vector machine) from the representations. The regularization parameter of the SVM were all selected using 10 values and three fold cross validation. For the rates we compare to the average JPEG size of images in each datasets (all the selected datasets are compressed by default in JPEG). For the rates of the raw CLIP model we losslessly compress the representation using numpy's `savez` function (zip) [131].

### E.7 Minimal code to train the CLIP compressor in < 5 min.

In this section we provide minimal code to train our zero-shot compressor and to use our compressor to entropy code an entire dataset. Note that the model is simplified (e.g. using factorized prior instead of a hyperprior, and training on CIFAR10) so the bit-rates is slightly increased but it still achieves orders of magnitude gains compared to JPEG. We use CIFAR10 for training the entropy coder and STL10 for downstream evaluation (as both are downloadable through torchvision). To evaluate the model, we use a linear support vector machine from our representation $Z$.

For this minimal code, training takes around 3 minutes on a single GPU. The theoretical bit-rate that we achieve is around $1400$, while the practical bit-rate achieved by entropy coding is around $1700$. In comparison the bit-rate of JPEG (with 95 quality) is 4.71e4. The entropy coder compresses around 200 images per seconds, and decompresses around around 3 images per seconds. Decompression is slow

as we do not perform it in batch (for simplicity of the code), while encoding is batched processed. Downstream classification accuracy on STL10 is $98.7\%$ which is better than the uncompressed representations from CLIP, from which linear probe achieves $98.6\%$ accuracy.

To run the code you need first need to install the following libraries:

```
pip install git+https://github.com/openai/CLIP.git
pip install scikit-learn==0.24.2 lightning-bolts==0.3.3 compressai==1.1.4
```

The minimal boilerplate code (Source Code 1) downloads the training data and the pretrained CLIP (from line 42), trains the compressor (from line 46), entropy code the evaluation data (from line 57), and finally evaluates downstream performance (from line 67). The actual compressor is defined in Source Code 2.

```
1   import clip, torch, pytorch_lightning, numpy, tqdm, math, time
2   from torchvision.datasets import STL10,CIFAR10
3   from compressai.entropy_models import EntropyBottleneck
4   from torch.optim import Adam,lr_scheduler
5   from torch.utils.data import DataLoader
6   from pl_bolts.datamodules import SklearnDataModule
7   from sklearn.svm import LinearSVC
8
9   def clip_featurize_data(dataset, device):
10      with torch.no_grad():
11          Z,Y = [],[]
12          for x, y in tqdm.tqdm(DataLoader(dataset, batch_size=128, num_workers=16)):
13              Z += [pretrained.encode_image(x.to(device).half()).cpu().numpy()]
14              Y += [y.cpu().numpy()]
15      return numpy.concatenate(Z), numpy.concatenate(Y)
16
17  def compress_data(trainer, dataset, device,**kwargs):
18      start = time.time()
19      Z,Y = clip_featurize_data(dataset, device)
20      dm = SklearnDataModule(Z,Y,**kwargs)
21      out = trainer.predict(dataloaders=dm.train_dataloader())
22      Z_bytes = [o[0] for o in out]
23      flat_z = [i for batch in Z_bytes for i in batch]
24      Y = numpy.concatenate([o[1] for o in out], axis=0)
25      coding_rate = sum([len(s) for s in flat_z]) * 8 / len(flat_z)
26      sec_per_img = (time.time() - start)/ len(flat_z)
27      return Z_bytes, Y, coding_rate, sec_per_img
28
29  def decompress_data(compressor,Z_bytes):
30      start = time.time()
31      with torch.no_grad():
32          Z_hat = [compressor.decompress(b).cpu().numpy() for b in Z_bytes]
33      sec_per_img = (time.time() - start)/len(Z_hat)
34      return numpy.concatenate(Z_hat), sec_per_img
35
36  data_dir = "data/"
37  if torch.cuda.is_available():
38      device, precision, gpus = "cuda",16,1
39  else:
40      device, precision, gpus = "cpu",32,0
41
42  print("Downloading data and CLIP")
43  pretrained, preprocess = clip.load("ViT-B/32", device)
44  cifar = CIFAR10(data_dir,download=True,train=True,transform=preprocess)
45
46  print("Training compressor.")
47  start = time.time()
48  Z_cifar,Y_cifar = clip_featurize_data(cifar, device)
49  data_kwargs = dict(num_workers=16,batch_size=128,pin_memory=True,val_split=0.,test_split=0)
50  dm_cifar = SklearnDataModule(Z_cifar,Y_cifar,**data_kwargs)
51  compressor = ArrayCompressor(z_dim=512,lmbda=4e-2,lr=1e-1,lr_step=2)
52  trainer = pytorch_lightning.Trainer(gpus=gpus,precision=precision,max_epochs=10,logger=False)
53  trainer.fit(compressor, datamodule=dm_cifar)
54  print(f"Compressor trained in {(time.time() - start)/60:.0f} minutes.")   # 3 minutes
55
56  print("Download evaluation data and entropy code it.")
57  stl10_train = STL10(data_dir,download=True,split="train",transform=preprocess)
58  stl10_test = STL10(data_dir,download=True,split="test",transform=preprocess)
59  Z_b_train,Y_train,*_ = compress_data(trainer,stl10_train,device,**data_kwargs)
60  Z_b_test,Y_test,rate,enc_time = compress_data(trainer,stl10_test,device,**data_kwargs)
61  print(f"Bit-rate: {rate:.1f}. \t Compression: {1/enc_time:.1f} img/sec.")   # 1703.6 bits, 230.2 img/sec
62
63  print("Decompressing data (no batch processing).")
64  Z_train,_ = decompress_data(compressor,Z_b_train)
65  Z_test,dec_time = decompress_data(compressor,Z_b_test)
66  print(f"Decompression: {1/dec_time:.1f} img/sec.")   # 2.8 img/sec
67  print("Downstream evaluation.")
68  clf = LinearSVC(C=4e-3)
69  start = time.time()
70  clf.fit(Z_train, Y_train)
71  delta_time = time.time() - start
72  acc = clf.score(Z_test, Y_test)
73  print(f"Downstream STL10 accuracy: {acc*100:.2f}%.  \t Training time: {delta_time:.1f} ") # 98.65% , 0.5 sec
```

Source Code 1: Minimal boilerplate code for training a zero-shot compressor in less than 5 minutes. For the actual compressor (`ArrayCompressor`) see Source Code 2.

```python
class ArrayCompressor(pytorch_lightning.LightningModule):
    def __init__(self, *args, **kwargs):
        super().__init__()
        self.save_hyperparameters()
        self.bottleneck = EntropyBottleneck(self.hparams.z_dim)
        self.scaling = torch.nn.Parameter(torch.ones(self.hparams.z_dim))
        self.biasing = torch.nn.Parameter(torch.zeros(self.hparams.z_dim))
        self.is_updated = False

    def forward(self, batch):
        z, y = batch
        z = (z + self.biasing) * self.scaling.exp()
        z_hat, q_z = self.bottleneck(z.unsqueeze(-1).unsqueeze(-1))
        z_hat = z_hat.squeeze() / self.scaling.exp() - self.biasing
        return z_hat, q_z.squeeze(), y.squeeze()

    def step(self, batch, *args, **kwargs):
        z_hat, q_z, _ = self(batch)
        rate = -torch.log(q_z).sum(-1).mean()
        distortion = torch.norm(batch[0] - z_hat, p=1, dim=-1).mean()
        self.log_dict({"rate":rate / math.log(2),"distortion":distortion}, prog_bar=True)
        return distortion + self.hparams.lmbda * rate

    def training_step(self, batch, _, optimizer_idx=0):
        return self.step(batch) if optimizer_idx == 0 else self.bottleneck.loss()

    def predict_step(self, batch, _, __):
        return self.compress(batch[0]), batch[1].cpu().numpy()

    def compress(self, z):
        if not self.is_updated:
            self.bottleneck.update(force=True)
            self.is_updated = True
        z = (z + self.biasing) * self.scaling.exp()
        return self.bottleneck.compress(z.unsqueeze(-1).unsqueeze(-1))

    def decompress(self, z_bytes):
        z_hat = self.bottleneck.decompress(z_bytes, [1,1]).squeeze()
        return (z_hat / self.scaling.exp()) - self.biasing

    def configure_optimizers(self):
        param = [p for n, p in self.named_parameters() if not n.endswith(".quantiles")]
        quantile_param = [p for n, p in self.named_parameters() if n.endswith(".quantiles")]
        optimizer = Adam(param, lr=self.hparams.lr)
        optimizer_coder = Adam(quantile_param,lr=self.hparams.lr)
        scheduler = lr_scheduler.StepLR(optimizer, self.hparams.lr_step)
        scheduler_coder = lr_scheduler.StepLR(optimizer_coder, self.hparams.lr_step)
        return [optimizer,optimizer_coder], [scheduler,scheduler_coder]
```

Source Code 2: Minimal code for training an entropy bottleneck to convert a pretrained SSL model into a powerful zero-shot compressor. For the training and evaluation code see Source Code 1.

# F  Additional experimental results

## F.1  Banana

In Sec. 5.1 we compared a classical compressor to our VIC in the case of rotation invariant tasks.

Here show results for invariances to different equivalences and provide more intuition as to what BINCE and VIC achieve.

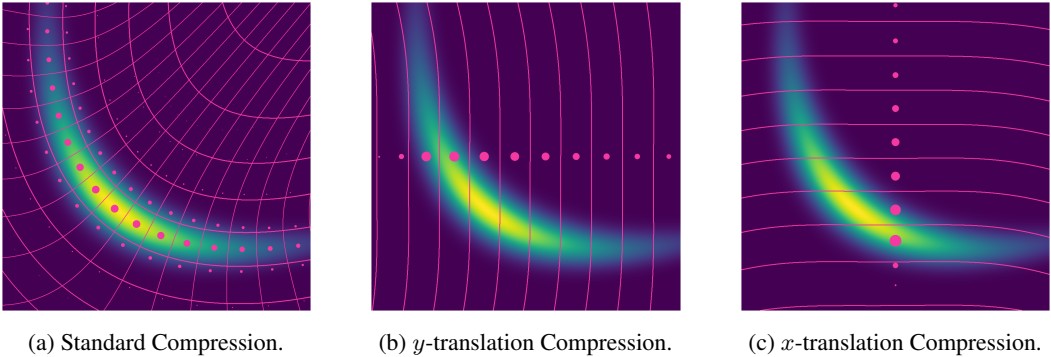

(a) Standard Compression.    (b) $y$-translation Compression.    (c) $x$-translation Compression.

Figure 1: Similar to **??** for the case of downstream tasks that are invariant to translations. (left) standard compression achieves $4.86$ bit rate; (center) our compressor VIC compressor for $y$-translations achieves $3.24$ bit rate; (center) our compressor VIC compressor for $x$-translations achieves a bit rate of $2.93$. All have a comparable similar invariant distortion.

$x$-**translation and BINCE. ??** considers the case where downstream tasks are invariant to $x$-translations. We used $M : x \mapsto [0, x_2]^T$ as the maximal during training. We see that our model can essentially perform as well on all downstream tasks for only $60\%$ of the bit-rate. Unsurprisingly we see that the codebook is in shape of horizontal stripes as these can cover the entire distribution with a few codes (small bit rate) while incurring a small invariance distortion (which only depends on the $y$ value).

We visualized a BINCE model (**??**) in addition to VIC. Although the exact partition for both models is quite different (BINCE does not seem to learn equal sized partitions), both models clearly learn to partition the space into horizontal stripes. Once important difference, is that VIC also provides a codebook (shown with pink dots), as it can reconstruct a quantized version of the input, while BINCE only learns a latent representation and does not provide any reconstructions.

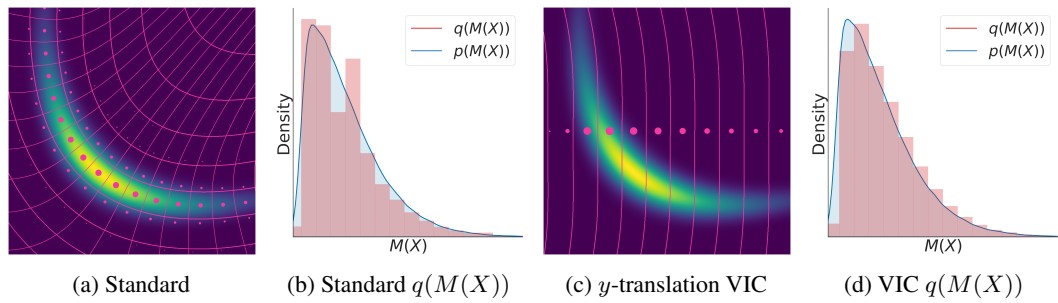

(a) Standard    (b) Standard $q(M(X))$    (c) $y$-translation VIC    (d) VIC $q(M(X))$

Figure 2: VIC improves compression of Banana distribution when downstream tasks are invariant to translation on the $y$-axis by (implicitly) estimating the density $p(M(X))$. (a) standard compression with a rate of $4.86$ bits and an invariant distortion of 7.67e-2 ; (b) the induced marginal distribution $q(M(X))$ of the $x$-value of the reconstructions from the standard neural compressor; (c) our compression with a rate of $3.24$ bits and an invariant distortion of 7.84e-2. (b) the induced marginal distribution $q(M(X))$ of the $x$-value of the reconstructions from the VIC;

$y$-**translation and induced distribution.** Fig. 2 considers the case where downstream tasks are invariant to $y$-translations. In this case the maximal invariant used during training is chosen to be $M : x \mapsto [0, x_2]^T$. Similarly to the case of $x$-translation and rotations, we see that our model can perform as well as a standard compressor for a fraction of the rate.

To provide a better intuition as to why this is the case we also plot the distribution of the reconstructions when marginalized over the $y$-axis. In other words we plot the distribution of $M(X)$ when applied to the reconstructions, i.e., the $x$ component of the reconstructions. We see that although the partition of the source space is very different for a standard compressor (Fig. 2a) and for our VIC (Fig. 2c), the induced distribution (and partition) in the marginalized space are actually very similar (Fig. 2b and Fig. 2d). This shows where our bit-rate gains come from. Indeed from Prop. 1 we know that in the case of invariant tasks one only needs to model the distribution of $M(X)$ (e.g. the distribution of the $x$ component here), and we see that both the standard compressor and VIC does that similarly well. The main difference being that VIC does so in an optimal way while the standard compressor needs to partition the input space in a finer way to achieve a similar induced partition in the $M(X)$ space.

**What is the relation between rate and predictions?** Theorem 2 shows that, for log loss, the minimum rate is linearly related to the loss $\delta$ in downstream performance. Our theory (Appx. B.6) suggests a logarithmic relationship for MSE. This is seen for VIC and VC in Fig. 5 of the main text (log scale $x$-axis).

**On lossy compression and equivalences.** Efficient lossy compression is about learning a partition (e.g. Voronoi diagrams, or **??** ) of the input space to map many inputs to the same code. We use the fact that any partition can be constructed from an equivalence relation [132] to learn compressors that are invariant to desired transformations. The shape of the partitions are then induced by the transformations, which perturb points in their quantization bins (equivalence classes), e.g., rotations in Fig. 5 of the main text. The size of the partition, e.g., disks width in Fig. 5 of the main text, depend on the desired performance $\delta$. The pink dots are representatives of the partition, i.e., maximal invariants. The key is that using our objectives we can learn arbitrary quantization using only desired transformations, which ML practicioners already use for data augmentations.

### F.2   MNIST

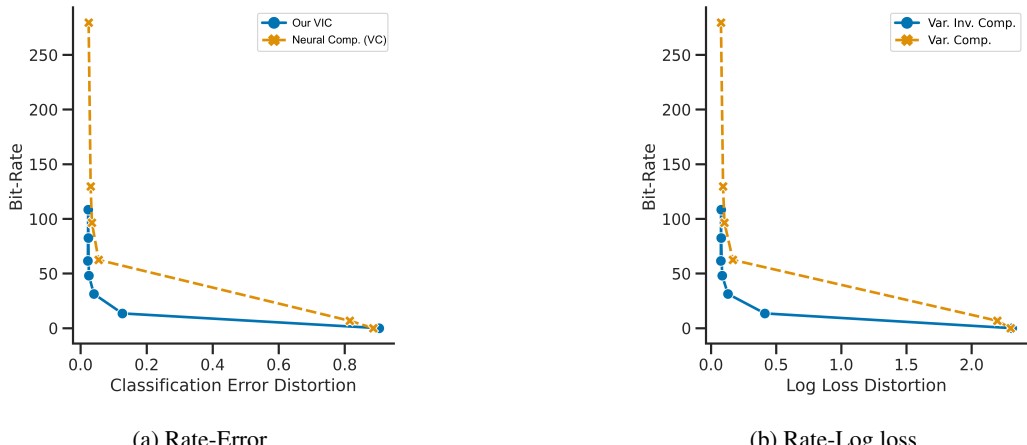

(a) Rate-Error          (b) Rate-Log loss

Figure 3: Augmented MNIST RD curves for downstream predictions are very similar when using classification error (left) instead of the log loss (right) from our theory.

**How do RD curves change if we use classification error instead of log loss?** Our theory Theorem 2 only ensures good downstream log loss risk. Nevertheless, we used here the classification error $(1 - \text{accuracy})$ throughout the main test as it is more commonly used for evaluating classification performance. Fig. 3 shows that RD curves are very similar for when using classification error instead of accuracy. This is not very surprising as log loss is the standard (differential) proxy of classification error in ML.

**What is the impact of the choice of augmentations?** The rate decreases when $A$ removes more information from $X$. Indeed, we have $I[X; A(X)] = I[X; M(X)] + I[X; M(X)] = I[X; M(X)] + 0 = H[M(X)]$, where the second equality comes from Assumption 7 and the last equality from the determinism of $M$. As a result we can rewrite Theorem 2 as $Rate(\delta) = I[X; A(X)] - \delta$. To illustrate this we trained our VIC and BINCE using three augmentation sets on MNIST, all of which keep the true label invariant but progressively discard more $X$ information: (i) standard image augmentations

Table 2: Using label-preserving augmentations that remove more information about $X$ decreases the rate without hindering classification performance. Single run.

|  | Augmentations | Rate $[\frac{\text{bits}}{\text{img}}]$ | Test Acc. [%] |
|---|---|---|---|
| VIC | Small set | 185.3 | 96.2 |
|  | Large set | 79.0 | 97.9 |
|  | Supervised (largest set) | 5.7 | 99.1 |
| BINCE | Small set | 256 | 95.3 |
|  | Large set | 131 | 97.8 |
|  | Supervised (largest set) | 5.9 | 97.8 |

such as random translations, shears, and rescalings; (ii) those same standard image augmentations, but drawn from larger ranges of possible translations, shears, scales, etc; (iii) supervised "augmentations" line in [133] that remove everything except label information, i.e., for every image $x$ let $x^+ = A(x)$ be a random image with the same label. Table 2 shows that using label-preserving augmentations that remove more information about $X$ greatly decreases the rate without hindering classification performance. The fact that the supervised augmentations achieve a much better rate, shows that typical SSL compression is still very far from single-task label compression. SSL compression retains information for at least $2^{79} \approx 10^{23}$ disjoint labels.

Table 3: End-to-end compression of augmented MNIST works much better than staggered compression for both VIC and BINCE. Single run.

|  |  | Rate $[\frac{\text{bits}}{\text{img}}]$ | Test Acc. [%] |
|---|---|---|---|
| VIC | Staggered | 477 | 98.0 |
|  | End-to-end | 79 | 97.9 |
| BINCE | Staggered | 358 | 97.4 |
|  | End-to-end | 131 | 97.8 |

**How much does end-to-end improve compared to staggered training?** We evaluated end-to-end training for both our losses against a staggered version that consists in first optimizing the distortion and then adding an entropy bottleneck to performing lossy compression of the learned representations (as in Sec. 5.3). Table 3 shows that end-to-end training can give large gains compared to the staggered method and that our compression gains with CLIP could be even further improved.

### F.3   STL10

In the main text we used the STL10 data set to answer some principled questions about our method in controlled experiments. We provide additional results here.

**How does the choice of distortion measures or bounds thereof affect RI curves?** Supplementing the results in the main text, we show more extensive results comparing the effect of the distortion measures (invariant or not) or bounds thereof (BINCE and VIC are different bounds on $R[M(X)\,|\,Z]$) on RI curves. When predicting from compressed representations $\hat{Z}$ (Fig. 4 left), BINCE achieves the best RI curves followed by VIC and VC. When predicting from reconstructions $\hat{X}$ (Fig. 4 right), VIC still performs a little better than VC although the gap shrinks. Table 1 shows all quantitative results for best achieved downstream performance (as in Table 1 from the main text).

Note that VIC and VC achieve much worst downstream performance than BINCE. Based on preliminary results, we believe that this comes from the fact that, for consistency, in all experiments we used ResNet18 encoders. Indeed, ResNet18 have an global averaging pooling layer that averages the "latent image" over spatial dimensions (width and height). As a result, the representations $\hat{Z}$ does not retain any spatial information, which is often useful for improving reconstructions. Preliminary results showed that removing this pooling layer improves downstream predictions significantly. Importantly, this impacts both VIC and VC so although the absolute performance improved by removing this layer the relative error did not seem to.

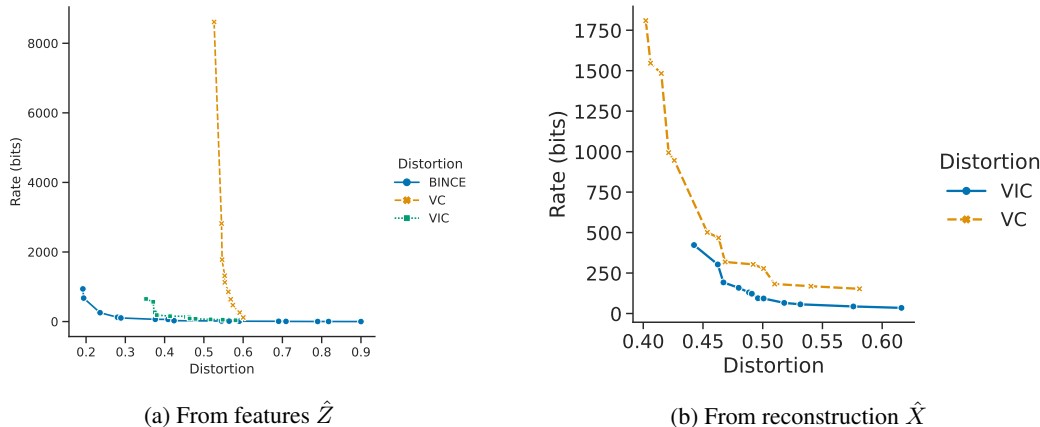

(a) From features $\hat{Z}$  (b) From reconstruction $\hat{X}$

Figure 4: BINCE achieves the best RI curves followed by VIC and then VC for STL10 data. Rate-error curves when predicting downstream tasks from: (left) compressed representations $\hat{Z}$, (right) reconstructions $\hat{X}$.

Table 4: We compare classical compression formats (PNG, JPEG, webP) to neural (VC) and invariant (BINCE, VIC) ones. BINCE achieves the same error rate but compresses $121.1\times$ better.

| Distortion | Predict from | Best error [%] | Rate [Mb/img] | Compression factor |
|---|---|---|---|---|
| PNG [115] | reconstructions | 19.2 | 14.20 | $1\times$ |
| JPEG [117] | reconstructions | 19.9 | 4.60 | $3.0\times$ |
| WebP [116] | reconstructions | 20.3 | 1.12 | $12.7\times$ |
| VC | reconstructions | 40.2 | 0.23 | $62.8\times$ |
| VC | features | 52.6 | 1.08 | $13.2\times$ |
| VIC (ours) | reconstructions | 44.3 | 0.05 | $268.6\times$ |
| VIC (ours) | features | 35.3 | 0.08 | $174.8\times$ |
| BINCE (ours) | features | 19.2 | 0.12 | $121.1\times$ |

For all our models the we estimated (using a naive sample estimate) the mutual information $\mathrm{I}[Z; Y]$ and found that $\hat{\mathrm{I}}[Z, X] = \hat{\mathrm{H}}[Y]$. This shows that all information about labels is retained, i.e., no images get compressed to the same Z but have different labels. The difference in test accuracy (which is also similar for training) must thus come from the fact that some information is easier to use/decode from [83, 134]. Indeed, our framework only concerns information retention rather than information usability. This is further supported by the fact that BINCE gets very good downstream accuracy, because it has been shown in theory [87, 88, 135] and in practice [47, 51] that contrastive representations are approximately linearly decodable.

Table 5: Hierarchical hyperprior works worst (higher rates) for low distortion, when compared to a factorized prior and a mutual information bottleneck on STL10 data.

| Bottleneck | Entropy model | Lossless Rate [bits/img] | Lossless loss |
|---|---|---|---|
| Entropy $\mathrm{H}[Z]$ | Factorized prior [109] | 598.5 | 0.3117 |
| Entropy $\mathrm{H}[Z]$ | Hierarchical prior [109] | 934.7 | 0.3100 |
| Mutual Information $\mathrm{I}[Z; X]$ | Unit Gaussian | 592.5 | 0.3074 |

**How does the choice of bounds on the rate term** $\mathrm{I}[Z; X]$ **impact RI curves?** For the main paper we always used the standard [44] neural compressor's upper bound on $\mathrm{I}[Z; X]$, namely, the entropy bound $\mathrm{I}[Z; X] \leq \mathrm{H}[Z] \leq \mathrm{E}_{p(Z,X)}[q_\theta(Z)]$. To understand the effect of using other bounds on $\mathrm{I}[Z; X]$ and different entropy models $q_\theta(Z)$. Specifically, in Fig. 5 we compare three different bounds on mutual information: (MI unitgaussian) the mutual information bound from VAE and VIB

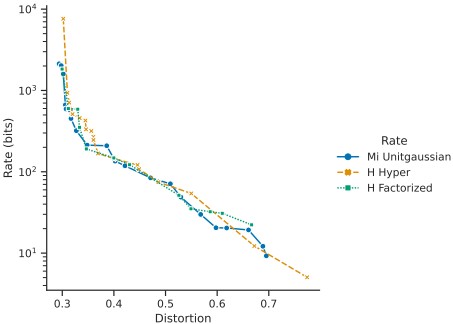

Figure 5: The choice of variational bounds on the rate term $I[Z; X]$ has little effect on RI curves for STL10 data. "MI unitgaussian" is the upper bound on mutual information used in VIB and VAE; "H factorized" is Ballé et al.'s [109] upper bound on $H[Z]$ with a factorized entropy model; "H hyper" is Ballé et al.'s [109] upper bound on $H[Z]$ with a hyperprior entropy model.

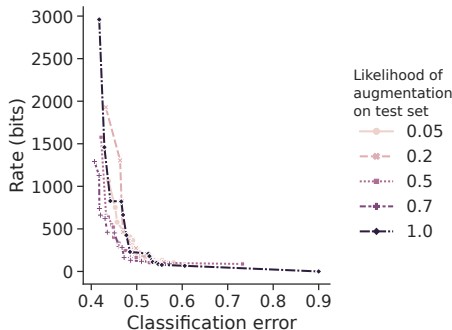

Figure 6: VIC is robust to distribution shifts in the augmentations as it is invariant to the augmentations. Specifically, test time shifts in augmentation probability seem to have little effect on the rate-distortion curve for the case of STL10 data.

$I[Z; X] \leq D_{KL}[p_\varphi(Z|X)] q_\theta(Z)$; (H Factorized) the entropy bound $H[Z] \leq E_{p(Z,X)}[q_\theta(Z)]$ where $q_\theta(Z)$ is Ballé et al.'s [109] factorized entropy model; (H Hyper) the entropy bound where $q_\theta(Z)$ is Ballé et al.'s [109] hyperprior entropy model. In our experiments, however, we find that neither of these choices influence the RD curves at typical distortion levels as seen Fig. 5. In our experiments we use "H Hyper", which does seem to enable very low rates (high distortions) but seems to perform worst at very high rates (see Table 5)

**How important is the distribution over augmentations?** As discussed in the main text, RD curves of our VIC show negligible difference when the distribution of augmentation shifts from training to test time. We provide this evidence in Fig. 6. Here we trained a VIC compressor on data with various augmentations, if these were (jointly) applied or not would be decided by a fair coin flip. At test time, we changed the coin to be biased with $p = 0.05, 0.2, 0.5, 0.7, 1.0$.

### F.4 Pretrained CLIP

In Table 6 we provide all the quantitative results for our zero-shot CLIP experiments from which we derived the tables in the main text.

We note that zero-shot compression can still be analysed using our framework. Indeed CLIP was trained on $400M$ sampled from a r.v. $X$ over images on the internet. As these datasets are on internet, they are samples from the joint $(X, Y)$ for a specific task $Y$. One can see this as a multi-task setting (each dataset is a distinct task).

**What is the effect of using a more powerful predictor from the representation?** In our framework we only discuss about information but never whether this information can easily be decoded by the predictors of interest. We investigated the effect of using more powerful predictors from our representation to understand how easy it is to decode the information in our representation. In particular, we evaluated all our CLIP compressors (i.e. at different $\beta$), by considering predictions from our compressed representation using a two layer MLP and using a linear classifier (SVM). Table 6 shows that the advantage of using an MLP compared to a linear model is small, which suggests that our CLIP compressed representation store information in a way that is easily decodable. This is typical from contrastive self-supervised models [47, 51].

**How does compressing SSL compare to compressing features form transfer learning?** In previous work, Singh et al. [136] had considered the case of compressing features from single-task transfer learning instead of self-supervised method. In Table 6 we compare compression of both type of features. Specifically "Transfer + EB" shows compression of a pretrained ResNet50 on ImageNet. We see that It generally performs worst than our CLIP compressor both in terms of test accuracies

Table 6: Converting a pretrained SSL model into a zero-shot compressor achieves substantial bit-rate gains while allowing test accuracies similar to predicting from raw images. CLIP refers to the original CLIP with lossless compression of the representations. CLIP+EB refers to our CLIP compressor. CLIP+EB$^-$ and CLIP+EB$^+$ are our CLIP compressors trained respectively for a larger and smaller bit-rate. We provide downstream evaluation using an MLP and a linear (SVM) predictor. Baselines: JPEG and compression of features from a ImageNet pretrained classifier (Transfer + EB).

| | | ImageNet | STL | PCam | Cars | CIFAR10 | CIFAR100 | Food | Pets | Caltech |
|---|---|---|---|---|---|---|---|---|---|---|
| **Rate [Bits/img]** | JPEG | 1.49e6 | 4.71e4 | 9.60e4 | 1.92e5 | 1.05e4 | 1.05e4 | 1.54e5 | 1.81e5 | 1.69e5 |
| | Transfer + EB | 3.95e3 | 3.33e3 | 3.99e3 | 3.18e3 | 3.92e3 | | 3.26e3 | 3.70e3 | 3.40e3 |
| | CLIP | 1.52e4 | 1.52e4 | 1.52e4 | 1.52e4 | 1.52e4 | 1.52e4 | 1.52e4 | 1.52e4 | 1.52e4 |
| | **CLIP+EB$^-$** | 2.47e3 | 2.46e3 | 2.61e3 | 2.59e3 | 2.53e3 | 2.54e3 | 2.39e3 | 2.33e3 | 2.46e3 |
| | **CLIP+EB** | 1.35e3 | 1.34e3 | 1.49e3 | 1.47e3 | 1.41e3 | 1.42e3 | 1.27e3 | 1.21e3 | 1.34e3 |
| | **CLIP+EB$^+$** | 9.63e2 | 9.52e2 | 1.49e3 | 1.52e2 | 1.02e2 | 1.09e3 | 8.89e2 | 8.35e2 | 9.53e2 |
| **Test Accuracies [%]** — [120] | JPEG | 76.6 | 99.0 | 82.6 | 49.1 | 96.7 | 86.3 | 81.8 | 90.4 | 94.5 |
| | CLIP [120] | 76.1 | 98.3 | 83.9 | 81.8 | 95.1 | 80.5 | 88.8 | 90.0 | 93.0 |
| MLP | Transfer + EB | 72.7 | 96.1 | 79.4 | 42.0 | 87.0 | | 66.8 | 91.3 | 89.9 |
| | CLIP | 76.5 | 98.6 | 84.5 | 80.8 | 95.3 | 80.9 | 88.5 | 89.7 | 93.2 |
| | **CLIP+EB$^-$** | 76.6 | 98.7 | 82.7 | 80.4 | 95.3 | 80.9 | 88.5 | 89.6 | 93.5 |
| | **CLIP+EB** | 76.3 | 98.7 | 80.9 | 79.6 | 95.2 | 80.1 | 88.3 | 89.5 | 93.4 |
| | **CLIP+EB$^+$** | 76.0 | 98.7 | 80.1 | 78.9 | 94.8 | 78.6 | 87.6 | 88.6 | 92.9 |
| Linear | CLIP | | 98.6 | 83.8 | 80.8 | 95.0 | 79.8 | 85.0 | 89.3 | 93.8 |
| | **CLIP+EB$^-$** | | 98.7 | 83.2 | 80.8 | 95.0 | 79.7 | 85.0 | 89.2 | 93.6 |
| | **CLIP+EB** | | 98.7 | 81.1 | 79.9 | 94.8 | 79.0 | 83.6 | 88.3 | 93.7 |
| | **CLIP+EB$^+$** | | 98.6 | 80.5 | 78.9 | 94.4 | 80.5 | 82.5 | 87.8 | 93.5 |

and bit-rate. One issue with this comparison is that the architectures of both models are not the same is likely that "transfer+EB" does not even perform better than CLIP on ImageNet.

## F.5  Galaxy Zoo

Table 7: Comparisons between pretrained CLIP BINCE, a BINCE trained end-to-end, and SOTA perceptual compressors on Galaxyzoo data. CLIP BINCE achieves the smallest bit-rate.

| Compressor | rate [Mb/img] | test loss [ ] | val. loss mean [ ] | median [$10^{-3}$] | max [ ] | min [$10^{-7}$] | std [$10^{-2}$] |
|---|---|---|---|---|---|---|---|
| PNG | 53.73 | 0.007 | 0.008 | 0.86 | 0.07 | 1.04 | 1.62 |
| JPEG | 1.68 | 0.012 | 0.013 | 1.25 | 0.11 | 1.23 | 2.61 |
| WebP | 0.48 | 0.010 | 0.011 | 1.20 | 0.10 | 1.16 | 2.29 |
| BINCE (CLIP) | 0.33 | 0.011 | 0.011 | 1.13 | 0.10 | 7.59 | 2.29 |
| BINCE (end to end) | 1.77 | 0.012 | 0.012 | 1.43 | 0.11 | 1.31 | 2.50 |

Humanity observes earth and sky at high temporal and spatial resolution, this can easily fill entire data centers. What is more, multiple copies of these series often exist over the world. At recording time it is usually not clear what kind of queries need to be answered about the recordings in the future; What was the weather like 10 years ago? Did a glacier resolve here? ect. To investigate our method in such real world scenario we compressed the GalaxyZoo telescope dataset (GZ2) and its 37 classification tasks. In Table 7, we compare a classical lossless and lossy method, to our BINCE at same distortions.

**How well does CLIP pretraining work compared to in domain training?** In the main paper we have seen that CLIP pretraining gives rise to a compressor that generalizes very well across different datasets. We evaluated how well the CLIP compressor generalizes compare to training BINCE directly end-to-end on GZ2's training set. Table 7 shows that the CLIP compressor works much better ($4\times$ rate gains) than the end-to-end BINCE. This suggests that pretraining can really be beneficial for training invariant compressors, and that our CLIP compressor can generalize very well across datasets.

**How does our CLIP compressor generalize to very different images compared to SOTA compressors?** In the main text we have seen that our CLIP compressor generalizes very well across

different datasets compared to high quality JPEG. To better understand the limits of the generalization capacity of our CLIP compressor, we compared it to a SOTA classical compressor (WebP) on images that are completely different than the ones CLIP was trained on, namely Galaxy images (not typical images on internet). We see in Table 7 that in this challenging setting our CLIP compressor only achieves relatively small gains (30%).