# OpenReview forum: "Lossy Compression for Lossless Prediction"
_NeurIPS.cc/2021/Conference — NeurIPS 2021 Spotlight_

### Official Review · Reviewer_YGFN · 2021-07-13

**Rating:** 6
**Confidence:** 4

**Summary:**

This paper proposes lossy compression algorithms that only compress task-relevant information in order to obtain a better bit-rate while maintaining the performance of the downstream task. Specifically, the paper proposes to augment the training samples before encoding, and the decoder aims to reconstruct the original image (another proposed model uses contrastive loss to do this). In this way, the model can learn representations that are invariant of the input augmentations, which could lead to smaller bit-rate.

**Limitations And Societal Impact:**

Limitations are discussed in Section 7. This includes the methods' irrecoverable loss of information, and instability when facing task changes.

**Main Review:**

Given that transfer learning is of great importance in machine learning, the task of compressing data only for certain downstream tasks is definitely worth exploring. However, though the paper made a good effort to formulate this problem, I think the experiments used in the paper might be insufficient to demonstrate the effect of their proposed technique — performing data augmentation during training leads to better lossy compressors:

 - The author claimed in the introduction that they aim to “ensure good performance on any future tasks of interest”. This setup was only tested in the CLIP experiment in Section 5.3, where the authors trained a compressor and validated on 8 datasets unseen during training. However, the good results may simply come from the fact that the latent representations of CLIP (a large pre-trained model analogous to BERT) already have a low bit-rate. That is, training any lossy compressor with the latent representations as input could lead to similar rate gains.

To address this concern, I would recommend the authors to at least compare the bit-rate also with other modern lossy compressors trained on latent representations generated by CLIP. It would also be great if the authors to demonstrate the performance gain provided by different data-augmentation techniques. In general, only comparing with JPEG does not provide useful information about how good the proposed model is since the goal of JPEG is to compress images while not changing its visual appearance by too much.

 - Need to justify the proposed models is not simply compressing the labels. For example, in the experiment in Section 5.2, performance drop and compression gain are compared. However, even by simply compressing the labels, one can get zero decreases in test accuracy and achieve a much better compression gain. The proposed models (VIC, BINCE) are of course not doing this explicitly. However, I am a bit concerned that this could be accidentally done implicitly. For example, consider an MLP with hidden layer size 32 - 16 - 8. If the bottleneck is set at the layer with 32 hidden neurons, the bit-rate could be worse than when the bottleneck is set at the last layer (8 hidden neurons).

I think the authors should design a better evaluation metric to avoid the above trivial case.

 - Overall, I like the high-level idea of the paper, but it seems hard for me to separate the cause of the bit-rate gain: is it caused by the good latent representations learned by SSL models, or the proposed invariant compression technique plays a critical role there? For example, if we have a powerful SSL model trained to classify ImageNet samples, then CIFAR10 can be faithfully solved only by looking at the 1000-class imagenet label. I believe this can be a good paper if this concern is addressed, and I would be willing to increase my score if the authors convince me during rebuttal.

**Time Spent Reviewing:**

3

---

> ### Author Response · Authors · 2021-08-10
> **Existing non-CLIP experiments demonstrate the impact of augmentation during training**
>
> > experiments used in the paper might be insufficient to demonstrate [...] performing data augmentation during training leads to better lossy compressors
>
> In addition to the new CLIP experiments discussed above, we believe that our existing experiments on other models support the claim that learning compressors with good augmentations leads to better compression. Indeed, we used the exact same compressor architecture both with augmentations (our VIC loss function) and without (the standard VC loss function) to demonstrate that augmentations result in large compression gains:
> - Banana: 48 -> 36 bits (line 226)
> - MNIST: 130 -> 48 bits (Figure 1)
> - STL10: 1080 -> 120 bits (Reported in Table 1 as percent improvement)
>
> Still, we agree that this point could have been more strongly made.  We are now including new experiments on MNIST ([see our response to Reviewer QdsL](https://openreview.net/forum?id=wZrOOO9XBn&noteId=-Pm_oxP0u6)) and with SimCLR ([see CLIP / SimCLR comparison above](https://openreview.net/forum?id=wZrOOO9XBn&noteId=uIsg_Ou_OYf)), which further demonstrate the impact of augmentation on compression.

---

> ### Author Response · Authors · 2021-08-10
> **We are implicitly compressing labels of many tasks in an unsupervised fashion (new experiment).**
>
> In short, implicit compression of the labels of many tasks is exactly what we are trying to do, the key is to do so in an *unsupervised* / *self-supervised* fashion for *many* tasks.
>
> > Need to justify the proposed models is not simply compressing the labels. However, I am a bit concerned that this could be accidentally done implicitly
>
> If we could compress jointly the labels from all possible tasks $\mathcal{T}$ of interest, then that would be the optimal thing to do and exactly what we would like to achieve. This not usually possible as:
> - the set of desired tasks $\mathcal{T}$ are rarely known/accessible at compression time;
> - even if they were known, $\mathcal{T}$ might be an extremely large collection of tasks and thus jointly compressing labels from all those tasks is not computational feasible.
>
> We overcome these issues with the use of maximal invariants (in our theory) to retain only the desired label information, and by using augmentations (in practice) to implicitly say what information can be removed.
>
> Still, your question is natural, and we want to clarify that our methods are still very far from retaining only the label information of the tasks on which we tested. To demonstrate this, we added a new experiment on MNIST that compares the following:
> 1. self-supervised compression using standard data augmentation. E.g., $x$ is a cat and $x^+$ a rotated version.
> 2. supervised compression using supervised augmentation ([Khosla et al., 2021](https://arxiv.org/abs/2004.11362)), i.e, $A(x)$ samples randomly in the training set an example that has the same label as y. E.g., $x$ is a cat and $x^+$ is another cat.
>
> |       | Augmentation | Bit-rate | Test Accuracy |
> |-------|--------------|:--------:|:-------------:|
> |  VIC     | Self-supervised       |   79.0   |      97.9     |
> |   | Supervised   |    5.7   |      99.1     |
> | BINCE   | Self-supervised   |    131   |      97.8     |
> | | Supervised   |    5.9   |      97.8     |
>
> Those results show that the supervised case achieves a much better rate (close to the optimal $log(10) \approx 3.32$) and so typical SSL compression is still very far from single-task label compression. SSL method retains information for at least $2^{79} \approx 10^{23}$ disjoint labels.

---

> ### Author Response · Authors · 2021-08-10
> **CLIP reps. outperform transfer learning reps. (new baseline).**
>
> > In general, only comparing with JPEG does not provide useful information since the goal of JPEG is to compress images while not changing its visual appearance by too much
>
> We completely agree and now added a sentence making it clear that is not a fair comparison. To address this, we now compare to [Singh et al’s (2020)](https://arxiv.org/abs/2007.11797) idea of compressing features from single-task transfer learning instead of self-supervised methods. While this is also not a perfectly fair comparison, we do not know of any perfectly fair comparisons, as we are the first to formalize this invariant multi-task setting.
>
> Specifically, we compare the lossy compression of features from a pretrained Resnet50 on Imagenet and CLIP:
>
> |                 |                   | STL  | PCam | Cars | CIFAR10 | Food | Pets | Caltech |
> |-----------------|-------------------|------|------|------|---------|------|------|---------|
> | Rate [bits/img] | CLIP              | 952  | 1090 | 1070 | 1020    | 889  | 835  | 953     |
> |                 | ImageNet Transfer | 3339 | 3994 | 3179 | 3923    | 3257 | 3699 | 3401    |
> | Test Acc. [%]   | CLIP              | 98.7 | 80.1 | 78.9 | 94.8    | 87.6 | 88.6 | 92.9    |
> |                 | ImageNet Transfer | 96.1 | 79.4 | 42.0 | 87.0    | 66.8 | 91.3 | 89.9    |
>
> As we see, on most dataset compressing CLIP gives better rate and downstream predictions.

---

> ### Author Response · Authors · 2021-08-10
> **We think that what we did in our CLIP experiments might be what you propose.**
>
> > I would recommend the authors to at least compare the bit-rate also with other modern lossy compressors trained on latent representations generated by CLIP.
>
> We believe that what you are proposing might be what we did, but we are not sure. To clarify, we trained a simple neural lossy compressor with a MSE distortion to compress CLIPs frozen representations. We used an extremely light-weight variant of Balle’s hyperprior entropy model (learning just the rounding precisions), but any modern lossy compressor of vectors should work.
>
> We would like to emphasize: we view our contribution in section 5.3. as the observation that SSL with an entropy bottleneck targets the optimal bit-rate for invariant multi-task compression. Together with the guidance that we provide on the choice of augmentations (see [CLIP vs SimCLR](https://openreview.net/forum?id=wZrOOO9XBn&noteId=uIsg_Ou_OYf)), this is what made us realize that lossy compression of CLIP would perform very well.  Although this method is very simple, we are not aware of *any* other paper that has used SSL for compression or pointed out any theoretical connection.  We now clarified this in the text.
>
> Ideally, we would have trained CLIP end-to-end with the lossy compressor, but that requires access to CLIP’s dataset.  We now added a simple MNIST experiment that investigates potential gains from end-to-end training. The results suggest that the gains can be substantial, [see our response to reviewer x5kJ](https://openreview.net/forum?id=wZrOOO9XBn&noteId=qfuh-eeyo_5).

---

> ### Author Response · Authors · 2021-08-10
> **Our method removes significant information from CLIP's reps.**
>
> > good results may simply come from the fact that the latent representations of CLIP [...] already have a low bit-rate.
>
> We investigated this hypothesis on line 283-289 by comparing our lossy CLIP compressor to losslessly compressing CLIP representations. The results show that we can get $11\times$ compression gain for little negative impact, which shows that CLIP retains significant amounts of unnecessary information. Indeed, CLIP is not trained explicitly with a term that minimizes information. We now emphasized that in the text.

---

> ### Author Response · Authors · 2021-08-10
> **CLIP is more compressible than SimCLR (new experiments).**
>
> > It would also be great if the authors demonstrate the performance gain provided by different data-augmentation techniques
> This is a very good point which was indeed missing in the experiments, which we now added.
>
> We now compare the compression performance of frozen, pretrained ResNet50 representations trained with CLIP and with SimCLR ([Chen et al., 2020](https://arxiv.org/abs/2002.05709)). Both use similar contrastive loss and essentially differ in the choice of augmentations. CLIP uses text-image transformation, while SimCLR uses standard image augmentations (resize crop, blur, grayscale, horizontal flip, color jitter). Our theory suggests that CLIP will result in more compressible representations ([see our response to Reviewer x5kJ](https://openreview.net/forum?id=wZrOOO9XBn&noteId=v1Io-5CD-gk)), because the equivalence class of images that correspond to a detailed text description is larger than the equivalence class under standard image augmentations. These experiments used the same staggered setup as Sec. 5.3:
>
> |                 | Augmentation | ImageNet |  STL | PCam | Cars | CIFAR10 | Food | Pets | Caltech |
> |-----------------|--------------|:--------:|:----:|:----:|:----:|:-------:|:----:|:----:|:-------:|
> | Rate [bits/img] | CLIP         |   2108   | 1962 | 1949 | 2421 |   2111  | 1991 | 1867 |   1968  |
> |                 | SimCLR       |   2811   | 2732 | 2769 | 2751 |   2950  | 2077 | 2839 |   2502  |
> | Test Acc. [%]   | CLIP         |   63.2   | 92.0 | 78.6 | 68.0 |   65.5  | 74.1 | 81.8 |   83.0  |
> |                 | SimCLR       |     62.8    | 91.9 | 81.4 | 29.6 |   78.6  | 60.0 | 78.9 |   79.0  |
>
>
> Indeed, we see that for all datasets besides CIFAR10, CLIP augmentation achieves both a better compression and downstream performance. This shows the importance of the choice of augmentations when turning self supervised models to compressors.
>
> We also added a second experiment investigating different augmentations with MNIST. [Please refer to our answer to Reviewer QdsL.](https://openreview.net/forum?id=wZrOOO9XBn&noteId=MSpljSsYyrO)

---

> ### Author Response · Authors · 2021-08-10
> **Thank you, your feedback improved our experimental section.**
>
> Thank you for your many questions and thoughtful review. We have added experiments and clarified the discussion of previous results, which we believe greatly improved our experimental section.
>
> From our reading, you had two important questions that we want to ensure are addressed:
> 1. Are our methods just (implicitly) compressing the labels?
> 2. Can our results be explained just by existing SSL?
>
> To summarize our answers (see posts below):
> 1. Our methods are indeed implicitly attempting to jointly compress the labels of many tasks. The challenge is to do so for *all* (possibly infinitely many) invariant tasks in an *unsupervised*/*self-supervised* fashion.
> 2. Our existing results show that SSL representations retain unnecessary information, and our new results support our theory by giving insight into which SSL representation to choose.

---

> ### Author Response · Authors · 2021-08-19
> **Did we address your initial concern ?**
>
> > I believe this can be a good paper if this concern is addressed, and I would be willing to increase my score if the authors convince me during rebuttal.
>
> Dear reviewer, did our explanations and additional experiments addressed your initial concern?

---

> > ### Comment · Reviewer_YGFN · 2021-08-26
> > **Thanks for your detailed response**
> >
> > Dear authors,
> >
> > Your comprehensive response well addressed my initial concern on the relation between the proposed methods and existing SSL approaches, and the experiments partly addressed by concern on whether the proposed method is simply compressing the labels. As said in your response, your goal is to "compress the labels" for all (possibly infinitely many) invariant tasks in an unsupervised/self-supervised fashion. As mentioned in my initial review, I like this idea and believe that the proposed method could be beneficial in this regard. It's just that I feel the experiment section could benefit from showing the proposed method is actually compressing the labels for "many" invariant tasks. Other than this potential weakness, I believe this is a good paper and thus will increase my score.
> >
> > Best,
> >
> > Reviewer YGFN

---

### Official Review · Reviewer_QdsL · 2021-07-14

**Rating:** 7
**Confidence:** 4

**Summary:**

The author proposes a compression algorithm by extracting invariant information from the data source, which significantly reduces the file size for different downstream tasks that share similar structures while maintaining high prediction accuracy.

**Limitations And Societal Impact:**

yes

**Main Review:**

pros:
1. The idea is interesting and novel. Compared to other compression work that is aimed at reconstruction quality, the author focus on extracting critical (and of course compressible) information from source data; the extracted data can be applied to different downstream tasks without further fine-tuning.
2. The author proposes 2 invariant neural compressors for this compression scheme; both of them show competitive rate-distortion(or rate-accuracy) results on different tasks.
3. Theoretical Justification

questions:
1. In table 3. Higher $\beta$ should correspond to lower distortion, but the table shows higher distortion and lower rate?
2. I wonder how the augmentation methods can influence the rate-distortion performance? The tasks seem to be highly correlated to the augmentation methods we use. For image classification, we can easily use different transforms, but for tasks like image captioning, it could be hard to find the right way to augment the data.
3. The pseudocode for Algorithm 1 is not clear for me. line 5- line 7: $\tilde{\textbf x}$ and $\textbf x$ are not the same thing? To make sure if I understand correctly, $A(x)$ and $A(x^-_i)$ use the same augmentation method and a $x^+$ is augmented differently.


**Time Spent Reviewing:**

3

---

> ### Author Response · Authors · 2021-08-10
> **We agree: finding the right augmentation for a task can be challenging**
>
> > but for tasks like image captioning, it could be hard to find the right way to augment the data.
>
> We agree that finding the right augmentations may not be trivial for some tasks and is a crucial aspect of our model. We extended our discussion in Sec. 7 to mention this possible limitation.
>
> Nevertheless, augmentations are ubiquitous in ML and the community continues to come up with useful task-specific augmentations. For your specific question about image captioning, we believe that our CLIP-based compressor would work extremely well, since the CLIP augmentations encode invariance to transformations from image to paired text.

---

> ### Author Response · Authors · 2021-08-10
> **We clarified Algorithm 1**
>
> > The pseudocode for Algorithm 1 is not clear for me. [...] $A(x)$ and $A(x_i^-)$ use the same augmentation method and a $x^+$ is augmented differently?
>
> We agree that this was not clear and now modified the pseudocode and the explanation next to it. In short, each separate call to $A$ returns an independent augmentation of its input. So, the augmentations are all different transformations sampled independently from the same random process, e.g., rotating by 7, 9, or 23 degrees all sampled uniformly in [0,45].

---

> ### Author Response · Authors · 2021-08-10
> **The augmentations impact the rate-distortion by defining the task-set, in theory and practice (new experiments).**
>
> > I wonder how the augmentation methods can influence the rate-distortion performance? The tasks seem to be highly correlated to the augmentation methods we use.
>
> This is a great question, which we now describe in more detail in the paper.  In short, the augmentation implicitly *defines* the task-set $\mathcal{T}$ on which the compressed representations have guarantees. You can think of $A(X)$ as a transformation that discards information from $X$. $\mathcal{T}$ is then the set of all tasks for which $A(X)$ does not change the task labels. Thus, a larger set of augmentations means a smaller $\mathcal{T}$ and lower rate $Rate(\delta)$.
>
> To illustrate this, we ran VIC and BINCE using three augmentation sets on MNIST, all of which keep the true label invariant, but which progressively remove more information:
> 1. Standard image augmentations such as random translations, shears, and rescalings.
> 2. Those same standard image augmentations, but drawn from larger ranges of possible translations, shears, scales, etc.
> 3. Supervised “augmentations” like in [Khosla et al., 2021](https://arxiv.org/abs/2004.11362), which remove everything except label information i.e., for every image $x$ let $x^+=A(x)$ be a random image with the same label.
>
> |       | Augmentation | Bit-rate | Test Accuracy |
> |-------|--------------|:--------:|:-------------:|
> |  VIC     | Small Set    |   185.3  |      96.2     |
> |       | Large Set       |   79.0   |      97.9     |
> |   | Supervised (Largest Set)    |    5.7   |      99.1     |
> | BINCE   | Small Set    |    256   |      95.3     |
> |       | Large Set    |    131   |      97.8     |
> | | Supervised (Largest Set)   |    5.9   |      97.8     |
>
> As you can see, using label-preserving augmentations that remove more information about X greatly decreases the rate without hindering classification performance. Note that wiping out information about X also helps generalization, which is why test accuracy improves. We also added an experiment comparing different augmentations for pretrained self-supervised models (CLIP vs SimCLR), [see our response to Reviewer YGFN](https://openreview.net/forum?id=wZrOOO9XBn&noteId=uIsg_Ou_OYf).
>
> On the theoretical side, the augmentations directly impact the rate-invariance curve. Specifically, we can rewrite theorem 2 as $Rate(\delta) = H[M(X)] - \delta = I[X,A(X)] - \delta$. I.e. the more information the augmentation removes from $X$ the smaller the rate will be.

---

> ### Author Response · Authors · 2021-08-10
> **Thank you, we made the necessary changes.**
>
> Thank you for your review and useful questions. We made the necessary changes, which we believe improved our paper.
>
> We corrected two typos based on your feedback:
> - We exchanged “high beta” and “low beta” in table 3.
> - On line 6, $p(Z|x)$ should be $p(Z|\tilde{x})$.

---

### Official Review · Reviewer_XSke · 2021-07-16

**Rating:** 7
**Confidence:** 4

**Summary:**

This paper proposes methods to introduce a self-supervised approach to neural image compression for classification tasks.
VIC is a modified neural compressor in which inputs are augmented but target reconstructions are not.
BINCE is a modified neural compressor same as VIC but trained by an entropy bottleneck and an InfoNCE [15] objective. This paper also introduce and evaluate a zero-shot compressor using pre-trained self-supervised models CLIP [38].

**Limitations And Societal Impact:**

The caveats are well described in this paper.

**Main Review:**

I think it is a novel idea to introduce a self-supervised approach to neural image compression for classification tasks and theoretical background is well described.

But I am curious about the following points.
- I don't know the unit of "bit-rate" in table 3 (it not seems to bit per pixel, is it bit per image?). Please write a clear definition.
-  I don't think this paper (the main paper) adequately describes how the quantizer is handled. However, the Appendix describes the quantizer process with a linear transformation using the learnable parameter. Also, you are probably using a uniform noise quantizer [24], but I think these need to be explained in the main paper.

**Time Spent Reviewing:**

8

---

> ### Author Response · Authors · 2021-08-10
> **We gave more detail on the quantization process in the main paper.**
>
> Thank you for your generally positive review. Given the relatively low score, we wonder if you have any feedback that we could address to improve the paper?
>
> Answers to your questions:
>
> > I don't think the main paper adequately describes how the quantizer is handled
>
> As explained on line 212, we use Balle’s standard hyperprior entropy model and quantizer ([Ballé et al., 2018](https://arxiv.org/abs/1802.01436)), which uses a uniform noise quantizer. This was not our contribution, and any entropy model and quantizer that works for neural compressors should work for us. We now provided more details in the paper.
>
> > the Appendix describes the quantizer process with a linear transformation using the learnable parameter.
>
> The linear transformation is only used in our pretrained CLIP experiments in Sec. 5.3. We had to do so because Balle’s entropy model quantizes the input by rounding to the nearest integer. Such rounding does not work well for frozen pretrained models as the scale of the representation cannot be modified.  We use a per-dimension linear transformation to learn the rounding precision without needing to modify the code of the entropy model. We moved this explanation to the main paper to make it more clear.
>
> > unit of "bit-rate" in table 3
>
> Bits per image, which we useto show that our compressor's rate is invariant to number of pixels in an image. We now clarified.

---

> > ### Comment · Reviewer_XSke · 2021-08-26
> > **Thank you for your explanation.**
> >
> > Thank you very much for your explanation. Your helpful explanations have addressed some of my concerns and I'm going to raise my rating.
> >
> > I think that achieving good performance on many future tasks essentially depends on how well we can design invariant under user-defined transformations. As other reviewer has pointed out, it would be beneficial to validate it under more and different tasks for future works.

---

### Official Review · Reviewer_x5kJ · 2021-07-16

**Rating:** 8
**Confidence:** 3

**Summary:**

The paper "Lossy Compression for Lossless Prediction" considers the problem of data compression for downstream predictive tasks, rather than preserving the perceptual quality of reconstructions, in order to achieve significantly higher compression rate with negligible or no performance loss in  downstream tasks. In particular, the paper theoretically characterizes the bit-rate required to ensure high performance on all predictive tasks that are invariant under a set of transformations, such as data augmentation, and derives two corresponding variational objectives, VIC and BINCE, for training end-to-end neural compressors. The method is validated empirically on image tasks, both training from scratch and adapting an existing SOTA vision transformer model with an entropy bottleneck, and is shown to achieve superior compression rates than both traditional and learned image compression baselines.

**Limitations And Societal Impact:**

**Limitations:** See the various points in the "Quality and Clarity" section above.

**Main Review:**

**Originality**

Although the paper is most similar in spirit to "End-to-End Learning of Compressible Features" (Singh et. al., 2020), and in fact Singh et. al.'s method  can be seen as a special case of the proposed VIC method (where there is only one downstream task of interest and thus no data augmentation), the current paper broadens the scope considerably by considering lossy compression for a *collection* of related invariant tasks, and is therefore a novel and original contribution.


**Quality and Clarity**

The paper is clearly written and well-organized. The theoretical results appear to be rigorously derived (in the supplementary material). However, the discussions in the experiments section leave a few things to be desired. In particular,
1. In section 5.2, the worse performance of VIC (e.g., its inability to achieve zero distortion) compared to BINCE can benefit from a better explanation; and more generally, a comparison between the two formulations and discussions on their pros and cons would be beneficial.
The worse performance of VIC was attributed to the channel averaging operation of the ResNet18 encoder used; however, since BINCE does not perform reconstructions (unlike VIC) and thus does not suffer from the poor encoder, it would make a more fair comparison by evaluating VIC performance using the compressed Z representation (with a separately trained predictor) as done for the evaluation of BINCE.  It would be reassuring to see that VIC is able to achieve comparable performance to BINCE, e.g., by increasing the expressivity of the variational family.

2. It's not clear how the performance of zero-shot compressor using a pre-trained SSL model compares to end-to-end training with the proposed method (either VIC or BINCE). The already good zero-shot performance from fitting an entropy model to the latent representations of a pre-trained model (in section 5.2) seems to render end-to-end training with VIC or BINCE unnecessary.

3. It's not clear how the pretraining of CLIP fits into the BINCE framework (e.g., it's not clear what kind of data augmentation transform $A$ is used during training).

**Significance**

As far as I'm aware, this paper is the first to consider the problem of learned lossy compression in a multi-task setting, and lays a solid foundation for future work (both in theory and methods), thus represents a significant contribution.


**Time Spent Reviewing:**

8

---

> ### Author Response · Authors · 2021-08-10
> **We added a comparison to Singh et. al (new experiment).**
>
> > Although the paper is most similar in spirit to Singh et. al., 2020, [...] the current paper broadens the scope considerably by considering [...] a collection of related invariant tasks.
>
> We agree with this statement. Given that this is the most similar paper, we now added an experiment that compares both settings. See results in [our answer to Reviewer YGFN](https://openreview.net/forum?id=wZrOOO9XBn&noteId=Otki8lQMscm).

---

> ### Author Response · Authors · 2021-08-10
> **VIC representations retain the same information as BINCE, but makes it harder to predict from.**
>
> > it would make a more fair comparison by evaluating VIC performance using the compressed Z representation (with a separately trained predictor) as done for the evaluation of BINCE.
>
> We agree with you. This experiment and results can already be found on lines 242-245, and the full results can be seen in table 2 of the appendices (page 35).  Using features instead of reconstructions improves the decrease of test accuracy from $25.1\%$ to $16.1\%$.
>
> Your question made us realize that this should be more prominent, so we have now added a column in Table 1 which shows the performance of VIC when predicting from the representations.
>
> > It would be reassuring to see that VIC is able to achieve comparable performance to BINCE.
>
> We believe that there are two properties of representations $Z$ that are required to achieve good downstream test performance:
> $Z$ needs to retain enough mutual information with the desired labels $Y$;
> the information about task labels needs to be easily separable/decodable.
>
> Our theory only concerns the first point, and we hypothesize that VIC’s representations satisfy (1) but not (2). To evaluate our hypothesis we used a naive sample estimate of the mutual information $I[Z, Y]$ and found that $\hat{I}[Z, Y] = \hat{H}[Y]$. This shows that indeed all information about labels is retained, i.e., no images get compressed to the same Z but have different labels.
>
> This suggests that the issue comes from (2), which, as mentioned above, is not an issue for contrastive learning / BINCE. For more information about decodable/usable information see [Xu et al, 2020](https://arxiv.org/abs/2002.10689) and  [Dubois et al, 2020](https://arxiv.org/abs/2009.12789).

---

> ### Author Response · Authors · 2021-08-10
> **The relative advantages of VIC and BINCE.**
>
> > A comparison between the two formulations and discussions on their pros and cons would be beneficial
>
> We agree and now expanded the current short comparison of BINCE and VIC on lines 203-204.
>
> BINCE advantages:
> + Does not directly predict X, which is better suited when X is high dimensional (e.g., real images).
> + Contrastive learning has been shown in theory ([Arora et al., 2021](https://arxiv.org/abs/1902.09229), [Tosh et al., 2021](https://arxiv.org/abs/2008.10150), [Lee et al., 2020](https://arxiv.org/abs/2008.01064)) and in practice ([Chen et al. 2020](https://arxiv.org/abs/2002.05709), [Oord et al. 2018](https://arxiv.org/abs/1807.03748)) to provide representations that are approximately linearly separable. This ensures that simple classifiers (e.g., linear or finite MLP) can predict very well using BINCE’s representations as input.
>
> VIC advantages:
> + More interpretable as we have access to reconstructions.
> + Does not require negative samples, which for BINCE has to be large. As a result, VIC is more computationally efficient as it does not require large batch sizes.
>
> In the posts below, we attempt to explain the experimental differences that we observed.

---

> ### Author Response · Authors · 2021-08-10
> **CLIP fits the BINCE framework, via image-to-text transformations.**
>
> > It's not clear how the pretraining of CLIP fits into the BINCE framework (e.g., it's not clear what kind of data augmentation transform is used during training).
>
> We discussed the link between CLIP and BINCE on lines 257-262, but we agree that this discussion was unclear. We have now clarified and expanded it.
>
> CLIP’s contrastive loss is essentially the same as the contrastive term in BINCE’s loss (i.e. BINCE without bottleneck) with the following choices:
> 1. **Augmentation:** text to image transformation. CLIP’s dataset contains pairs of associated images and detailed text  $(x_{img},x_{txt})$. The “augmentation” is then a function that maps $x_{img}$ to its associated  $x_{txt}$. Since the preimage of a function always forms an equivalence class, this will partition the images into sets, each of which are associated with a common text description.
> 2. **Discriminator:** a dot product, i.e, $f_{\psi}(Z’,Z) = Z’^\top Z$.
> 3.  **Encoder:** a deterministic function defined by cases, i.e., sampling from $p(Z|X)$ gives $Z = ViT(X)$ if $X$ is an image and $Z = transformer(X)$ if $X$ is a sentence.
>
> The only minor difference is that CLIP performs contrastive learning between text-image and image-text but never text-text and image-image. BINCE would instead make no distinction between modalities. Both are nevertheless valid approximations to $R[M(X)|Z]$.

---

> ### Author Response · Authors · 2021-08-10
> **End-to-end training is much better than staggered training (new experiment).**
>
> > It's not clear how the performance of zero-shot compressor using a pre-trained SSL model compares to end-to-end training with the proposed method
>
> This is a really great point. We agree that the staggered training that we used for the zero-shot compression needs to be compared to our end-to-end method. To address this, we now added the following comparison of the staggered variants of VIC and BINCE on MNIST:
>
> |                 | Bit-rate | Test Accuracy |
> |-----------------|:--------:|:-------------:|
> | Staggered VIC   |    477   |      98.0     |
> | End-End VIC     |    79    |      97.9     |
> | Staggered BINCE |    358   |      97.4     |
> | End-End BINCE   |    131   |      97.8     |
>
> This suggests that end-to-end training can give large gains compared to the staggered method and that our compression gains with CLIP could be even further improved. As we do not have access to CLIP’s dataset, we, unfortunately, cannot train end-to-end CLIP’s compressor but hope that the above results will encourage others to do so.

---

> ### Author Response · Authors · 2021-08-10
> **Your feedback has greatly improved the paper.**
>
> Thank you for your incisive questions. Addressing them has improved the quality of our paper and helped us identify discussions that needed to be expanded.

---

### Author Response · Authors · 2021-08-10
**Response summary**

We thank the reviewers for their insightful feedback on our experimental section.

The reviewers found our paper novel [x5kJ,QdsL,XSke] and well written [x5kJ,QdsL]. They appreciated the idea [YGFN],  and found that we laid a solid foundation for future work [x5kJ] on a problem that is important [YGFN].

Reviewer feedback concerned the experimental section. So, we have put significant effort into adding new experiments and extended the discussion of previous results. The main novelties are the following:

- **The importance of end-to-end training [x5kJ]**: we added experiments showing that  end-to-end training can perform much better than staggered setting.
- **The importance of augmentations [QdsL,YGFN]**: we compared the use of different augmentations in two settings and discussed how it relates to our theory.
- **Added a baseline [YGFN]**: we compared CLIP compressor to the compression of features from a supervised classifier pre-trained on ImageNet.

Given that we see our theory as the bulk of our contribution, we were surprised that some of the reviewers didn’t mention our theory in their summary of our paper. We would like to address this mismatch by improving our writing. While we have already made some modifications to the introduction and abstract, we would appreciate any feedback that the reviewers can give to this end.

These revisions have greatly improved our paper. We thank you for your time, feedback, and contributions.

---

### Decision · Program_Chairs · 2021-09-27

**Decision:**

Accept (Spotlight)

**Comment:**

The paper analyzes "supervised" data compression for downstream predictive tasks, aiming to achieve higher compression rates with negligible or no performance loss. It theoretically characterizes the bitrate required to ensure high performance on all predictive tasks that are invariant under a set of transformations, such as data augmentation, and derives two corresponding variational objectives, VIC and BINCE. The method is validated empirically on image tasks, both training from scratch and adapting an existing SOTA vision transformer model with an entropy bottleneck.

The method can be seen as a generalization of "End-to-End Learning of Compressible Features" (Singh et al., 2020) but significantly expands its scope by considering lossy compression for a collection of related invariant tasks and by providing a rigorous information-theoretical justification of the approach. Original weaknesses pointed out by the reviewers were sufficiently addressed in the review period. Experiments are thorough and exhaustive.